# Noisy Test-Time Adaptation in Vision-Language Models

**Chentao Cao**[1]  **Zhun Zhong**[2][†]  **Zhanke Zhou**[1]  **Tongliang Liu**[3]  **Yang Liu**[4]
**Kun Zhang**[5,6]  **Bo Han**[1][†]

[1]TMLR Group, Department of Computer Science, Hong Kong Baptist University
[2]School of Computer Science and Information Engineering, Hefei University of Technology
[3]Sydney AI Centre, The University of Sydney
[4]Computer Science and Engineering, University of California, Santa Cruz
[5]Mohamed bin Zayed University of Artificial Intelligence  [6]Carnegie Mellon University

## Abstract

Test-time adaptation (TTA) aims to address distribution shifts between source and target data by relying solely on target data during testing. In open-world scenarios, models often encounter noisy samples, i.e., samples outside the in-distribution (ID) label space. Leveraging the zero-shot capability of pre-trained vision-language models (VLMs), this paper introduces *Zero-Shot Noisy TTA* (ZS-NTTA), focusing on adapting the model to target data with noisy samples during test-time in a zero-shot manner. In the preliminary study, we reveal that existing TTA methods suffer from a severe performance decline under ZS-NTTA, often lagging behind even the frozen model. We conduct comprehensive experiments to analyze this phenomenon, revealing that the negative impact of unfiltered noisy data outweighs the benefits of clean data during model updating. In addition, as these methods adopt the adapting classifier to implement ID classification and noise detection sub-tasks, the ability of the model in both sub-tasks is largely hampered. Based on this analysis, we propose a novel framework that decouples the classifier and detector, focusing on developing an individual detector while keeping the classifier (including the backbone) frozen. Technically, we introduce the **Ada**ptive **N**oise **D**etector (AdaND), which utilizes the frozen model's outputs as pseudo-labels to train a noise detector for detecting noisy samples effectively. To address clean data streams, we further inject Gaussian noise during adaptation, preventing the detector from misclassifying clean samples as noisy. Beyond the ZS-NTTA, AdaND can also improve the zero-shot out-of-distribution (ZS-OOD) detection ability of VLMs. Extensive experiments show that our method outperforms in both ZS-NTTA and ZS-OOD detection. On ImageNet, AdaND achieves a notable improvement of $8.32\%$ in harmonic mean accuracy ($\text{Acc}_{\text{H}}$) for ZS-NTTA and $9.40\%$ in FPR95 for ZS-OOD detection, compared to state-of-the-art methods. Importantly, AdaND is computationally efficient and comparable to the model-frozen method. The code is publicly available at: https://github.com/tmlr-group/ZS-NTTA.

## 1 Introduction

Machine learning models suffer performance degradation when the target distribution differs from the source distribution. To mitigate this issue, test-time adaptation (TTA) (Wang et al., 2021; Niu et al., 2023; Wang et al., 2022; Gao et al., 2023a; Liang et al., 2023) has been introduced, aiming to enhance models' generalization to the target distribution in test-time. However, TTA assumes the labels of testing samples are within the in-distribution (ID) label space, which is not practical in an open-world setting (Yang et al., 2022; 2021) where models often encounter noisy samples[1].

This paper introduces the *Zero-Shot Noisy TTA* (ZS-NTTA) setting, which leverages off-the-shelf pre-trained vision-language models (VLMs) (Radford et al., 2021) to adapt target data containing

---

[†]Correspondence to Bo Han (bhanml@comp.hkbu.edu.hk) and Zhun Zhong (zhunzhong007@gmail.com).
[1]Noisy samples refer to data that lie outside the ID label space, whereas clean samples stay within it.

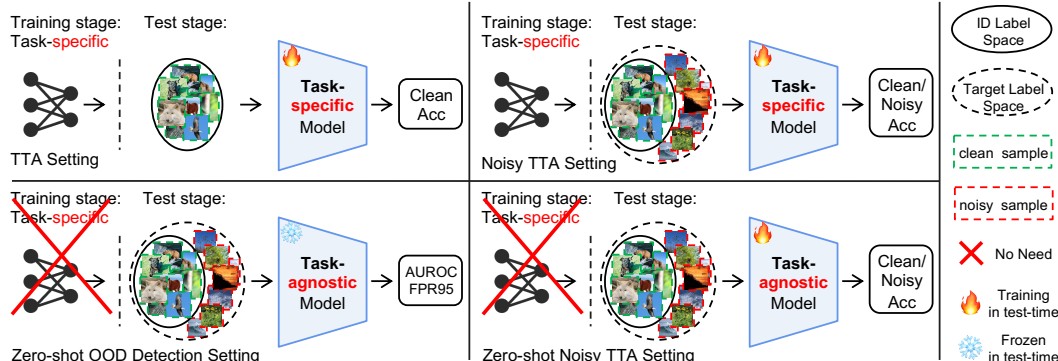

Figure 1: Comparison between TTA, noisy TTA, zero-shot OOD detection, and the proposed zero-shot noisy TTA. Only zero-shot noisy TTA focuses on both clean/noisy classification accuracy and performs in a task-agnostic / zero-shot manner. ZS-NTTA requires online detection of noisy samples.

noisy samples during test-time in a zero-shot way. Different from Zero-Shot Out-Of-Distribution (ZS-OOD) Detection (Ming et al., 2022; Esmaeilpour et al., 2022; Wang et al., 2023), ZS-NTTA requires detecting noisy samples online and emphasizes classification accuracy more. Recently, several works (Li et al., 2023b; Gong et al., 2023) have explored the challenge of noisy samples in TTA, which require task-specific models that are pre-trained with specific source datasets. However, Li et al. (2023b) requires prototypes of the training data, which are unavailable in VLMs. On the other hand, Gong et al. (2023) focuses solely on the classification of clean data, neglecting the recognition of noisy samples. The comparison of different settings is illustrated in Figure 1.

In this paper, we first build the ZS-NTTA benchmarks by leveraging CLIP as the VLM and evaluate the performance of existing TTA methods. We equip each method with the advanced OOD detection technique (Ming et al., 2022) and an adaptive threshold to filter out noisy samples. Figure 2 shows the performance rankings of existing methods in ZS-NTTA across 44 ID-OOD dataset pairs. We find the zero-shot CLIP (ZS-CLIP), which is frozen during adaptation, shows promising performance, particularly in distinguishing between clean and noisy samples. Despite filtering out noisy samples before updating the model, most TTA methods still underperform ZS-CLIP.

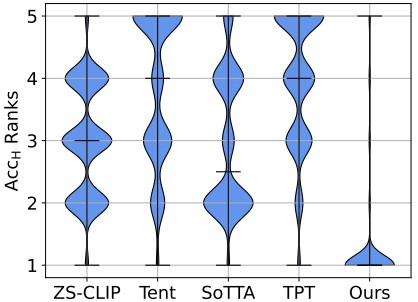

Figure 2: Performance ranking distribution of five TTA methods across 44 ID-OOD dataset pairs. The ranks of different methods on one ID-OOD pair are ranked according to accuracy $Acc_H$. A rank closer to 1 denotes better performance, and a larger bottom area reflects superior overall performance. We also evaluate these methods using absolute accuracy in Figure 7 in Appendix G.

We design three model adaptation pipelines to understand the above phenomenon and analyze the impact of noisy and clean samples on gradients during adaptation. Our findings reveal that noisy samples commonly lead to much larger gradients, often by an order of magnitude, compared to clean samples. Therefore, for methods (Wang et al., 2021) that continuously optimize the parameters during the adaptation, the model is prone to overfitting to noisy samples. Furthermore, even for methods (Shu et al., 2022) that reset parameters at each step, their ability to distinguish between clean and noisy samples will be diminished after each update with noisy data. This underscores the detrimental effect of unfiltered noisy samples on model adaptation, outweighing the benefits of clean samples. Moreover, since these TTA methods implement ID classification and noise detection sub-tasks with the adapting classifier, the ability of models to handle both sub-tasks will be significantly reduced. Thus, we raise a question:

*How to effectively detect noisy samples to mitigate their negative impacts in test-time adaptation?*

To this end, we propose a novel framework inspired by the above observation, which decouples the classifier and detector with a focus on developing an individual detector while keeping the classifier (including the backbone) frozen. This framework offers two key benefits: 1) better distinguishing between noisy and clean samples, and 2) preventing detrimental effects caused by the classifier

adapting to noisy samples. Technically, we propose **Ada**ptive **N**oise **D**etector, termed AdaND. Given that ZS-CLIP can effectively distinguish the most clean and noisy samples, we utilize data filtered by ZS-CLIP to train a detector while keeping the rest of the model frozen during the testing phase. When encountering clean data streams, the detector tends to misclassify numerous clean samples as noisy ones. To handle such a situation, we propose intentionally introducing Gaussian noise during adaption, leading to an effective detector that is robust to both clean and noisy scenarios.

AdaND offers several advantages: 1) *Zero-shot*: By leveraging off-the-shelf VLMs, AdaND can accommodate various ID datasets and scale effectively to ImageNet and its variants; 2) *Noise-agnostic*: AdaND can handle a range of noise scenarios, including various types of noisy samples and different noise ratios (including scenario with exactly clean data); 3) *High-performance*: AdaND exhibits superior performance in ZS-NTTA. In addition, AdaND can extend to ZS-OOD detection task and produce state-of-the-art performance; 4) *Low computational overhead*: The computational cost of AdaND is comparable to that of frozen CLIP. Our contributions can be summarized as follows:

- We propose a more practical setting, *i.e.*, Zero-Shot Noisy TTA (ZS-NTTA), and build benchmarks for evaluation. Based on the built benchmarks, we analyze why adapted methods suffer from performance decline and underperform the model-frozen method in ZS-NTTA (Sec. 2 & Sec. 3).
- We propose AdaND, a simple and effective method that can cover both noisy and clean data streams in ZS-NTTA, offering computational efficiency comparable to model-frozen method (Sec. 4).
- Our method demonstrates superior performance in both ZS-NTTA and ZS-OOD detection tasks. Notably, in ImageNet, AdaND achieves a $8.32\%$ improvement in $\text{Acc}_\text{H}$ compared to existing TTA methods and a $9.40\%$ improvement in FPR95 over current OOD detection methods (Sec. 5).

## 2   Zero-shot Noisy TTA

**Definition of In-Distribution in VLMs.** Following zero-shot OOD detection (Ming et al., 2022; Esmaeilpour et al., 2022; Jiang et al., 2024), in our setting, the in-distribution (ID) classes are defined based on the classification task of interest rather than the classes used in pre-training. Accordingly, noisy samples are defined as data outside the ID label space.

**Problem Formulation.** We define the test set $\mathcal{D} = \{\mathcal{X}, \mathcal{Y}_\text{id} \cup \mathcal{Y}_\text{noisy}\}$, where $\mathcal{X}$ indicates the input space, $\mathcal{Y}_\text{id}$ represents the ID label space, and $\mathcal{Y}_\text{noisy}$ denotes the noisy label space. We are given input samples $\{x_i\} \in \mathcal{X}$, the ID class names $\mathcal{Y}_\text{id} = \{y_1, y_2, ..., y_K\}$ with $K$ classes, and pre-trained VLMs. Owing to being trained on vast amounts of data, VLMs have learned robust feature representations, thereby enabling classification in a zero-shot manner. Due to noisy samples in the test data stream, we first detect whether an input sample is noisy. If the sample is identified as clean, it is classified using the VLM. It is directly categorized without further classification if recognized as a noisy sample.

**Why Investigating ZS-NTTA is Meaningful and Practical.** One cannot ignore the noisy samples in real-world TTA deployment since the real world is open and full of unknown samples. We have demonstrated that the noisy sample is a significant obstacle to existing TTA methods in Sec. 3.2. While ZS-OOD detection (Ming et al., 2022; Jiang et al., 2024; Esmaeilpour et al., 2022) considers noisy samples, it primarily focuses on the model's detection capability rather than improving the classification capability for ID data. More critically, the ID classification in existing ZS-OOD detection methods is typically evaluated in a closed-world setting, assuming a clean data stream. In contrast, ZS-NTTA requires detecting noisy samples online, placing greater emphasis on classification accuracy in open-world settings. We also discuss and compare existing test-time OOD detection work (Fan et al., 2024; Gao et al., 2023b) in Appendix A.1. What's more, leveraging VLMs, ZS-NTTA can be performed in a zero-shot manner, making it more practical than noisy TTA.

**Evaluation Protocol.** We use three metrics to evaluate the performance in ZS-NTTA: $\text{Acc}_\text{S}$, $\text{Acc}_\text{N}$, and $\text{Acc}_\text{H}$. $\text{Acc}_\text{S}$ measures classification accuracy on clean samples[2], $\text{Acc}_\text{N}$ measures detection accuracy on noisy samples, and $\text{Acc}_\text{H}$ is the harmonic mean of $\text{Acc}_\text{S}$ and $\text{Acc}_\text{N}$, providing a balanced measure of both accuracies. The specific formulations of these metrics are as follows:

$$\text{Acc}_\text{S} = \frac{\sum_{x_i, y_i \in \mathcal{D}} \mathbb{1}(y_i = \hat{y}_i) \cdot \mathbb{1}(y_i \in \mathcal{Y}_\text{id})}{\sum_{x_i, y_i \in \mathcal{D}} \mathbb{1}(y_i \in \mathcal{Y}_\text{id})}, \ \text{Acc}_\text{N} = \frac{\sum_{x_i, y_i \in \mathcal{D}} \mathbb{1}(\hat{y}_i \in \mathcal{Y}_\text{nosiy}) \cdot \mathbb{1}(y_i \in \mathcal{Y}_\text{nosiy})}{\sum_{x_i, y_i \in \mathcal{D}} \mathbb{1}(y_i \in \mathcal{Y}_\text{nosiy})}, \ \text{Acc}_\text{H} = 2 \cdot \frac{\text{Acc}_\text{S} \cdot \text{Acc}_\text{N}}{\text{Acc}_\text{S} + \text{Acc}_\text{N}}.$$
(1)

---

[2] Note that, if a clean sample is recognized as a noisy sample, it is wrongly classified.

**Simple Baseline.** We introduce a simple yet effective baseline named ZS-CLIP, employing the MCM (Ming et al., 2022) score as our score function to evaluate the confidence of the model's output for detecting noisy samples. Following zero-shot OOD detection (Ming et al., 2022), we construct the classifier using ID class names and perform classification based on the cosine similarity between the input image feature $\mathcal{I}(x_i)$ and text features $\{\mathcal{T}(t_k)\}_{k=1}^{K}$. We define the cosine similarity between the image and text features as follows: $s_k(x_i) = \frac{\mathcal{I}(x_i) \cdot \mathcal{T}(t_k)}{\|\mathcal{I}(x_i)\| \cdot \|\mathcal{T}(t_k)\|}$. Here, $\mathcal{I}$ denotes the image encoder, and $\mathcal{T}$ signifies the text encoder. $x_i$ represents the input sample, and $t_k$ is the text prompt "`this is a photo of a` $\langle y_k \rangle$" corresponding to the ID class name $y_k$. We can detect the input sample through the noise detector $G(\cdot)$:

$$G_\lambda(x_i) = \begin{cases} \text{Clean} & S(x_i) \geq \lambda \\ \text{Noise} & S(x_i) < \lambda \end{cases}, \quad \text{where} \quad S(x_i) = \max_k \frac{e^{s_k(x_i)/\tau}}{\sum_{j=1}^{K} e^{s_j(x_i)/\tau}}, \quad (2)$$

where $\lambda$ is the threshold, $S(\cdot)$ denotes the MCM score, and $\tau$ is the temperature. If the sample is detected as clean, we then use the text-based classifier to classify it.

**Adaptive Threshold.** Various ID datasets can be encountered in ZS-NTTA, making a fixed threshold $\lambda$ suboptimal. Therefore, an adaptive threshold is a better choice. According to OWTTT (Li et al., 2023b), the distribution of OOD scores follows a bimodal distribution. Based on this observation, Li et al. (2023b) proposes minimizing intra-class variance to determine the adaptive threshold:

$$\min_\lambda \frac{1}{N_{\text{id}}} \sum_i [S(x_i) - \frac{1}{N_{\text{id}}} \sum_j \mathbb{1}(S(x_j) > \lambda)S(x_j)]^2 + \frac{1}{N_{\text{ood}}} \sum_i [S(x_i) - \frac{1}{N_{\text{ood}}} \sum_j \mathbb{1}(S(x_j) \leq \lambda)S(x_j)]^2,$$
$$(3)$$

where $N_{\text{id}} = \sum_i^{N_q} \mathbb{1}(S(x_i) > \lambda)$, $N_{\text{ood}} = \sum_i^{N_q} \mathbb{1}(S(x_i) \leq \lambda)$ and $N_q$ is the length of a queue at test-time to update the score distribution. However, the score in OWTTT relies on source prototypes, which are unavailable in pre-trained VLMs. Here, we propose using the MCM score as an alternative. Furthermore, we conduct experiments with various fixed thresholds ranging from 0.1 to 0.9 to validate the reliability of our adaptive threshold, as detailed in Appendix C. The averaged results across different ID datasets indicate that the adaptive threshold outperforms fixed threshold.

## 3 A COMPREHENSIVE ANALYSIS OF ZERO-SHOT NOISY TTA

In this section, we introduce our ZS-NTTA benchmark and provide a comprehensive analysis of the performance of current TTA methods for this task.

### 3.1 ZERO-SHOT NOISY TTA BENCHMARK

**Benchmark Datasets.** To prevent overlap in label spaces of noisy and clean samples, we use established ID-OOD dataset[3] pairs from standard OOD detection benchmarks. The ID datasets include CIFAR-10/100 (Krizhevsky et al., 2009), CUB-200-2011 (Wah et al., 2011), STANFORD-CARS (Krause et al., 2013), Food-101 (Bossard et al., 2014), Oxford-IIIT Pet (Parkhi et al., 2012), ImageNet (Deng et al., 2009), ImageNet-V2 (Recht et al., 2019), ImageNet-A (Hendrycks et al., 2021b), ImageNet-R (Hendrycks et al., 2021a), and ImageNet-Sketch (Wang et al., 2019). The OOD datasets encompass SVHN (Netzer et al., 2011), LSUN (Yu et al., 2015), iNaturalist (Van Horn et al., 2018), SUN (Xiao et al., 2010), Places (Zhou et al., 2017), and Texture (Cimpoi et al., 2014). The specific ID-OOD pairs are detailed in Table 8 in Appendix D.1.

**Evaluated Methods.** We evaluate ZS-CLIP (Radford et al., 2021), Tent (Wang et al., 2021), SoTTA (Gong et al., 2023), and TPT (Shu et al., 2022) in our benchmarks. ZS-CLIP keeps all parameters frozen and utilizes Eq. (2) to determine whether an input sample $x_i$ belongs to the clean or noisy set. Samples identified as clean, denoted as $x_i'$, are then subjected to further classification. The other methods also utilize Eq. (2) to filter samples, subsequently using $x_i'$ to update the model. Specifically, Tent updates the normalization layers within the image encoder by entropy minimization. SoTTA stores $x_i'$ to a memory bank and selects the highest confidence samples to update the model every 64 steps using entropy-sharpness minimization. TPT applies data augmentation to $x_i'$ and updates the text prompt through entropy minimization.

---

[3] ID datasets and clean datasets are interchangeable, as are OOD datasets and noisy datasets.

Table 1: Failure case study of existing TTA methods with CIFAR-10 as the ID dataset. Green indicates an improvement over ZS-CLIP in average Acc$_H$, while red indicates the opposite.

| Method | SVHN | | | LSUN | | | Texture | | | Places | | | Avg | | |
|---|---|---|---|---|---|---|---|---|---|---|---|---|---|---|---|
| | Acc$_S$ | Acc$_N$ | Acc$_H$ | Acc$_S$ | Acc$_N$ | Acc$_H$ | Acc$_S$ | Acc$_N$ | Acc$_H$ | Acc$_S$ | Acc$_N$ | Acc$_H$ | Acc$_S$ | Acc$_N$ | Acc$_H$ |
| ZS-CLIP | 83.55 | 98.39 | 90.36 | 83.11 | 97.82 | 89.87 | 82.18 | 91.82 | 86.73 | 81.73 | 76.26 | 78.90 | 82.64 | 91.07 | 86.47 |
| Tent (GT) | 90.77 | 96.99 | 93.78 | 90.40 | 93.55 | 91.95 | 90.07 | 90.22 | 90.14 | 89.87 | 74.50 | 81.47 | 90.28 | 88.81 | 89.34 (+2.87%) |
| Tent (Normal) | 87.18 | 52.90 | 65.85 | 89.03 | 73.96 | 80.80 | 89.78 | 88.48 | 89.13 | 88.78 | 65.44 | 75.34 | 88.69 | 70.19 | 77.78 (-8.69%) |
| Tent (All-update) | 81.74 | 43.13 | 56.47 | 80.17 | 55.59 | 65.65 | 89.28 | 84.64 | 86.90 | 87.86 | 56.27 | 68.60 | 84.76 | 59.91 | 69.41 (-17.06%) |
| SoTTA (GT) | 90.45 | 97.47 | 93.83 | 90.03 | 94.88 | 92.39 | 89.68 | 91.39 | 90.53 | 89.30 | 75.96 | 82.09 | 89.87 | 89.92 | 89.71 (+3.25%) |
| SoTTA (Normal) | 90.21 | 81.71 | 85.75 | 90.13 | 91.06 | 90.59 | 89.56 | 90.96 | 90.25 | 89.04 | 74.17 | 80.93 | 89.73 | 84.47 | 86.88 (+0.42%) |
| SoTTA (All-update) | 89.69 | 73.13 | 80.57 | 89.88 | 90.76 | 90.32 | 89.47 | 90.54 | 90.00 | 89.05 | 74.50 | 81.13 | 89.52 | 82.23 | 85.50 (-0.96%) |
| TPT (GT) | 85.86 | 98.46 | 91.73 | 85.86 | 98.00 | 91.53 | 85.19 | 92.30 | 88.60 | 84.88 | 77.33 | 80.93 | 85.45 | 91.52 | 88.20 (+1.73%) |
| TPT (Normal) | 81.76 | 98.85 | 89.50 | 81.53 | 97.93 | 88.98 | 80.43 | 92.11 | 85.87 | 79.88 | 77.18 | 78.51 | 80.90 | 91.52 | 85.72 (-0.75%) |
| TPT (All-update) | 85.18 | 96.98 | 90.70 | 84.84 | 91.15 | 87.88 | 83.92 | 75.36 | 79.41 | 83.59 | 54.11 | 65.69 | 84.38 | 79.40 | 80.92 (-5.55%) |

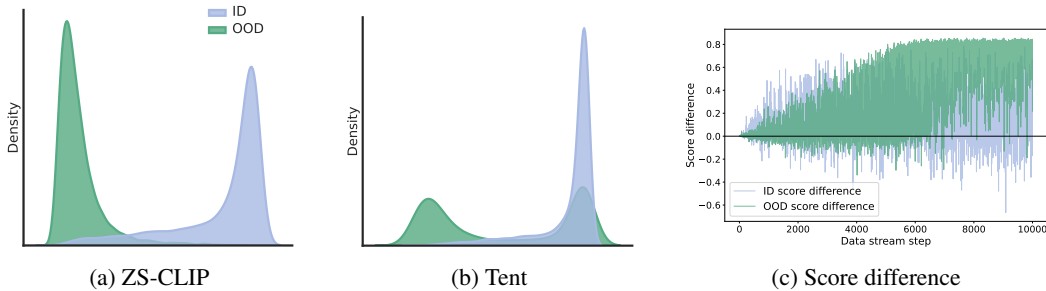

    (a) ZS-CLIP               (b) Tent               (c) Score difference

Figure 3: Failure case analysis of Tent (Wang et al., 2021) in ZS-NTTA. **(a)** and **(b)** show the score distributions of ZS-CLIP and Tent, respectively, revealing that Tent makes it difficult to distinguish between clean and noisy samples. The horizontal axis is the value of OOD score. **(c)** illustrates the score difference between Tent and ZS-CLIP, indicating that the confidence of noisy samples tends to increase in Tent. ID dataset: CIFAR-10; OOD dataset: SVHN.

## 3.2 FAILURE CASE STUDY

In this subsection, we analyze the failure case illustrated in Figure 2, *i.e.*, ZS-CLIP outperforms most tuning-based methods on most ID datasets, highlighting three key observations. We begin by introducing three designed model adaptation pipelines to illustrate the impact of noisy samples on model adaptation (Observation 3.1). Subsequently, we visualize the score difference between ZS-CLIP and tuning-based methods to understand the failure case (Observation 3.2). Finally, we delve into the underlying reasons for the significant negative impact of noisy samples on model adaptation by conducting analyses of the model's gradients (Observation 3.3).

**Observation 3.1.** *Noisy samples have a significant negative impact on model adaptation during TTA.*

To investigate the impact of noisy samples in TTA, we construct three pipelines for each fine-tuning approach: `Ground Truth (GT)`, `Normal`, and `All-update` pipelines. The `GT` pipeline updates the model parameters using only the ground truth clean data, which is unavailable in practice. The `Normal` pipeline updates the parameters using the data filtered by Eq. (2), which may include some noisy data, and this is the pipeline adopted in our main results (Sec. 5). The `All-update` pipeline updates the model parameters using all the available data, *i.e.*, it includes all the noisy data.

Table 1 presents the performance of the three pipelines using CIFAR-10 as the ID dataset. The performance hierarchy observed for most methods is `GT` > ZS-CLIP > `Normal` > `All-update`. This indicates that for the `Normal` pipeline, the negative impact of the unfiltered noisy data on model adaptation outweighs the benefits of the clean data, resulting in performance inferior to that of ZS-CLIP. SoTTA is on par with ZS-CLIP within the `Normal` pipeline due to its refined sample selection for model adaptation. SoTTA employs a memory bank to store high-confidence samples, utilizing only those with the highest confidence samples for updating the model. This strategy effectively filters out the majority of noisy samples, aligning with our assertion that noisy samples significantly and negatively impact model adaptation. Nonetheless, the improvement of SoTTA over ZS-CLIP remains marginal. For failure cases involving more ID datasets, please refer to Appendix G.1.

**Observation 3.2.** *Throughout the model adaptation process in Tent, the scores of noisy samples gradually increase, ultimately rendering the MCM score incapable of distinguishing noisy samples.*

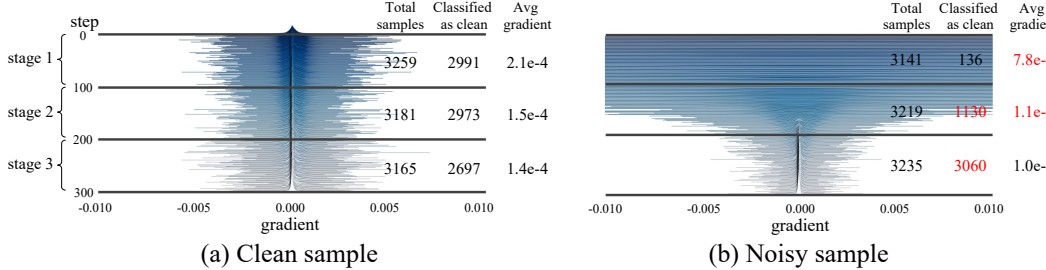

|  | (a) Clean sample | (b) Noisy sample |

Figure 4: The impact of clean and noisy samples on the gradients. Note that the gradients of noisy samples are substantially larger in the first and second stages. The model effectively filters out noisy samples in the first stage but gradually struggles to distinguish between clean and noisy samples. ID dataset: CIFAR-10; OOD dataset: SVHN; Batch size: 64. Please see Figure 9 for an enlarged view.

We show the score distributions for ZS-CLIP and Tent under the `Normal` pipeline in Figures 3a and 3b to better understand the impact of unfiltered noisy samples on model adaptation. Additionally, Figure 3c depicts the score differences for the same input sample between Tent and ZS-CLIP. ZS-CLIP effectively separates ID and OOD score distributions. In contrast, the increase in scores for most noisy samples in Tent makes the distinction between clean and noisy samples difficult. For the analysis of TPT, please refer to Appendix G.2.

**Observation 3.3.** *MCM score with the adaptive threshold can detect most noisy samples during the early stages of TTA in Tent, though some inaccuracies may remain. However, these few inaccuracies during the early TTA stages can gradually lead the model to overfit to noisy samples.*

We analyze the model's gradients in Tent under the `Normal` pipeline to understand why noisy samples negatively impact model adaptation. Figure 4 shows how clean and noisy samples affect the gradients of the final layer normalization in the image encoder during TTA. As for clean samples, the model's gradients gradually decrease and remain relatively stable. The impact of noisy samples on the model's gradients can be roughly divided into three stages.

- **First Stage:** The model effectively filters out noisy samples, with only a minimal number being erroneously classified as clean samples.

- **Second Stage:** The model's performance progressively declines as the impact of noisy samples becomes more apparent. The reliability of the MCM score weakens, and the model increasingly struggles to identify noisy samples. Moreover, the gradient magnitude of the noisy samples remains significant during this stage.

- **Final Stage:** The model overfits to the noisy samples, resulting in a decrease in the model's gradient magnitude. At this stage, it almost loses the ability to distinguish between clean and noisy samples.

Note that TPT resets the model at each step, meaning noisy samples' influence on the model's updates does not be accumulated. As a result, the impact of noisy samples on TPT is relatively smaller compared to Tent. Nonetheless, learning with noisy samples, with model reset at each step, still results in TPT performing worse than ZS-CLIP.

To this end, we naturally consider whether decoupling the classifier and detector might be a superior strategy for the ZS-NTTA task. On one hand, focusing on developing a robust detector can more effectively distinguish noisy samples. On the other hand, keeping the classifier frozen can prevent it from the adverse effects of adapting to noisy samples.

## 4 METHOD

This section demonstrates how to develop the framework that decouples the classifier and detector to better cope with the ZS-NTTA task based on the analysis presented in Sec. 3.2. The proposed framework focuses on training an adaptive noise detector to distinguish noisy samples while keeping the classifier frozen. Specifically, our method consists of two modules: (1) an Adaptive Noise Detector (AdaND), and (2) intentionally injecting Gaussian noise to cover the clean data stream case. The overall framework is illustrated in Figure 5.

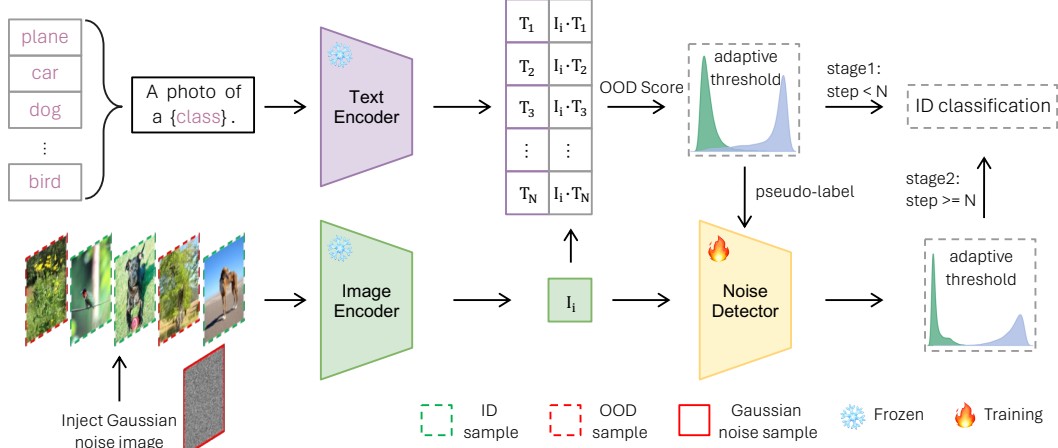

Figure 5: Overview of the proposed framework. We use the detection results from ZS-CLIP as pseudo-labels to train the Adaptive Noise Detector (AdaND). In the early stage, we directly use the ZS-CLIP to distinguish clean-noise samples, while in the later stage, we use the AdaND instead. The predicted clean samples are then classified based on the text-based classifier. To further handle the clean data stream case, we intentionally inject Gaussian noise as additional noisy samples to avoid wrongly assigning too many clean samples as noisy ones.

## 4.1 ADAPTIVE NOISE DETECTOR

We use the image feature $\mathcal{I}(x)$ extracted from the frozen model as the training data during TTA. Given that ZS-CLIP can effectively distinguish most ID and noisy samples, we use the detection results from ZS-CLIP as pseudo-labels in test-time throughout the process. In addition, we employ a single linear layer as the noise detector, leveraging the standard cross-entropy loss for training, i.e., $\mathcal{L} = -\sum_{i=1}^{C} y_i^{\text{pse}} \log(\hat{y}_i), \hat{y}_i = e^{z_i} / \sum_{j=1}^{C} e^{z_j}$. Here, $y_i^{\text{pse}}$ is the pseudo-label generated by ZS-CLIP, $z_j$ denotes the logit of the noise detector for class $i$, and $C = 2$. Our computational overhead is low since only the noise detector is updated during training.

After each training step, the test sample will be re-evaluated for clean-noise detection and classification using its image feature. Since the noise detector may not adapt sufficiently in the early steps of the data stream, we divide the clean-noise detection process into two stages. In the first stage, e.g., for the initial $N$ optimization steps, we use the output from ZS-CLIP as the detection result. In the second stage, we switch to using the output from the noise detector as the detection result. We also use the adaptive threshold in Eq. (3) as the detection threshold rather than directly set $\lambda = 0.5$.

To handle scenarios involving a single input sample, i.e., the batch size is 1, we introduce a queue with a capacity of $L$ to store the outputs from the noise detector. We update the noise detector with the queue's data every $L$ samples, and empty the queue after each update. Note that each sample yields an immediate test result upon input and does not require the accumulation of $L$ samples. What's more, our queue only stores the input features, outputs, and pseudo-labels, ensuring privacy while maintaining minimal and negligible computational and storage overheads.

## 4.2 GAUSSIAN NOISE INJECTING

**How to handle the clean data stream without data stream prior?**    Although the noise detector effectively differentiates between noisy and clean samples within a noisy data stream, it encounters challenges when the test data lacks noisy samples. In such cases, the detector tends to misclassify many clean samples as noisy, leading to a significant drop in performance. To address this, we intentionally inject noise as additional noisy samples to cover the clean data stream case. In this way, all manually injected noise will be included in the adaptive threshold calculation, preventing the misclassification of clean samples as noisy. During testing, we exclusively consider samples from the original data stream, excluding manually injected noise samples.

**How to choose the appropriate noise before inference?**    The injected noise must 1) lie outside the ID label space and 2) be easily accessible without incurring extra costs for auxiliary data collection. The choice of injected noise is flexible; for simplicity and effectiveness, we choose Gaussian noise.

Table 2: Zero-shot noisy TTA results for CUB-200-2011, STANFORD-CARS, Food-101, and Oxford-IIIT Pet as the ID datasets. The **bold** indicates the best performance on each dataset.

| ID | Method | iNaturalist | | | SUN | | | Texture | | | Places | | | Avg | | |
|---|---|---|---|---|---|---|---|---|---|---|---|---|---|---|---|---|
| | | $Acc_S$ | $Acc_N$ | $Acc_H$ | $Acc_S$ | $Acc_N$ | $Acc_H$ | $Acc_S$ | $Acc_N$ | $Acc_H$ | $Acc_S$ | $Acc_N$ | $Acc_H$ | $Acc_S$ | $Acc_N$ | $Acc_H$ |
| CUB-200-2011 | ZS-CLIP | 38.13 | 88.06 | 53.22 | 38.10 | 87.86 | 53.15 | 37.56 | 79.11 | 50.94 | 38.00 | 87.81 | 53.04 | 37.95 | 85.71 | 52.59 |
| | Tent | 37.02 | 46.95 | 41.40 | 38.61 | 55.55 | 45.56 | 34.98 | 41.77 | 38.07 | 40.41 | 74.83 | 52.48 | 37.75 | 54.78 | 44.38 |
| | SoTTA | 41.67 | 84.37 | 55.79 | 42.08 | 86.83 | 56.69 | 41.44 | 77.58 | 54.02 | 42.04 | 86.52 | 56.59 | 41.81 | 83.82 | 55.77 |
| | TPT | 37.41 | 89.57 | 52.78 | 37.49 | 89.67 | 52.87 | 36.88 | **81.67** | 50.81 | 37.44 | 89.45 | 52.79 | 37.30 | 87.59 | 52.31 |
| | AdaND (Ours) | 52.34 | 96.40 | 67.84 | 52.41 | 93.91 | 67.27 | 51.82 | 81.24 | 63.28 | 51.82 | 91.51 | 66.17 | 52.10 | 90.77 | 66.14 |
| STANFORD-CARS | ZS-CLIP | 50.18 | 96.62 | 66.05 | 53.48 | 98.81 | 69.40 | 53.59 | 99.05 | 69.55 | 53.36 | 98.05 | 69.11 | 52.65 | 98.13 | 68.53 |
| | Tent | 44.12 | 52.33 | 47.88 | 54.27 | 94.51 | 68.95 | 54.60 | 97.37 | 69.97 | 54.33 | 96.65 | 69.56 | 51.83 | 85.22 | 64.09 |
| | SoTTA | 51.51 | 92.84 | 66.26 | 54.81 | 97.57 | 70.19 | 55.06 | 98.50 | 70.64 | 54.75 | 96.96 | 69.98 | 54.03 | 96.47 | 69.27 |
| | TPT | 49.24 | 96.97 | 65.31 | 52.40 | 98.83 | 68.49 | 52.75 | 99.27 | 68.89 | 52.42 | 98.39 | 68.40 | 51.70 | 98.36 | 67.77 |
| | AdaND (Ours) | 62.80 | 99.79 | 77.09 | 62.73 | 99.82 | 77.04 | 62.91 | 99.75 | 77.16 | 62.76 | 99.29 | 76.91 | 62.80 | 99.66 | 77.05 |
| Food-101 | ZS-CLIP | 80.60 | 94.76 | 87.11 | 80.75 | 96.08 | 87.75 | 80.51 | 93.12 | 86.36 | 80.62 | 94.62 | 87.06 | 80.62 | 94.65 | 87.07 |
| | Tent | 75.83 | 25.09 | 37.70 | 82.86 | 85.10 | 83.97 | 82.54 | 87.03 | 84.73 | 82.26 | 80.13 | 81.18 | 80.87 | 69.34 | 71.90 |
| | SoTTA | 81.84 | 84.09 | 82.95 | 82.49 | 93.34 | 87.58 | 82.05 | 90.10 | 85.89 | 82.44 | 91.62 | 86.79 | 82.20 | 89.79 | 85.80 |
| | TPT | 79.70 | 94.93 | 86.65 | 79.92 | 96.19 | 87.30 | 79.70 | 93.86 | 86.20 | 79.76 | 95.14 | 86.77 | 79.77 | 95.03 | 86.73 |
| | AdaND (Ours) | 86.50 | 99.87 | 92.71 | 86.40 | 99.64 | 92.55 | 86.44 | 96.51 | 91.20 | 86.42 | 99.40 | 92.46 | 86.44 | 98.85 | 92.23 |
| Oxford-IIIT Pet | ZS-CLIP | 78.58 | 88.30 | 83.16 | 79.75 | 87.30 | 83.35 | 80.20 | 91.16 | 85.33 | 79.59 | 84.17 | 81.82 | 79.53 | 87.73 | 83.41 |
| | Tent | 80.07 | 78.09 | 79.07 | 81.19 | 68.30 | 74.19 | 81.48 | 74.72 | 77.95 | 80.64 | 62.51 | 70.43 | 80.84 | 70.91 | 75.41 |
| | SoTTA | 80.07 | 83.54 | 81.77 | 81.78 | 83.83 | 82.79 | 82.09 | 87.52 | 84.72 | 81.49 | 81.25 | 81.37 | 81.36 | 84.03 | 82.66 |
| | TPT | 77.56 | 89.71 | 83.19 | 78.87 | 89.82 | 83.99 | 79.17 | 92.26 | 85.22 | 78.62 | 87.32 | 82.74 | 78.56 | 89.78 | 83.78 |
| | AdaND (Ours) | 85.81 | 98.78 | 91.84 | 85.82 | 98.19 | 91.59 | 85.86 | 98.68 | 91.82 | 85.88 | 96.58 | 90.92 | 85.84 | 98.06 | 91.54 |

During testing, we insert a Gaussian noise sample for every $M$ input sample in the data stream, regardless of whether the stream is clean or noisy. Note that we don't have prior knowledge about whether the data stream is clean or noisy. The detailed algorithm for AdaND is provided in Algorithm 1 in Appendix D.2.

## 5 EXPERIMENTS

### 5.1 SETUPS

**Compared Methods and Evaluation Metrics.** We compare our method with existing TTA methods mentioned in Sec. 3.1 on the ZS-NTTA task using 11 ID datasets from Sec. 3.1, evaluating with $Acc_S$, $Acc_N$, and $Acc_H$. Additionally, we compare with leading OOD detection methods on the ZS-OOD task, including Energy(Liu et al., 2020), MaxLogit (Hendrycks et al., 2019a), MCM (Ming et al., 2022), CLIPN (Wang et al., 2023), and NegLabel (Jiang et al., 2024), using AUROC and FPR95 as metrics. Please see Appendix D.2 for the implementation details of compared methods.

**AdaND Setups.** In our main results, we maintain consistent hyper-parameters across all datasets. Specifically, we use CLIP (Radford et al., 2021) as our VLM, with ViT-B/16 (Dosovitskiy et al., 2020) as the image encoder and masked self-attention Transformer (Vaswani et al., 2017) as the text encoder, both keeping frozen. We employ a single linear layer as our noise detector, which remains learnable throughout the TTA process. We optimize with Adam (Kingma & Ba, 2014), using a learning rate of 0.0005 and no weight decay. Gaussian noise is injected every 8 samples ($M = 8$). The noise detector's queue length ($L$) is set to 128, and the adaptive threshold's queue length ($N_q$) follows OWTTT (Li et al., 2023b) with a value of 512. We use $N = 10$ for the first stage. As for the ZS-OOD detection task, we use MCM (Ming et al., 2022) score from the output logit of the noise detector as our score function. Unless otherwise specified, we set the batch size ($bs$) to 1 for AdaND.

### 5.2 MAIN RESULTS

**Zero-Shot Noisy TTA Task.** Table 2 and Table 3 present a detailed comparison of ZS-NTTA task results across various ID datasets. On ImageNet, AdaND enhances the average performance by 8.32% in terms of $Acc_H$. Although we filter the data using the MCM score and adaptive threshold, a considerable portion of noisy data remains unfiltered. Consequently, when Tent leverages the filtered data to update the model's normalization layers, it inadvertently causes a substantial performance decline. SoTTA improves data selection by focusing on the highest confidence samples, slightly outperforming ZS-CLIP on some datasets, but the gains are limited. Despite TPT resetting the model before each sample input, the unfiltered noisy data causes TPT to perform worse than ZS-CLIP on most ID datasets. Since our method decouples the classifier and detector, which focuses on developing the noise detector and keeping the classifier frozen, our AdaND can better identify noisy samples and prevent unfiltered ones from affecting the classifier. Due to space constraints, the results for CIFAR-10/100 are provided in Table 10 in Appendix E. In summary, our AdaND demonstrates superior performance over the compared methods, achieving the best results across all datasets.

Table 3: Zero-shot noisy TTA results for ImageNet and its variants as the ID datasets. The **bold** indicates the best performance on each dataset.

| ID | Method | iNaturalist | | | SUN | | | Texture | | | Places | | | Avg | | |
|---|---|---|---|---|---|---|---|---|---|---|---|---|---|---|---|---|
| | | $Acc_S$ | $Acc_N$ | $Acc_H$ | $Acc_S$ | $Acc_N$ | $Acc_H$ | $Acc_S$ | $Acc_N$ | $Acc_H$ | $Acc_S$ | $Acc_N$ | $Acc_H$ | $Acc_S$ | $Acc_N$ | $Acc_H$ |
| ImageNet | ZS-CLIP | 54.01 | 86.53 | 66.51 | 53.43 | 83.96 | 65.30 | 52.71 | 78.52 | 63.08 | 53.35 | 80.50 | 64.17 | 53.38 | 82.38 | 64.77 |
| | Tent | 48.56 | 35.74 | 41.18 | 55.44 | 75.54 | 63.95 | 54.94 | 70.93 | 61.92 | 55.76 | 73.98 | 63.59 | 53.67 | 64.05 | 57.66 |
| | SoTTA | 53.15 | 62.68 | 57.52 | 53.16 | 68.76 | 59.96 | 53.64 | 68.05 | 59.99 | 53.60 | 69.16 | 60.39 | 53.39 | 67.16 | 59.47 |
| | TPT | 52.58 | 88.93 | 66.09 | 51.91 | 86.09 | 64.77 | 51.11 | 80.01 | 62.38 | 51.80 | 82.89 | 63.76 | 51.85 | 84.48 | 64.25 |
| | AdaND (Ours) | **63.26** | **96.87** | **76.54** | **61.34** | **89.44** | **72.77** | **62.45** | **83.54** | **71.47** | **61.92** | **84.82** | **71.58** | **62.24** | **88.67** | **73.09** |
| ImageNet-K | ZS-CLIP | 34.17 | 83.46 | 48.49 | 33.46 | 81.20 | 47.39 | 32.61 | 75.57 | 45.56 | 33.40 | 77.10 | 46.61 | 33.41 | 79.33 | 47.01 |
| | Tent | 30.46 | 26.86 | 28.55 | 36.57 | 71.82 | 48.46 | 36.37 | 66.63 | 47.06 | 36.87 | 70.32 | 48.38 | 35.07 | 58.91 | 43.11 |
| | SoTTA | 36.18 | 61.70 | 45.61 | 36.28 | 67.19 | 47.12 | 35.91 | 65.31 | 46.34 | 36.57 | 67.09 | 47.34 | 36.23 | 65.32 | 46.60 |
| | TPT | 32.16 | 86.52 | 46.89 | 31.55 | 83.86 | 45.85 | 30.74 | **77.39** | 44.00 | 31.56 | 80.05 | 45.27 | 31.50 | 81.95 | 45.50 |
| | AdaND (Ours) | **40.97** | **93.54** | **56.98** | **40.25** | **85.06** | **54.64** | **38.31** | 74.43 | **50.58** | **39.60** | **79.57** | **52.88** | **39.78** | **83.15** | **53.77** |
| ImageNet-A | ZS-CLIP | 34.73 | 80.69 | 48.56 | 34.20 | 78.83 | 47.70 | 33.97 | 76.60 | 47.07 | 33.96 | 75.11 | 46.77 | 34.22 | 77.81 | 47.53 |
| | Tent | 34.99 | 77.19 | 48.15 | 34.83 | 77.05 | 47.97 | 34.36 | 75.19 | 47.17 | 34.60 | 73.83 | 47.12 | 34.70 | 75.81 | 47.60 |
| | SoTTA | 36.85 | 76.83 | 49.81 | 36.47 | 77.08 | 49.51 | 35.60 | 75.37 | 48.36 | 36.07 | 73.87 | 48.47 | 36.25 | 75.79 | 49.04 |
| | TPT | 34.12 | 81.17 | 48.04 | 33.20 | 80.23 | 46.97 | 33.12 | 79.92 | 46.83 | 33.05 | **77.00** | 46.25 | 33.37 | 79.58 | 47.02 |
| | AdaND (Ours) | **43.59** | **91.19** | **58.98** | **41.96** | **80.93** | **55.27** | **45.04** | **79.97** | **57.62** | **42.85** | 72.13 | **53.76** | **43.36** | **81.06** | **56.41** |
| ImageNet-V2 | ZS-CLIP | 48.01 | 85.72 | 61.55 | 47.37 | 83.23 | 60.38 | 46.81 | 77.54 | 58.38 | 47.39 | **79.41** | 59.36 | 47.39 | 81.47 | 59.92 |
| | Tent | 47.94 | 76.98 | 59.08 | 48.28 | 80.50 | 60.36 | 47.56 | 74.47 | 58.05 | 48.34 | 77.37 | 59.50 | 48.03 | 77.33 | 59.25 |
| | SoTTA | 48.24 | 78.59 | 59.78 | 47.80 | 78.67 | 59.47 | 47.27 | 74.82 | 57.94 | 48.26 | 76.05 | 59.05 | 47.89 | 77.03 | 59.06 |
| | TPT | 46.63 | 88.37 | 61.05 | 46.12 | 85.58 | 59.94 | 45.21 | 79.14 | 57.55 | 46.02 | 81.95 | 58.94 | 46.00 | 83.76 | 59.37 |
| | AdaND (Ours) | **56.32** | **97.06** | **71.28** | **54.78** | **86.64** | **67.12** | **57.28** | **80.61** | **66.97** | **55.81** | 79.24 | **65.49** | **56.05** | **85.89** | **67.72** |
| ImageNet-R | ZS-CLIP | 61.99 | 94.39 | 74.83 | 61.82 | 88.95 | 72.94 | 60.91 | 77.05 | 68.04 | 61.68 | 84.86 | 71.44 | 61.60 | 86.31 | 71.81 |
| | Tent | 65.22 | 91.45 | 76.14 | 65.06 | 85.61 | 73.93 | 63.33 | 69.99 | 66.49 | 64.93 | 82.38 | 72.62 | 64.64 | 82.36 | 72.30 |
| | SoTTA | 66.78 | 86.98 | 75.55 | 66.71 | 83.99 | 74.36 | 65.92 | 72.69 | 69.14 | 66.60 | 80.53 | 72.91 | 66.50 | 81.05 | 72.99 |
| | TPT | 60.95 | 94.80 | 74.20 | 60.85 | 89.98 | 72.60 | 59.98 | 77.79 | 67.73 | 60.67 | 85.79 | 71.08 | 60.61 | 87.09 | 71.40 |
| | AdaND (Ours) | **72.21** | **99.59** | **83.72** | **71.02** | **95.94** | **81.62** | **70.44** | **81.43** | **75.54** | **70.85** | **92.14** | **80.10** | **71.13** | **92.28** | **80.25** |

Table 4: Runtime and GPU memory with varying batch sizes ($bs$) on ImageNet for a single sample.

| Resource | ZS-CLIP ($bs = 1$) | SoTTA ($bs = 1$) | TPT ($bs = 1$) | Ours ($bs = 1$) | ZS-CLIP ($bs = 128$) | Tent ($bs = 128$) | Ours ($bs = 128$) |
|---|---|---|---|---|---|---|---|
| Time (s)↓ | 0.1125 | 0.1193 | 0.3219 | 0.1272 | 0.0015 | 0.0037 | 0.0017 |
| Memory (GiB)↓ | 3.80 | 9.13 | 21.23 | 3.83 | 4.54 | 14.99 | 4.57 |

Table 4 shows the time and computational resources required to test a single sample on the ImageNet. All comparisons were conducted on the same 80G A800 GPU. We tested $6,400$ samples and then averaged the results to ensure result stability. Since our method freezes the VLM and uses only a single linear layer for the noise detector, our time consumption is nearly equivalent to ZS-CLIP. Tent's result is reported with $bs = 128$, as performance drops significantly at $bs = 1$ (See Table 11). TPT consumes the most time and memory due to its 64-fold data augmentation and gradient backpropagation through the entire text encoder. Our method proves to be more resource-efficient than Tent, SoTTA, and TPT.

**Zero-Shot OOD Detection Task.** The results for ZS-OOD detection are presented in Table 5, using ImageNet as the ID dataset. Our approach demonstrates competitive performance compared to state-of-the-art OOD detection methods, with significant improvements of $1.37\%$ in AUROC and $9.40\%$ in FPR95. Notably, CLIPN requires an additional dataset to train a text encoder, and NegLabel needs to mine extra semantic information from a large-scale corpus database. In contrast, our method requires no additional external data, making it simpler and more efficient. The results indicate that learning an adaptive noise detector is a simple yet effective strategy for ZS-OOD detection task.

## 5.3 ABLATION STUDIES

**The Effectiveness of Noise Detector and Injected Noise.** We evaluate the effectiveness of the noise detector and Gaussian noise modules under both clean and noisy data streams using ImageNet as the ID dataset, as shown in Table 6. Without Gaussian noise, the noise detector alone is ineffective for clean data streams. When the noise detector is not present, the performance in noisy data streams decreases significantly. Our full method is both effective under clean and noisy data streams, demonstrating the soundness of our design. Results for other ID datasets are in Table 16.

**Intentionally Injected Noise in AdaND.** We conduct ablation studies on intentionally injected noise from two aspects: noise types (Gaussian, Uniform, Salt-and-pepper, Poisson) and injection frequency (every $2$, $4$, or $8$ test samples). As shown in Table 17, all noise types effectively manage both clean and noisy data streams, demonstrating that our method is robust to the choice of injected noise. Table 18 shows the results for noise injection frequency using Gaussian noise. Our experiments show that injecting Gaussian noise every $2$, $4$, or $8$ samples yields excellent performance. Considering efficiency and performance, we choose to inject Gaussian noise every $8$ samples.

Table 5: Zero-shot OOD detection results for ImageNet as the ID dataset. The **bold** indicates the best.

| Method | iNaturalist | | SUN | | Texture | | Places | | Avg | |
|---|---|---|---|---|---|---|---|---|---|---|
| | AUROC↑ | FPR95↓ | AUROC↑ | FPR95↓ | AUROC↑ | FPR95↓ | AUROC↑ | FPR95↓ | AUROC↑ | FPR95↓ |
| Max-Logit | 89.31 | 61.66 | 87.43 | 64.39 | 71.68 | 86.61 | 85.95 | 63.67 | 83.59 | 69.08 |
| Energy | 85.09 | 81.08 | 84.24 | 79.02 | 65.56 | 93.65 | 83.38 | 75.08 | 79.57 | 82.21 |
| MCM | 94.61 | 30.91 | 92.57 | 37.59 | 86.11 | 57.77 | 89.77 | 44.69 | 90.77 | 42.74 |
| CLIPN | 95.27 | 23.94 | 93.93 | 26.17 | 90.93 | 40.83 | 92.28 | 33.45 | 93.10 | 31.10 |
| NegLabel | **99.49** | **1.91** | 95.49 | 20.53 | 90.22 | 43.56 | 91.64 | 35.59 | 94.21 | 25.40 |
| AdaND (Ours) | 98.91 | 4.19 | **95.86** | **17.08** | **93.01** | **21.76** | **94.55** | **20.95** | **95.58** | **16.00** |

Table 6: Ablation studies for each module in the method using ImageNet as the ID dataset. Results in noisy data stream are averaged over four OOD datasets: iNaturalist, SUN, Texture, and Places. '×' indicates the exclusion of a module and '✓' indicates inclusion of a module.

| Noise Detector | Gaussian Noise | Clean Data Stream | | | Noisy Data Stream | | |
|---|---|---|---|---|---|---|---|
| | | $Acc_S$ | $Acc_N$ | $Acc_H$ | $Acc_S$ | $Acc_N$ | $Acc_H$ |
| × | × | 47.68 | - | - | 53.38 | 82.38 | 64.77 |
| × | ✓ | 50.07 | - | - | 53.95 | 81.65 | 64.95 |
| ✓ | × | 37.54 | - | - | 60.64 | **91.73** | 73.00 |
| ✓ | ✓ | **63.96** | - | - | **62.24** | 88.67 | **73.09** |

**Simulating Real-world Adaptation.** We simulate real-world adaptation by mixing ID and OOD datasets from two perspectives. The first involves varying noise ratios ($0\%, 25\%, 50\%, 75\%$) to mimic real-world conditions. The second considers the order of ID and OOD samples, which we simulate using different random seeds. Table 19 presents the results for data streams with varying noise ratios. Since we cannot assume prior knowledge of whether a data stream is clean or contains noisy samples, all methods retain an adaptive OOD detection threshold module. As a result, comparative methods exhibit significant performance degradation on clean data streams. In contrast, our method, which deliberately injects Gaussian noise, effectively addresses clean data streams. Moreover, as the proportion of noise in the data stream increases, most comparative methods show a marked decline in performance, whereas our method continues to deliver strong results across different noise ratios. The experimental results for different random seeds are provided in Table 20 and Table 21. Due to computational constraints, we only conduct experiments using CIFAR-10 and CIFAR-100 as ID datasets, with random seeds ranging from 0 to 4. The results demonstrate that our method consistently achieves superior performance, regardless of the input order of the data streams.

**Hyper-parameters Selection in AdaND.** We conducted ablation experiments in AdaND with varying queue capacities $L$ ($32, 64, 128, 256, 512$). As shown in Table 22, our method demonstrates insensitivity to the choice of $L$, and we selected $L = 128$ for the main experiments Table 23 presents the ablation studies on the queue length $N_q$, which is used to update the score distribution for determining the adaptive threshold. Similar to $L$, AdaND is also robust to changes in $N_q$, and following OWTTT, we set $N_q = 512$. The results for different values of $N$ are shown in Table 24. We found that $N = 10$ optimization steps are sufficient to initialize the noise detector. In summary, AdaND exhibits low sensitivity to hyper-parameter selection, allowing us to use consistent hyper-parameter settings across all datasets, which yields the best results compared to other methods.

We explore the performance of using different backbones in Table 25. Our AdaND is significantly better than the other methods when using different backbones. We also discuss using pseudo-labels generated by the noise detector as pseudo-labels in Appendix F.4. Using noise detector outputs as pseudo-labels can improve performance on some datasets but cause intolerable drops in others.

## 6 CONCLUSION

In this paper, we introduce the Zero-Shot Noisy TTA (ZS-NTTA) setting and construct benchmarks for evaluating this task. We investigate why existing TTA methods fail in this setting by designing three model adaptation pipelines, visualizing the score difference, and analyzing the gradients to understand the impact of noisy samples on model adaptation. Based on these analyses, we propose AdaND, which decouples the classifier and detector, focusing on developing an adaptive detector while keeping the classifier frozen. Our AdaND can handle both noisy and clean data streams by intentionally injecting Gaussian noise, preventing the noise detector from misclassifying excessive ID samples as noise during adaptation. Empirically, our method achieves state-of-the-art results in both ZS-NTTA and ZS-OOD detection tasks. Moreover, our approach is computationally efficient.

## ACKNOWLEDGEMENTS

CTC, ZKZ, and BH were supported by RGC Young Collaborative Research Grant No. C2005-24Y, NSFC General Program No. 62376235, Guangdong Basic and Applied Basic Research Foundation Nos. 2022A1515011652 and 2024A1515012399, HKBU Faculty Niche Research Areas No. RC-FNRA-IG/22-23/SCI/04, and HKBU CSD Departmental Incentive Scheme. TLL was partially supported by the following Australian Research Council projects: FT220100318, DP220102121, LP220100527, LP220200949, and IC190100031.

## ETHICS STATEMENT

This work does not involve potential malicious or unintended uses, fairness considerations, privacy considerations, security considerations, crowdsourcing, or research with human subjects.

## REPRODUCIBILITY STATEMENT

We provide details to reproduce our results in Sec. 2, Sec. 5.1, and Sec. D. We also provide pseudo-code in Algorithm 1, and the code is publicly available at: `https://github.com/tmlr-group/ZS-NTTA`.

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

APPENDIX

## A    DISCUSSION

### A.1    A FURTHER DISCUSSION ON ZS-NTTA SETTING

We have elaborated on the distinctions between ZS-NTTA and ZS-OOD detection in Sec. 2, which primarily lie in task objectives and evaluation settings. *These differences also apply to the comparison between ZS-NTTA and test-time OOD detection, as the latter essentially shares the same objective as classical OOD detection.* We further summarize the task definition differences between ZS-NTTA and Fan et al. (2024); Gao et al. (2023b) in Table 7. In this section, we also discuss and compare existing test-time OOD detection works (Fan et al., 2024; Gao et al., 2023b) regarding methodology. RTL (Fan et al., 2024) used linear regression to make a more precise OOD prediction. In other words, RTL leverages the TTA method to enhance OOD detection while fundamentally remaining an OOD detection task. Different from RTL, we focus on the TTA setting itself, where test samples may contain noise, resulting in severe performance degradation of existing TTA methods. ATTA (Gao et al., 2023b) primarily addresses dense OOD detection in semantic segmentation; however, ATTA cannot be extended to the ZS-NTTA setting since it relies on measuring the distributional distance between test and training features in the normalization layers of the segmentation network. In the context of pretrained VLMs like CLIP, we don't have access to the training data, making ATTA's approach inapplicable to our setting.

In the era of Foundation Models (FMs), we believe noisy TTA can be further explored. The input to FMs may encompass diverse types of noise, including irrelevant or erroneous information (Zhou et al., 2024; Shi et al., 2023), as well as malicious prompts (Wei et al., 2023; Li et al., 2023a), which can significantly undermine the reasoning capabilities of FMs. How to address these noise inputs during testing while enhancing the reasoning capabilities of FMs is an important research direction.

Table 7: Comparison between ZS-NTTA and test-time OOD detection setting (Fan et al., 2024; Gao et al., 2023b).

|  | Fan et al. (2024) | Gao et al. (2023b) | ZS-NTTA |
|---|---|---|---|
| Focus on ID classification | × | × | ✓ |
| Focus on OOD detection | ✓ | ✓ | ✓ |
| Evaluate ID classification | Clean data stream | Clean data stream | Noisy data stream |
| Metrics | AUROC, FPR95 | AUROC, FPR95 | Harmonic mean accuracy ($Acc_H$) |
| Domain shift | × | ✓ | ✓ |
| Online evaluation | × | × | ✓ |
| Zero-shot | × | × | ✓ |

### A.2    LIMITATION

In our ablation study (Table 26), we discussed why we use the outputs of the frozen model as pseudo-labels. However, in practice, using the outputs of the noise detector as pseudo-labels can perform better than the frozen model on most ID datasets. Due to cumulative errors in the noise model's outputs, performance can *significantly drop* on a few ID datasets. Therefore, we chose the frozen model's outputs for a more balanced performance. In future work, we aim to explore how to use the noise detector's outputs as pseudo-labels while ensuring they work well on all datasets, thus achieving stronger performance in the ZS-NTTA task.

Moreover, we utilize the detection results from ZS-CLIP as pseudo labels because CLIP's zero-shot OOD detection capabilities have been thoroughly investigated (Ming et al., 2022; Wang et al., 2023; Jiang et al., 2024; Esmaeilpour et al., 2022) and have demonstrated exceptional accuracy across diverse ID/OOD datasets. However, our method may also falter in scenarios where zero-shot CLIP's detection accuracy is significantly low. Under such circumstances, all existing zero-shot OOD detection methods would also fail. To address this, We may leverage the target data to fine-tune the model, potentially achieving better classification and detection accuracy.

# B RELATED WORK

**Test-time Adaptation.** Test-time adaptation (TTA) (Wang et al., 2021; Liang et al., 2023; Niu et al., 2022; Fleuret et al., 2021; Boudiaf et al., 2022; Prabhudesai et al., 2023; Lee et al., 2024; Gui et al., 2024) aims to bolster a model's generalization to the target distribution. Given the unavailability of source distribution data in the test phase, various TTA methods have been proposed. Some methods (Wang et al., 2021; Niu et al., 2022; Fleuret et al., 2021) leverage self-supervised strategies like entropy minimization, while others employ techniques such as batchnorm statistics adaptation (Schneider et al., 2020; Nado et al., 2020) to improve performance on the target distribution. Some works (Shu et al., 2022; Feng et al., 2023; Karmanov et al., 2024; Samadh et al., 2023; Ma et al., 2023; Zhao et al., 2024; Yoon et al., 2024) tackle the TTA problem with VLMs. TPT (Shu et al., 2022) and DiffTPT (Feng et al., 2023) learn adaptive text prompts with a single test sample employing entropy minimization. TDA (Karmanov et al., 2024) uses a training-free dynamic adapter to enable efficient TTA in vision-language models. However, they did not consider how to handle the presence of noisy samples in the data stream. In this work, we consider the possibility of noisy data streams during the TTA process and cover the clean data stream case.

**Noisy Test-time Adaptation.** Recent works have considered noisy scenarios during the TTA process, and their emphasis has been solely on task-specific models utilizing visual data exclusively. Specifically, SoTTA (Gong et al., 2023) proposed using high-confidence samples to update the model, but they did not consider detecting noisy samples and only focused on the classification accuracy of ID samples. OWTTT (Li et al., 2023b) developed an adaptive threshold strategy for noisy TTA, but OWTTT relies on source domain prototype clustering, which is unavailable for VLMs like CLIP. Lee et al. (2023) proposed utilizing the confidence difference between the original and adaptation models, but Lee et al. (2023) considered the long-term adaptation scenario, and this strategy may not effectively filter out the desired samples in the short-term adaptation scenario. Differing from these works, we introduce the zero-shot noisy TTA setting, which is more practical by leveraging the zero-shot capability of pre-trained VLMs.

**OOD Detection.** Different from the TTA setting, OOD detection (Hendrycks & Gimpel, 2017; Yang et al., 2022; 2021; Fang et al., 2022; Du et al., 2022; Huang & Li, 2021; Hendrycks et al., 2022; 2019b; Sehwag et al., 2021) focuses on data with different label spaces. The goal is to detect OOD samples that are outside the label space of the training set. Most OOD detection methods (Hendrycks & Gimpel, 2017; Hendrycks et al., 2019a; Liu et al., 2020; Ming et al., 2022; Jiang et al., 2024; Esmaeilpour et al., 2022) design a score function based on the confidence of the model's output, implementing detection in a post-hoc manner. While SAL (Du et al., 2024) also leverages unlabeled test data to train robust OOD classifiers, our work differs in its focus and contribution. We primarily address the ZS-NTTA task, where our core contribution lies in proposing a conceptual framework that decouples the detector from the classifier. This decoupling prevents classifier degradation during noisy sample adaptation, with pseudo-label-based detector training serving merely as one implementation detail of our approach. Recent work (Ming et al., 2022; Jiang et al., 2024; Wang et al., 2023; Cao et al., 2024) explores zero-shot OOD detection by leveraging pre-trained VLMs. MCM (Ming et al., 2022) constructs the classifier using ID class names and uses the maximum predicted softmax value between image and text features as the OOD score. CLIPN (Wang et al., 2023) and NegLabel (Jiang et al., 2024) enhance detection performance by mining negative information. EOE (Cao et al., 2024) leverages LLMs' embedded expert knowledge to envision outlier exposure without requiring actual OOD data. Unlike the zero-shot OOD detection setting, ZS-NTTA requires noisy samples to be detected online. What's more, existing OOD detection methods focus more on detecting OOD samples and do not consider how to improve the classification accuracy of ID samples.

**Pre-trained Vision-Language Models.** Pre-trained vision-language models such as CLIP (Radford et al., 2021), ALIGN (Jia et al., 2021), and GroupViT (Xu et al., 2022) typically comprise an image encoder and a text encoder. They are trained on hundred-million-level image-text pair data using self-supervised contrastive learning (Chen et al., 2020). In the testing phase, VLMs encode input images and texts into embedding vectors and then carry out classification by comparing the similarity between image and text features. VLMs demonstrate excellent generalization capabilities due to the broad coverage of the training data distribution and the robust feature representations learned through

contrastive learning. They have also been effectively applied to downstream tasks like image retrieval and image classification in a zero-shot manner.

## C ADAPTIVE THRESHOLD VS. FIXED THRESHOLD

To verify the reliability of the adaptive threshold used in our experiments, we compared the performance of the adaptive threshold with fixed thresholds across various ID datasets, where the fixed threshold ranges from 0.1 to 0.9. Due to time and computational resource limitations, we conduct experiments on the following ID datasets: CIFAR-10, CIFAR-100, CUB-200-2011, STANFORD-CARS, Food-101, and Oxford-IIIT Pet. All ID datasets are tested on their respective four OOD datasets, and the specific ID-OOD dataset correspondences can be found in Table D.1. The average metrics are shown in Fig. 6. It is clear that in terms of $Acc_H$, the adaptive threshold consistently surpassed all fixed thresholds in ZS-CLIP, SoTTA, and our AdaND. The average performance of Tent and TPT using adaptive thresholds is comparable to that of the optimal fixed thresholds. We suppose this is because the classifiers of Tent and TPT experience performance degradation due to noisy samples. Since adaptive thresholds do not require hyperparameter tuning for different ID datasets, we employ the adaptive threshold strategy across all methods.

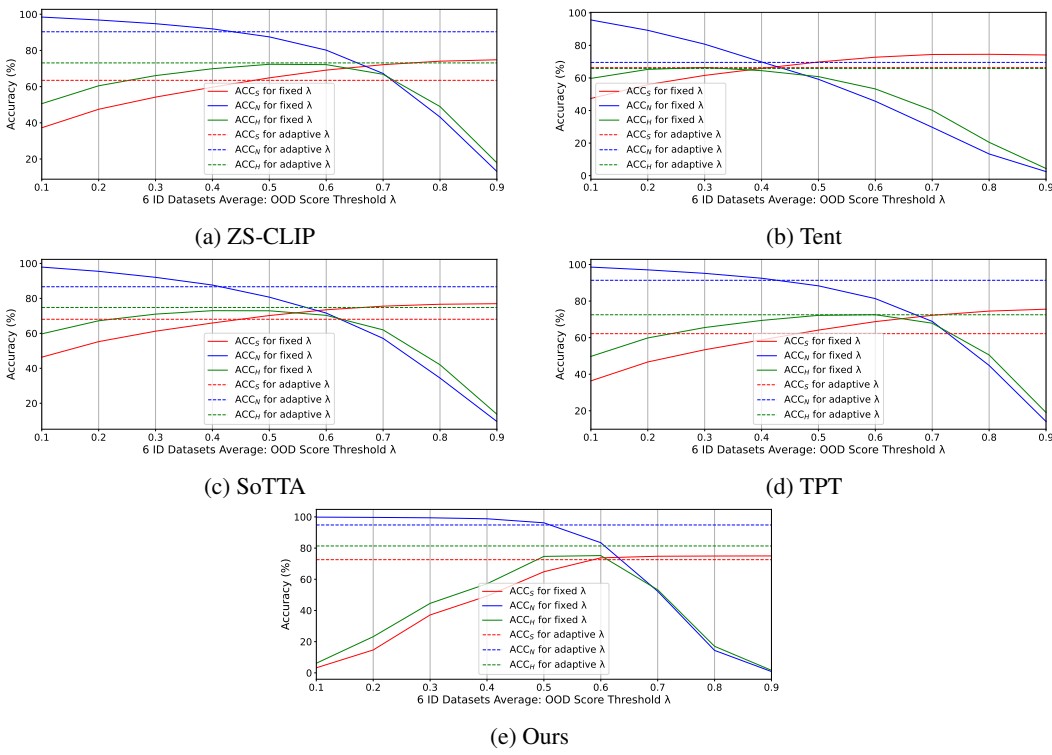

Figure 6: Results about Adaptive threshold(dashed line) and fixed threshold(solid line) range from 0.1 to 0.9. Best viewed with zoom-in.

## D EXPERIMENTAL DETAILS

### D.1 DATASET DETAILS

The division between ID and OOD datasets in the ZS-NTTA benchmarks is shown in Table 8. Note that the label spaces of the ID and OOD datasets do not overlap. We also report the ratio of class numbers between noisy and clean datasets in Table 9. To avoid label space overlap between ID and OOD datasets, the iNaturalist, SUN, and Places datasets used in our experiments are subsets constructed by Huang & Li (2021).

Table 8: ID/OOD Dataset Division. ✓ indicates an ID-OOD pair, while × indicates it is not.

| ID | iNaturalist | SUN | Texture | Places | SVHN | LSUN |
|---|---|---|---|---|---|---|
| CIFAR-10 | × | × | ✓ | ✓ | ✓ | ✓ |
| CIFAR-100 | × | × | ✓ | ✓ | ✓ | ✓ |
| CUB-200-2011 | ✓ | ✓ | ✓ | ✓ | × | × |
| STANFORD-CARS | ✓ | ✓ | ✓ | ✓ | × | × |
| Food-101 | ✓ | ✓ | ✓ | ✓ | × | × |
| Oxford-IIIT Pet | ✓ | ✓ | ✓ | ✓ | × | × |
| ImageNet | ✓ | ✓ | ✓ | ✓ | × | × |
| ImageNet-K | ✓ | ✓ | ✓ | ✓ | × | × |
| ImageNet-A | ✓ | ✓ | ✓ | ✓ | × | × |
| ImageNet-V2 | ✓ | ✓ | ✓ | ✓ | × | × |
| ImageNet-R | ✓ | ✓ | ✓ | ✓ | × | × |

Table 9: Number of classes in ID and OOD datasets. Each row shows an ID-OOD dataset pair with their respective number of classes.

| ID | iNaturalist | SUN | Texture | Places | SVHN | LSUN |
|---|---|---|---|---|---|---|
| CIFAR-10 | × | × | 10:47 | 10:50 | 10:10 | 10:10 |
| CIFAR-100 | × | × | 100:47 | 100:50 | 100:10 | 100:1 |
| CUB-200-2011 | 200:110 | 200:50 | 200:47 | 200:50 | × | × |
| STANFORD-CARS | 196:110 | 196:50 | 196:47 | 196:50 | × | × |
| Food-101 | 101:110 | 101:50 | 101:47 | 101:50 | × | × |
| Oxford-IIIT Pet | 37:110 | 37:50 | 37:47 | 37:50 | × | × |
| ImageNet | 1000:110 | 1000:50 | 1000:47 | 1000:50 | × | × |
| ImageNet-K | 1000:110 | 1000:50 | 1000:47 | 1000:50 | × | × |
| ImageNet-A | 200:110 | 200:50 | 200:47 | 200:50 | × | × |
| ImageNet-V2 | 1000:110 | 1000:50 | 1000:47 | 1000:50 | × | × |
| ImageNet-R | 200:110 | 200:50 | 200:47 | 200:50 | × | × |

## D.2 IMPLEMENTATION DETAILS

For the ZS-NTTA task, we integrated the advanced OOD detection method, $i.e.$, MCM (Ming et al., 2022), into each comparative approach to filter out noisy samples. For the Tent and TPT methods, all hyperparameter settings are kept consistent with their original papers. And we use the layer normalization in Tent when the image encoder is ViT-B/16 or ViT-L/14. For the SoTTA method, considering the generalization across ID datasets and based on the performance of different thresholds in the memory bank, we set the confidence level of the memory bank to $0.5$. For the ZS-OOD detection task, we directly used the results reported in MCM, CLIPN, and NegLabel. Max-Logit and Energy are implemented by ourselves based on CLIP backbone. Additionally, to clearly illustrate our method, we present AdaND in Algorithm 1.

## D.3 ENVIRONMENT

The experiments presented in this paper are conducted utilizing PyTorch 1.13 (Paszke et al., 2019) and Python 3.10.8 within an Ubuntu 22.04 LTS environment, running on NVIDIA A100 80GB PCIe GPUs and AMD EPYC 7H12 CPU.

## E ADDITIONAL RESULTS

We report the main results of CIFAR-10 and CIFAR-100 in Table 10 due to the space limitation in the main text. Compared to other methods, our AdaND achieves the best performance in these two datasets. When using layer normalization, Tent supports $bs = 1$. We conducted experiments with Tent ($bs = 1$) in Table 11 and found that it performs well on clean data streams. However, its performance degrades significantly when dealing with noisy data streams.

We conduct additional experiments on more complex datasets, with results shown in Table 12. The results demonstrate that our method can outperform all the baseline methods. What's more, our method achieves the best ID classification accuracy Acc$_S$ among all approaches. Note that ZS-NTTA is inherently more challenging than traditional OOD detection, as it requires simultaneous classification and detection capabilities under the noisy data stream. Specifically, ZS-NTTA requires online, real-time classification and detection results: for each input sample, the model must immediately

---

**Algorithm 1** AdaND for ZS-NTTA and ZS-OOD detection tasks.

---

**Require:** test data stream $\{x_i\}_{i=1}^{T}$, ID class names $\mathcal{Y}_{\text{id}}$, text encoder $\mathcal{T}$, image encoder $\mathcal{I}$, noise detector $f$, queue $Q$ with capacity $L$, $K = \text{len}(\mathcal{Y}_{\text{id}})$, temperature $\tau = 0.01$, $M = 8$.

1: **for** test-time $i \in \{1, \cdots, T\}$ **do**
2:     Calculate cosine similarity scores:
3:     $\{s_k(x_i) \leftarrow \frac{\mathcal{I}(x_i) \cdot \mathcal{T}(t_k)}{\|\mathcal{I}(x_i)\| \cdot \|\mathcal{T}(t_k)\|}\}_{i=1}^{K}, \quad t_k \in \mathcal{Y}_{\text{id}}$
4:     Calculate OOD score:
5:     $S(x_i) \leftarrow \max_k \frac{e^{s_k(x_i)/\tau}}{\sum_{j=1}^{K} e^{s_j(x_i)/\tau}}$
6:     Calculate $\lambda_{\text{ZS-CLIP}}$ by Eq. 3
7:     **if** $S(x_i) > \lambda_{\text{ZS-CLIP}}$ **then**
8:         $y_i^{\text{pse}} = 1$                                  ▷ Pseudo-label: clean sample.
9:     **else**
10:         $y_i^{\text{pse}} = -1$                                ▷ Pseudo-label: noisy sample.
11:     $\text{logit} = f(\mathcal{I}(x_i))$
12:     Update queue $Q$:
13:     $Q \leftarrow Q \cup \{\mathcal{I}(x_i), \text{logit}, y_i^{\text{pse}}\}$
14:     **if** $\text{len}(Q) = L$ **then**
15:         Train noise detector $f$:
16:         Calculate loss $\mathcal{L}$ using standard CE loss, input data: $Q$
17:         Update $f$ using $\mathcal{L}$
18:         $Q \leftarrow \emptyset$                                     ▷ Empty queue $Q$.
19:     **if** $i \bmod M = 0$ **then**                         ▷ Gaussian noise injection.
20:         $g \sim \mathcal{N}(0, 1)$
21:         Add noise sample to queue $Q$:
22:         $\text{logit}_{g_i} = f(\mathcal{I}(g))$
23:         $Q \leftarrow Q \cup \{\mathcal{I}(g), \text{logit}_g, -1\}$
24:         **if** $\text{len}(Q) = L$ **then**
25:             Train noise detector $f$:
26:             Calculate loss $\mathcal{L}$ using standard CE loss, input data: $Q$
27:             Update $f$ using $\mathcal{L}$
28:             $Q \leftarrow \emptyset$
29:     Generate output:
30:     **if** $i < N$ **then**                               ▷ Stage 1: use ZS-CLIP.
31:         $\text{output} \leftarrow \arg\max_k \frac{e^{s_k(x_i)/\tau}}{\sum_{j=1}^{K} e^{s_j(x_i)/\tau}}$ if $S(x_i) > \lambda_{\text{ZS-CLIP}}$ else $-1$
32:     **else**                                     ▷ Stage 2: use noise detector.
33:         $S(x_i) \leftarrow \max_k \frac{e^{z_k}}{\sum_{j=1}^{2} e^{z_j}}$
34:         Calculate $\lambda_{\text{AdaND}}$ by Eq. 3
35:         $\text{output} \leftarrow \arg\max_k \frac{e^{s_k(x_i)/\tau}}{\sum_{j=1}^{K} e^{s_j(x_i)/\tau}}$ if $S(x_i) > \lambda_{\text{AdaND}}$ else $-1$
    **return** output

---

Table 10: Zero-shot noisy TTA results for CIFAR-10/100 as the ID dataset. The **bold** indicates the best performance on each dataset.

| ID | Method | SVHN | | | LSUN | | | Texture | | | Places | | | Avg | | |
|---|---|---|---|---|---|---|---|---|---|---|---|---|---|---|---|---|
| | | Acc$_S$ | Acc$_N$ | Acc$_H$ | Acc$_S$ | Acc$_N$ | Acc$_H$ | Acc$_S$ | Acc$_N$ | Acc$_H$ | Acc$_S$ | Acc$_N$ | Acc$_H$ | Acc$_S$ | Acc$_N$ | Acc$_H$ |
| CIFAR-10 | ZS-CLIP | 83.55 | 98.39 | 90.36 | 83.11 | 97.82 | 89.87 | 82.18 | 91.82 | 86.73 | 81.73 | 76.26 | 78.90 | 82.64 | 91.07 | 86.47 |
| | Tent | 87.18 | 52.90 | 65.85 | 89.03 | 73.96 | 80.80 | 89.78 | 88.48 | 89.13 | 88.78 | 65.44 | 75.34 | 88.69 | 70.19 | 77.78 |
| | SoTTA | **90.21** | 81.71 | 85.75 | **90.13** | 91.06 | 90.59 | 89.56 | 90.96 | 90.25 | 89.04 | 74.17 | 80.93 | **89.73** | 84.47 | 86.88 |
| | TPT | 81.76 | 98.85 | 89.50 | 81.53 | 97.93 | 88.98 | 80.43 | 92.11 | 85.87 | 79.88 | 77.18 | 78.51 | 80.90 | 91.52 | 85.72 |
| | AdaND (Ours) | 89.46 | **99.90** | **94.39** | 88.56 | **99.66** | **93.78** | 89.60 | **98.54** | **93.86** | 89.65 | **93.04** | **91.31** | 89.32 | **97.79** | **93.34** |
| CIFAR-100 | ZS-CLIP | 48.52 | 97.58 | 64.81 | 49.29 | 94.97 | 64.90 | 46.76 | 81.58 | 59.45 | 45.36 | 64.52 | 53.27 | 47.48 | 84.66 | 60.61 |
| | Tent | 55.39 | 42.41 | 48.04 | 60.06 | 83.37 | 69.82 | 59.31 | 79.13 | 67.80 | 57.52 | 62.24 | 59.79 | 58.07 | 66.79 | 61.36 |
| | SoTTA | 60.56 | 89.24 | 72.15 | 60.28 | 88.89 | 71.84 | 58.79 | 81.56 | 68.33 | 57.01 | 65.73 | **61.06** | 59.16 | 81.36 | 68.34 |
| | TPT | 46.09 | 97.87 | 62.67 | 46.90 | 95.36 | 62.88 | 43.87 | 83.10 | 57.42 | 42.48 | **66.86** | 51.95 | 44.84 | **85.80** | 58.73 |
| | AdaND (Ours) | **64.44** | **99.78** | **78.31** | **62.42** | **99.15** | **76.61** | **65.17** | 84.84 | **73.72** | **63.50** | 44.21 | 52.13 | **63.88** | 81.99 | **70.19** |

Table 11: Performance of Tent with Layer Normalization ($bs = 1$) on Clean and Noisy Data Streams

| ID | Method | Clean Data Stream | | | Noisy Data Stream | | |
|---|---|---|---|---|---|---|---|
| | | Acc$_S$ | Acc$_N$ | Acc$_H$ | Acc$_S$ | Acc$_N$ | Acc$_H$ |
| CIFAR-10 | ZS-CLIP | 77.96 | - | - | 82.64 | 91.07 | 86.47 |
| | Tent (bs=1) | 91.84 | - | - | 89.83 | 7.46 | 13.66 |
| | Tent (bs=64) | 84.39 | - | - | 88.69 | 70.19 | 77.78 |
| CIFAR-100 | ZS-CLIP | 44.69 | - | - | 47.48 | 84.66 | 60.61 |
| | Tent (bs=1) | 63.66 | - | - | 37.86 | 20.64 | 19.82 |
| | Tent (bs=64) | 41.90 | - | - | 58.07 | 66.79 | 61.36 |

determine whether it is ID/OOD, and if ID, perform classification. In contrast, existing OOD detection methods typically report ID classification accuracy under the assumption of a clean data stream.

Table 12: Zero-shot noisy TTA results on more complex datasets. The ID dataset is ImageNet, and the OOD dataset is NINCO (Bitterwolf et al., 2023).

| Method | Acc$_S$ | Acc$_N$ | Acc$_H$ |
|---|---|---|---|
| ZS-CLIP | 51.44 | 71.90 | 59.97 |
| Tent | 54.14 | 65.84 | 59.42 |
| SoTTA | 52.87 | 60.50 | 56.43 |
| Ours (With Gaussian noise) | **60.10** | 55.70 | 57.82 |
| Ours (Without Gaussian noise) | 50.25 | **77.99** | **61.12** |

We also compare our approach to recent training-free TTA work, TDA (Karmanov et al., 2024), in Table 13. Our experimental results demonstrate that TDA's performance is inferior to ours. This indicates the necessity of training a noise detector to detect noisy samples in the ZS-NTTA setting.

Furthermore, our method is designed to be plug-and-play, making it naturally compatible with existing TTA methods to enhance ID classifiers in noisy data streams. To validate this compatibility, we integrate our method with Tent. Specifically, after updating the OOD detector for $N$ steps ($N = 10$ in our implementation), the samples identified as ID by the detector are utilized to update the classifier. As shown in Table 14, the experimental results demonstrate that our method not only improves OOD detection accuracy but also enhances the classifier's intrinsic classification capabilities on ID samples under noisy data streams. These results validate that our approach effectively enhances the robustness of existing TTA methods when dealing with noisy data streams.

To comprehensively evaluate our method's performance, we conduct extensive experiments on widely-used corruption benchmarks, including ImageNet-C, CIFAR10-C, and CIFAR100-C (Hendrycks & Dietterich, 2019). Due to computing resource limitations, we evaluate our method using iNaturalist as the OOD dataset for ImageNet-C experiments, and SVHN as the OOD dataset for both CIFAR-10-C and CIFAR-100-C experiments, with corruption severity set to level-1. The experimental results in Table 15 demonstrate that our method consistently outperforms existing approaches on these corrupted datasets.

Table 13: Performance comparison with TDA using ImageNet as ID dataset. Results are averaged across four OOD datasets: iNaturalist, SUN, Texture, and Places.

| Method | Acc$_S$ | Acc$_N$ | Acc$_H$ |
|---|---|---|---|
| ZS-CLIP | 53.38 | 82.38 | 64.77 |
| TDA | 53.47 | 82.37 | 64.84 |
| Ours (With Gaussian noise) | **62.24** | 88.67 | **73.09** |
| Ours (Without Gaussian noise) | 60.64 | **91.73** | 73.00 |

Table 14: Results of integrating AdaND (Ours) with Tent for enhanced classifiers' classification performance in noisy data streams. Results are averaged across four OOD datasets (SVHN, LSUN, Texture, and Places) with CIFAR-10 as the ID dataset.

| Method | Acc$_S$ | Acc$_N$ | Acc$_H$ |
|---|---|---|---|
| ZS-CLIP | 82.64 | 91.07 | 86.47 |
| Tent | 88.69 | 70.19 | 77.78 |
| Ours | 89.32 | **97.79** | 93.34 |
| Ours (with Tent) | **93.61** | 94.30 | **93.79** |

# F    FULL RESULTS OF ABLATION STUDIES

## F.1    ABLATION STUDIES ON THE MODULES OF ADAND

Table 16 presents the ablation study results on each module in our method across different ID datasets. Experiments show that the noise detector is effective across different ID datasets, and after injecting Gaussian noise, the noise detector can also handle the clean stream case well. Additionally, injecting Gaussian noise does not result in a performance drop for our method on noisy data streams.

## F.2    ABLATION STUDIES ON THE INTENTIONALLY INJECTED NOISE

Table 17 presents the results for Gaussian, Uniform, Salt-and-pepper, and Poisson noise as the injected noise types. The results demonstrate that all noise types effectively manage both clean and noisy data streams, suggesting that our method is robust to different choices of injected noise. Table 18 presents the ablation results for varying frequencies of Gaussian noise injection. Our experiments indicate that injecting Gaussian noise every 2, 4, or 8 samples consistently produces strong performance.

## F.3    ABLATION STUDIES ON SIMULATING REAL-WORLD ADAPTATION

Table 19 shows the results for the zero-shot noisy TTA task across data streams with varying noise ratios. All competing methods show significant performance degradation on clean data streams while our AdaND effectively handles clean data streams. What's more, our method consistently achieves strong results across different noise levels. The results of different orders are shown in Table 20 and Table 21. Experiments demonstrate that our method consistently achieves top performance, regardless of the data stream's input order.

## F.4    ABLATION STUDIES ON HYPER-PARAMETERS SELECTION

Ablation studies on varying queue capacities $L$, queue lengths $N_q$, and optimization steps $N$ are presented in Table 22, Table 23, and Table 24, respectively. The results demonstrate that AdaND is robust to changes in these hyper-parameters. For the main experiments, we set $L = 128$, $N_q = 512$, and found that $N = 10$ optimization steps are sufficient to initialize the noise detector. Overall, AdaND shows low sensitivity to hyper-parameter choices, achieving optimal performance across all datasets. Table 25 explores the performance with different backbones. Our AdaND consistently outperforms other methods across all backbone configurations.

Intuitively, once the noise detector outperforms ZS-CLIP in detection results, it would be more accurate to use its outputs as pseudo-labels. We conducted experiments with various noise ratios and ID datasets in Table 26. Although using the outputs of the noise detector as pseudo-labels

Table 15: Experiments on CIFAR-10-C, CIFAR-100-C, and ImageNet-C. Results are averaged across 15 corruption types.

| Method | Acc$_S$ | Acc$_N$ | Acc$_H$ | Acc$_S$ | Acc$_N$ | Acc$_H$ | Acc$_S$ | Acc$_N$ | Acc$_H$ |
|---|---|---|---|---|---|---|---|---|---|
| ZS-CLIP | 74.14 | 98.24 | 83.96 | 39.43 | 96.74 | 55.49 | 44.14 | 84.73 | 57.91 |
| Tent | 80.25 | 57.86 | 66.83 | 45.29 | 41.15 | 42.83 | 37.10 | 16.50 | 22.79 |
| SoTTA | **83.68** | 85.00 | 84.06 | 51.46 | 88.70 | 64.73 | 46.41 | 62.48 | 53.21 |
| Ours | 82.15 | **99.89** | **89.84** | **54.89** | **99.18** | **70.18** | **52.63** | **94.78** | **67.55** |

Table 16: Ablation studies for each module in the method. For CIFAR-10/100, results are averaged across four OOD datasets: SVHN, LSUN, Texture, and Places. For other ID datasets, averaging includes four OOD datasets: iNaturalist, SUN, Texture, and Places. '×' indicates the exclusion of a module and '✓' indicates inclusion of a module.

| ID | Noise Detector | Gaussian Noise | Clean Data Stream | | | Noisy Data Stream | | |
|---|---|---|---|---|---|---|---|---|
| | | | Acc$_S$ | Acc$_N$ | Acc$_H$ | Acc$_S$ | Acc$_N$ | Acc$_H$ |
| CIFAR-10 | × | × | 77.96 | - | - | 82.64 | 91.07 | 86.47 |
| | × | ✓ | 82.07 | - | - | 83.19 | 90.39 | 86.40 |
| | ✓ | × | 67.89 | - | - | 89.14 | **98.37** | **93.51** |
| | ✓ | ✓ | **89.16** | - | - | **89.32** | 97.79 | 93.34 |
| CIFAR-100 | × | × | 44.69 | - | - | 47.48 | 84.66 | 60.61 |
| | × | ✓ | 43.40 | - | - | 46.07 | 85.84 | 59.77 |
| | ✓ | × | 35.21 | - | - | 61.65 | **90.42** | **73.23** |
| | ✓ | ✓ | **62.52** | - | - | **63.88** | 81.99 | 70.19 |
| CUB-200-2011 | × | × | 33.08 | - | - | 37.95 | 85.71 | 52.59 |
| | × | ✓ | 36.74 | - | - | 40.34 | 82.82 | 54.22 |
| | ✓ | × | 30.01 | - | - | 50.42 | **93.49** | 65.51 |
| | ✓ | ✓ | **49.47** | - | - | **52.10** | 90.77 | **66.14** |
| STANFORD-CARS | × | × | 39.02 | - | - | 52.65 | 98.13 | 68.53 |
| | × | ✓ | 44.08 | - | - | 53.69 | 97.81 | 69.32 |
| | ✓ | × | 34.80 | - | - | 62.59 | **99.67** | 76.89 |
| | ✓ | ✓ | **58.53** | - | - | **62.80** | 99.66 | **77.05** |
| Food-101 | × | × | 72.93 | - | - | 80.62 | 94.65 | 87.07 |
| | × | ✓ | 77.61 | - | - | 80.79 | 94.56 | 87.13 |
| | ✓ | × | 56.75 | - | - | **86.46** | **99.00** | **92.30** |
| | ✓ | ✓ | **86.21** | - | - | 86.44 | 98.85 | 92.23 |
| Oxford-IIIT Pet | × | × | 70.17 | - | - | 79.53 | 87.73 | 83.41 |
| | × | ✓ | 78.41 | - | - | 80.47 | 86.39 | 83.31 |
| | ✓ | × | 62.95 | - | - | 85.54 | **98.27** | 91.47 |
| | ✓ | ✓ | **84.91** | - | - | **85.84** | 98.06 | **91.54** |
| ImageNet | × | × | 47.68 | - | - | 53.38 | 82.38 | 64.77 |
| | × | ✓ | 50.07 | - | - | 53.95 | 81.65 | 64.95 |
| | ✓ | × | 37.54 | - | - | 60.64 | **91.73** | 73.00 |
| | ✓ | ✓ | **63.96** | - | - | **62.24** | 88.67 | **73.09** |
| ImageNet-K | × | × | 30.48 | - | - | 33.41 | 79.33 | 47.01 |
| | × | ✓ | 31.43 | - | - | 33.72 | 78.67 | 47.20 |
| | ✓ | × | 26.03 | - | - | 37.70 | **88.27** | 52.82 |
| | ✓ | ✓ | **36.54** | - | - | **39.78** | 83.15 | **53.77** |
| ImageNet-A | × | × | 31.47 | - | - | 34.22 | 77.81 | 47.53 |
| | × | ✓ | 34.03 | - | - | 36.32 | 74.85 | 48.90 |
| | ✓ | × | 26.13 | - | - | 39.39 | **90.47** | 54.87 |
| | ✓ | ✓ | **45.20** | - | - | **43.36** | 81.06 | **56.41** |
| ImageNet-V2 | × | × | 43.12 | - | - | 47.39 | 81.47 | 59.92 |
| | × | ✓ | 44.93 | - | - | 48.03 | 80.88 | 60.25 |
| | ✓ | × | 32.17 | - | - | 54.43 | **91.31** | **68.18** |
| | ✓ | ✓ | **58.42** | - | - | **56.05** | 85.89 | 67.72 |
| ImageNet-R | × | × | 57.34 | - | - | 61.60 | 86.31 | 71.81 |
| | × | ✓ | 60.70 | - | - | 62.93 | 84.95 | 72.19 |
| | ✓ | × | 47.20 | - | - | 70.25 | **94.52** | **80.57** |
| | ✓ | ✓ | **71.54** | - | - | **71.13** | 92.28 | 80.25 |

can result in better performance on some datasets, it can also lead to severe performance drops in certain cases, which is intolerable. For example, using ImageNet as the ID dataset with a $50\%$ noise ratio, the performance *drops from* $73.09\%$ *to* $41.34\%$ when using the outputs of the noise detector as pseudo-labels. We suppose this discrepancy arises from cumulative errors when using the noise

Table 17: Ablation studies for the different injected noise in the method. For CIFAR-10/100, results are averaged across four OOD datasets under the noisy data stream: SVHN, LSUN, Texture, and Places. For other ID datasets, averaging includes four OOD datasets under the noisy data stream: iNaturalist, SUN, Texture, and Places.

| ID | Noise Type | Clean Data Stream | | | Noisy Data Stream | | |
|---|---|---|---|---|---|---|---|
| | | $Acc_S$ | $Acc_N$ | $Acc_H$ | $Acc_S$ | $Acc_N$ | $Acc_H$ |
| CIFAR-10 | Gaussian | 89.14 | - | - | 89.32 | 97.79 | 93.34 |
| | Uniform | 89.07 | - | - | 89.25 | 97.80 | 93.31 |
| | Salt-and-pepper | 89.08 | - | - | 89.23 | 97.91 | 93.35 |
| | Poisson | 89.07 | - | - | 89.28 | 97.90 | 93.37 |
| CIFAR-100 | Gaussian | 62.70 | - | - | 63.88 | 81.99 | 70.19 |
| | Uniform | 62.79 | - | - | 64.48 | 82.92 | 71.25 |
| | Salt-and-pepper | 63.43 | - | - | 64.24 | 80.70 | 69.25 |
| | Poisson | 62.80 | - | - | 63.98 | 80.94 | 69.38 |
| CUB-200-2011 | Gaussian | 49.53 | - | - | 52.10 | 90.77 | 66.14 |
| | Uniform | 49.53 | - | - | 52.09 | 90.95 | 66.19 |
| | Salt-and-pepper | 48.83 | - | - | 51.85 | 91.41 | 66.13 |
| | Poisson | 48.91 | - | - | 52.01 | 90.98 | 66.13 |
| STANFORD-CARS | Gaussian | 58.61 | - | - | 62.80 | 99.66 | 77.05 |
| | Uniform | 58.83 | - | - | 62.83 | 99.67 | 77.07 |
| | Salt-and-pepper | 57.76 | - | - | 62.70 | 99.65 | 76.97 |
| | Poisson | 58.44 | - | - | 62.79 | 99.67 | 77.04 |
| Food-101 | Gaussian | 86.23 | - | - | 86.44 | 98.85 | 92.23 |
| | Uniform | 86.26 | - | - | 86.46 | 98.86 | 92.24 |
| | Salt-and-pepper | 86.25 | - | - | 86.45 | 98.89 | 92.25 |
| | Poisson | 86.21 | - | - | 86.43 | 98.88 | 92.23 |
| Oxford-IIIT Pet | Gaussian | 84.95 | - | - | 85.84 | 98.06 | 91.54 |
| | Uniform | 84.68 | - | - | 85.81 | 98.15 | 91.57 |
| | Salt-and-pepper | 84.88 | - | - | 85.82 | 98.12 | 91.55 |
| | Poisson | 84.56 | - | - | 85.78 | 98.21 | 91.57 |
| ImageNet | Gaussian | 63.99 | - | - | 62.24 | 88.67 | 73.09 |
| | Uniform | 64.63 | - | - | 62.58 | 88.11 | 73.13 |
| | Salt-and-pepper | 64.34 | - | - | 62.42 | 88.21 | 73.05 |
| | Poisson | 64.20 | - | - | 62.30 | 88.41 | 73.03 |
| ImageNet-K | Gaussian | 36.43 | - | - | 39.78 | 83.15 | 53.77 |
| | Uniform | 37.28 | - | - | 40.19 | 82.29 | 53.95 |
| | Salt-and-pepper | 37.28 | - | - | 40.20 | 82.34 | 53.97 |
| | Poisson | 36.92 | - | - | 40.10 | 82.48 | 53.92 |
| ImageNet-A | Gaussian | 45.31 | - | - | 43.36 | 81.06 | 56.41 |
| | Uniform | 45.28 | - | - | 43.38 | 81.09 | 56.43 |
| | Salt-and-pepper | 44.24 | - | - | 42.75 | 82.47 | 56.24 |
| | Poisson | 44.39 | - | - | 42.90 | 82.25 | 56.31 |
| ImageNet-V2 | Gaussian | 58.39 | - | - | 56.05 | 85.89 | 67.72 |
| | Uniform | 58.57 | - | - | 56.36 | 85.29 | 67.75 |
| | Salt-and-pepper | 58.44 | - | - | 55.99 | 85.51 | 67.56 |
| | Poisson | 58.20 | - | - | 55.89 | 85.68 | 67.53 |
| ImageNet-R | Gaussian | 71.52 | - | - | 71.13 | 92.28 | 80.25 |
| | Uniform | 71.54 | - | - | 71.14 | 92.32 | 80.27 |
| | Salt-and-pepper | 71.08 | - | - | 70.98 | 92.52 | 80.25 |
| | Poisson | 71.19 | - | - | 71.07 | 92.35 | 80.23 |

detector's results as pseudo-labels. To better handle varying ID datasets and noise ratios, we use ZS-CLIP's result as pseudo-labels, which is more robust.

# G    FULL RESULTS OF FAILURE CASE

Besides evaluating different TTA methods using the rank distribution in Figure 2, we also evaluate them using the absolute accuracy in Figure 7. Most TTA methods perform worse than ZS-CLIP under the ZS-NTTA setting, and our method still performs best.

## G.1    THREE MODEL ADAPTATION PIPELINES

Table 27 and Table 28 presents the performance of the three model adaptation pipelines using different datasets as the ID. For comprehensive evaluation, we also include AUROC and FPR95 metrics for different datasets in Table 29 and Table 30. Higher AUROC and lower FPR95 scores indicate superior performance. Note that AUROC and FPR95 cannot be calculated at test-time and can only be determined after evaluating all samples. The above results show that for most datasets, the model performance degrades as the number of noisy samples used for model updates increases.

Table 18: Ablation studies for the ratio of Gaussian noise in the method. '-' indicates that no Gaussian noise is inserted, while '8' means that 1 Gaussian noise sample is inserted for every 8 test samples. For CIFAR-10/100, results are averaged across four OOD datasets under the noisy data stream: SVHN, LSUN, Texture, and Places. For other ID datasets, averaging includes four OOD datasets under the noisy data stream: iNaturalist, SUN, Texture, and Places.

| ID | Ratio | Clean Data Stream | | | Noisy Data Stream | | |
|---|---|---|---|---|---|---|---|
| | | $Acc_S$ | $Acc_N$ | $Acc_H$ | $Acc_S$ | $Acc_N$ | $Acc_H$ |
| CIFAR-10 | 2 | 89.49 | - | - | 89.67 | 90.70 | 90.02 |
| | 4 | 89.32 | - | - | 89.41 | 96.28 | 92.67 |
| | 8 | 89.14 | - | - | 89.32 | 97.79 | 93.34 |
| CIFAR-100 | 2 | 65.79 | - | - | 66.02 | 67.83 | 61.75 |
| | 4 | 65.68 | - | - | 64.90 | 74.64 | 65.18 |
| | 8 | 62.70 | - | - | 63.88 | 81.99 | 70.19 |
| CUB-200-2011 | 2 | 54.03 | - | - | 54.29 | 74.27 | 62.30 |
| | 4 | 52.94 | - | - | 53.72 | 85.17 | 65.68 |
| | 8 | 49.53 | - | - | 52.10 | 90.77 | 66.14 |
| STANFORD-CARS | 2 | 62.84 | - | - | 63.20 | 98.88 | 77.11 |
| | 4 | 62.19 | - | - | 63.08 | 99.56 | 77.23 |
| | 8 | 58.61 | - | - | 62.80 | 99.66 | 77.05 |
| Food-101 | 2 | 86.46 | - | - | 86.57 | 97.71 | 91.80 |
| | 4 | 86.34 | - | - | 86.55 | 98.60 | 92.17 |
| | 8 | 86.23 | - | - | 86.44 | 98.85 | 92.23 |
| Oxford-IIIT Pet | 2 | 85.71 | - | - | 86.00 | 94.01 | 89.80 |
| | 4 | 85.30 | - | - | 85.94 | 97.52 | 91.36 |
| | 8 | 84.95 | - | - | 85.84 | 98.06 | 91.54 |
| ImageNet | 2 | 65.80 | - | - | 63.91 | 81.72 | 71.58 |
| | 4 | 65.43 | - | - | 63.14 | 86.00 | 72.73 |
| | 8 | 63.99 | - | - | 62.24 | 88.67 | 73.09 |
| ImageNet-K | 2 | 43.06 | - | - | 42.41 | 72.62 | 53.34 |
| | 4 | 40.98 | - | - | 41.16 | 78.76 | 53.97 |
| | 8 | 36.43 | - | - | 39.78 | 83.15 | 53.77 |
| ImageNet-A | 2 | 46.24 | - | - | 47.07 | 36.45 | 40.29 |
| | 4 | 46.05 | - | - | 45.72 | 62.37 | 52.20 |
| | 8 | 45.31 | - | - | 43.36 | 81.06 | 56.41 |
| ImageNet-V2 | 2 | 59.29 | - | - | 58.85 | 65.84 | 61.56 |
| | 4 | 58.77 | - | - | 57.52 | 77.51 | 65.68 |
| | 8 | 58.39 | - | - | 56.05 | 85.89 | 67.72 |
| ImageNet-R | 2 | 73.28 | - | - | 72.41 | 85.17 | 77.86 |
| | 4 | 72.94 | - | - | 71.83 | 89.61 | 79.55 |
| | 8 | 71.52 | - | - | 71.13 | 92.28 | 80.25 |

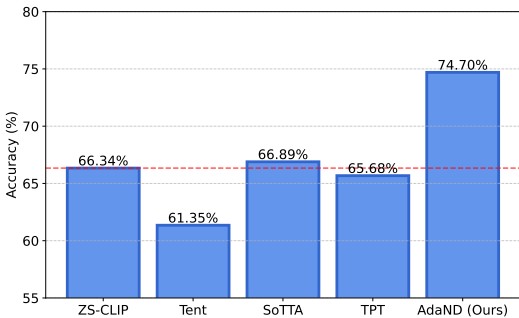

Figure 7: Average absolute accuracy for all methods across 44 ID-OOD dataset pairs. The red dashed line indicates the performance of ZS-CLIP.

## G.2 SCORE DIFFERENCE

The score distributions for TPT under the `Normal` pipeline are shown in Figures 8. Since TPT resets the model after updating each sample, the impact of unfiltered noisy samples is limited to the current step and does not accumulate. Despite this, the score of some noisy samples may increase, while the score of some ID samples may decrease, leading to a decline in performance.

Table 19: Ablation studies for different noise ratios in the data stream. For CIFAR-10/100, results are averaged across four OOD datasets: SVHN, LSUN, Texture, and Places. For other ID datasets, averaging includes four OOD datasets: iNaturalist, SUN, Texture, and Places. Note that 0% indicates the clean data stream. The **bold** indicates the best performance on each noise ratio.

| ID | Method | 0% $Acc_S$ | $Acc_N$ | $Acc_H$ | 25% $Acc_S$ | $Acc_N$ | $Acc_H$ | 50% $Acc_S$ | $Acc_N$ | $Acc_H$ | 75% $Acc_S$ | $Acc_N$ | $Acc_H$ |
|---|---|---|---|---|---|---|---|---|---|---|---|---|---|
| CIFAR-10 | ZS-CLIP | 77.96 | - | - | 81.83 | 91.82 | 86.39 | 82.64 | 91.07 | 86.47 | 83.29 | 90.27 | 86.42 |
| | Tent | 84.39 | - | - | 88.62 | 89.07 | 88.66 | 88.69 | 70.19 | 77.78 | 80.99 | 30.33 | 42.93 |
| | SoTTA | 83.82 | - | - | 88.29 | 90.58 | 89.26 | 89.73 | 84.47 | 86.88 | 90.14 | 64.30 | 73.56 |
| | TPT | 76.37 | - | - | 80.02 | 92.22 | 85.56 | 80.90 | 91.52 | 85.72 | 81.53 | 90.83 | 85.72 |
| | AdaND (Ours) | **89.16** | - | - | **89.29** | **95.85** | **92.43** | **89.32** | **97.79** | **93.34** | **89.10** | **95.75** | **92.21** |
| CIFAR-100 | ZS-CLIP | 44.69 | - | - | 46.35 | 85.17 | 59.77 | 47.48 | 84.66 | 60.61 | 48.82 | 83.65 | 61.43 |
| | Tent | 53.54 | - | - | 56.77 | 82.97 | 67.19 | 58.07 | 66.79 | 61.36 | 53.17 | 41.63 | 45.14 |
| | SoTTA | 52.62 | - | - | 56.25 | 85.24 | 67.57 | 59.16 | 81.36 | 68.34 | 62.07 | 72.69 | 66.86 |
| | TPT | 41.90 | - | - | 43.72 | **86.30** | 57.84 | 44.84 | **85.80** | 58.73 | 46.01 | **85.10** | 59.57 |
| | AdaND (Ours) | **62.52** | - | - | **63.24** | 75.14 | 65.29 | **63.88** | 81.99 | 70.19 | **64.28** | 82.21 | 70.83 |
| CUB-200-2011 | ZS-CLIP | 33.08 | - | - | 35.65 | 87.96 | 50.72 | 37.95 | 85.71 | 52.59 | 40.86 | 82.05 | 54.54 |
| | Tent | 36.69 | - | - | 39.14 | 82.75 | 53.08 | 37.75 | 54.78 | 44.38 | 31.45 | 21.34 | 25.21 |
| | SoTTA | 36.16 | - | - | 39.07 | 87.43 | 53.99 | 41.81 | 83.82 | 55.77 | 45.13 | 76.30 | 56.69 |
| | TPT | 32.07 | - | - | 34.93 | **89.49** | 50.24 | 37.30 | 87.59 | 52.31 | 39.84 | **84.66** | 54.18 |
| | AdaND (Ours) | **49.47** | - | - | **51.00** | 86.08 | 63.98 | **52.10** | 90.77 | 66.14 | **53.39** | 83.99 | 65.17 |
| STANFORD-CARS | ZS-CLIP | 39.02 | - | - | 47.44 | 98.84 | 64.10 | 52.65 | 98.13 | 68.53 | 54.82 | 97.44 | 70.16 |
| | Tent | 40.95 | - | - | 49.33 | 97.88 | 65.60 | 51.83 | 85.22 | 64.09 | 33.88 | 29.26 | 30.72 |
| | SoTTA | 40.60 | - | - | 49.44 | 98.55 | 65.84 | 54.03 | 96.47 | 69.27 | 54.66 | 87.40 | 67.25 |
| | TPT | 38.38 | - | - | 46.19 | 99.02 | 62.98 | 51.70 | 98.36 | 67.77 | 54.06 | 97.79 | 69.63 |
| | AdaND (Ours) | **58.53** | - | - | **62.41** | **99.03** | **76.57** | **62.80** | **99.66** | **77.05** | **63.10** | **99.75** | **77.30** |
| Food-101 | ZS-CLIP | 72.93 | - | - | 79.34 | 95.59 | 86.71 | 80.62 | 94.65 | 87.07 | 81.50 | 93.96 | 87.28 |
| | Tent | 76.20 | - | - | 81.47 | 85.00 | 82.68 | 80.87 | 69.34 | 71.90 | 63.38 | 30.37 | 39.10 |
| | SoTTA | 75.02 | - | - | 81.14 | 93.70 | 86.96 | 82.20 | 89.79 | 85.80 | 82.33 | 79.18 | 80.61 |
| | TPT | 71.92 | - | - | 78.49 | 95.79 | 86.28 | 79.77 | 95.03 | 86.73 | 80.64 | 94.33 | 86.95 |
| | AdaND (Ours) | **86.21** | - | - | **86.36** | **98.31** | **91.95** | **86.44** | **98.85** | **92.23** | **86.51** | **98.53** | **92.12** |
| Oxford-IIIT Pet | ZS-CLIP | 70.17 | - | - | 77.99 | 89.34 | 83.27 | 79.53 | 87.73 | 83.41 | 80.96 | 85.69 | 83.24 |
| | Tent | 73.36 | - | - | 79.90 | 86.64 | 83.10 | 80.84 | 70.91 | 75.41 | 74.81 | 32.87 | 45.64 |
| | SoTTA | 72.58 | - | - | 79.61 | 87.38 | 83.30 | 81.36 | 84.03 | 82.66 | 82.84 | 78.21 | 80.44 |
| | TPT | 69.44 | - | - | 76.98 | 90.96 | 83.38 | 78.56 | 89.78 | 83.78 | 79.95 | 87.75 | 83.66 |
| | AdaND (Ours) | **84.91** | - | - | **85.39** | **96.94** | **90.80** | **85.84** | **98.06** | **91.54** | **85.89** | **97.59** | **91.36** |
| ImageNet | ZS-CLIP | 47.68 | - | - | 51.00 | 84.72 | 63.66 | 53.38 | 82.38 | 64.77 | 55.64 | 79.64 | 65.50 |
| | Tent | 49.86 | - | - | 53.18 | 78.58 | 63.43 | 53.67 | 64.05 | 57.66 | 49.74 | 45.93 | 45.89 |
| | SoTTA | 49.82 | - | - | 52.36 | 74.92 | 61.63 | 53.39 | 67.16 | 59.47 | 52.53 | 57.03 | 54.68 |
| | TPT | 46.12 | - | - | 49.48 | 86.38 | 62.91 | 51.85 | 84.48 | 64.25 | 54.04 | 82.24 | 65.21 |
| | AdaND (Ours) | **63.96** | - | - | **62.53** | **86.82** | **72.62** | **62.24** | **88.67** | **73.09** | **61.53** | **85.52** | **71.52** |
| ImageNet-K | ZS-CLIP | 30.48 | - | - | 31.92 | 81.26 | 45.83 | 33.41 | 79.33 | 47.01 | 34.76 | 77.17 | 47.93 |
| | Tent | 33.66 | - | - | 35.40 | 71.84 | 47.37 | 35.07 | 58.91 | 43.11 | 31.60 | 41.60 | 34.34 |
| | SoTTA | 34.20 | - | - | 35.70 | 72.08 | 47.74 | 36.23 | 65.32 | 46.60 | 35.44 | 56.08 | 43.43 |
| | TPT | 28.78 | - | - | 30.15 | 83.59 | 44.30 | 31.50 | 81.95 | 45.50 | 32.73 | **80.22** | 46.49 |
| | AdaND (Ours) | **36.54** | - | - | **38.40** | **85.81** | **52.98** | **39.78** | 83.15 | **53.77** | **40.02** | 78.07 | **52.91** |
| ImageNet-A | ZS-CLIP | 31.47 | - | - | 32.94 | **79.21** | 46.52 | 34.22 | 77.81 | 47.53 | 35.67 | 75.87 | 48.52 |
| | Tent | 32.09 | - | - | 33.55 | 78.14 | 46.94 | 34.70 | 75.81 | 47.60 | 35.38 | 70.22 | 47.05 |
| | SoTTA | 33.43 | - | - | 34.72 | 78.06 | 48.06 | 36.25 | 75.79 | 49.04 | 38.23 | 71.33 | 49.77 |
| | TPT | 30.45 | - | - | 32.05 | 80.92 | 45.91 | 33.37 | 79.58 | 47.02 | 34.87 | **78.02** | 48.20 |
| | AdaND (Ours) | **45.20** | - | - | **42.84** | 70.45 | **52.86** | **43.36** | **81.06** | **56.41** | **44.06** | 73.46 | **55.00** |
| ImageNet-V2 | ZS-CLIP | 43.12 | - | - | 45.60 | 83.36 | 58.93 | 47.39 | 81.47 | 59.92 | 49.25 | 78.94 | 60.64 |
| | Tent | 43.46 | - | - | 46.11 | 81.95 | 58.99 | 48.03 | 77.33 | 59.25 | 48.25 | 65.09 | 55.11 |
| | SoTTA | 43.87 | - | - | 46.38 | 80.75 | 58.91 | 47.89 | 77.03 | 59.06 | 49.01 | 70.30 | 57.75 |
| | TPT | 41.53 | - | - | 44.04 | **85.38** | 58.09 | 46.00 | 83.76 | 59.37 | 47.83 | 81.68 | 60.33 |
| | AdaND (Ours) | **58.42** | - | - | **56.37** | 76.77 | **64.70** | **56.05** | **85.89** | **67.72** | **56.34** | **83.12** | **67.04** |
| ImageNet-R | ZS-CLIP | 57.34 | - | - | 59.92 | 87.67 | 71.11 | 61.60 | 86.31 | 71.81 | 63.14 | 84.79 | 72.30 |
| | Tent | 60.14 | - | - | 62.96 | 86.16 | 72.68 | 64.64 | 82.36 | 72.30 | 60.97 | 65.95 | 63.07 |
| | SoTTA | 61.62 | - | - | 64.81 | 85.56 | 73.67 | 66.50 | 81.05 | 72.99 | 68.14 | 70.13 | 69.07 |
| | TPT | 56.20 | - | - | 59.01 | 88.41 | 70.71 | 60.61 | 87.09 | 71.40 | 62.09 | 85.84 | 71.98 |
| | AdaND (Ours) | **71.54** | - | - | **71.23** | **91.61** | **80.05** | **71.13** | **92.28** | **80.25** | **70.95** | **88.95** | **78.79** |

## G.3 GRADIENT ANALYSIS

Figure 9 shows the impact of clean and noisy samples on the gradients in Tent. To present a clear view, Figure 9 only displays the portion of the gradient magnitudes less than 0.0010.

Table 20: Zero-shot noisy TTA results for CIFAR-10/100 as the ID datasets with different random seeds. The results are the mean ± standard deviation with five random seeds. The **bold** indicates the best performance on each dataset.

| ID | Method | SVHN | | | LSUN | | | Texture | | | Places | | | Avg | | |
|---|---|---|---|---|---|---|---|---|---|---|---|---|---|---|---|---|
| | | $Acc_S$ | $Acc_N$ | $Acc_H$ | $Acc_S$ | $Acc_N$ | $Acc_H$ | $Acc_S$ | $Acc_N$ | $Acc_H$ | $Acc_S$ | $Acc_N$ | $Acc_H$ | $Acc_S$ | $Acc_N$ | $Acc_H$ |
| CIFAR-10 | ZS-CLIP | 83.53 ± 0.02 | 98.35 ± 0.04 | 90.33 ± 0.02 | 83.10 ± 0.03 | 97.83 ± 0.01 | 89.87 ± 0.02 | 82.20 ± 0.04 | 91.83 ± 0.01 | 86.75 ± 0.02 | 81.73 ± 0.02 | 76.46 ± 0.12 | 79.00 ± 0.06 | 82.64 ± 0.03 | 91.11 ± 0.05 | 86.49 ± 0.03 |
| | Tent | 87.31 ± 0.30 | 54.02 ± 3.13 | 66.70 ± 2.50 | 88.54 ± 0.33 | 70.43 ± 2.71 | 78.43 ± 1.80 | 89.66 ± 0.07 | 88.67 ± 0.13 | 89.16 ± 0.05 | 88.65 ± 0.10 | 64.85 ± 0.58 | 74.90 ± 0.40 | 88.54 ± 0.20 | 69.49 ± 1.64 | 77.30 ± 1.19 |
| | SoTTA | 89.96 ± 0.14 | 80.08 ± 2.06 | 84.72 ± 1.19 | 90.14 ± 0.11 | 91.26 ± 0.72 | 90.69 ± 0.38 | 89.51 ± 0.12 | 90.94 ± 0.11 | 90.22 ± 0.07 | 89.22 ± 0.17 | 74.07 ± 0.18 | 80.94 ± 0.13 | 89.71 ± 0.14 | 84.09 ± 0.77 | 86.64 ± 0.44 |
| | TPT | 81.79 ± 0.09 | 98.89 ± 0.04 | 89.53 ± 0.04 | 81.38 ± 0.09 | 97.96 ± 0.02 | 88.90 ± 0.05 | 80.46 ± 0.05 | 92.10 ± 0.04 | 85.89 ± 0.03 | 79.90 ± 0.03 | 77.39 ± 0.12 | 78.62 ± 0.06 | 80.88 ± 0.06 | 91.58 ± 0.06 | 85.74 ± 0.05 |
| | ZS-NTTA (Ours) | 89.36 ± 0.16 | 99.87 ± 0.04 | 94.32 ± 0.10 | 88.30 ± 0.54 | 99.66 ± 0.03 | 93.64 ± 0.30 | 89.55 ± 0.19 | 98.68 ± 0.23 | 93.89 ± 0.15 | 89.63 ± 0.08 | 93.43 ± 0.56 | 91.49 ± 0.25 | 89.21 ± 0.24 | 97.91 ± 0.21 | 93.33 ± 0.20 |
| CIFAR-100 | ZS-CLIP | 48.50 ± 0.07 | 97.59 ± 0.04 | 64.80 ± 0.07 | 49.17 ± 0.08 | 95.05 ± 0.06 | 64.81 ± 0.06 | 46.78 ± 0.05 | 81.63 ± 0.03 | 59.48 ± 0.04 | 45.37 ± 0.06 | 64.44 ± 0.13 | 53.25 ± 0.06 | 47.46 ± 0.07 | 84.68 ± 0.06 | 60.59 ± 0.06 |
| | Tent | 54.72 ± 0.42 | 41.45 ± 0.89 | 47.17 ± 0.71 | 59.80 ± 0.31 | 83.27 ± 0.24 | 69.61 ± 0.27 | 59.07 ± 0.22 | 79.42 ± 0.22 | 67.74 ± 0.09 | 57.36 ± 0.10 | 62.08 ± 0.26 | 59.63 ± 0.12 | 57.74 ± 0.26 | 66.55 ± 0.40 | 61.04 ± 0.30 |
| | SoTTA | 60.30 ± 0.16 | 89.43 ± 0.44 | 72.03 ± 0.18 | 59.91 ± 0.21 | 89.24 ± 0.30 | 71.69 ± 0.16 | 58.63 ± 0.17 | 81.70 ± 0.18 | 68.27 ± 0.09 | 56.92 ± 0.13 | 65.76 ± 0.07 | 61.02 ± 0.06 | 58.94 ± 0.17 | 81.53 ± 0.25 | 68.25 ± 0.12 |
| | TPT | 45.97 ± 0.07 | 97.88 ± 0.03 | 62.56 ± 0.06 | 46.69 ± 0.15 | 95.41 ± 0.06 | 62.70 ± 0.13 | 43.92 ± 0.13 | 83.30 ± 0.16 | 57.51 ± 0.11 | 42.47 ± 0.12 | 66.71 ± 0.13 | 51.90 ± 0.12 | 44.76 ± 0.12 | 85.83 ± 0.09 | 58.67 ± 0.10 |
| | ZS-NTTA (Ours) | 63.71 ± 0.74 | 99.74 ± 0.06 | 77.75 ± 0.55 | 61.59 ± 1.01 | 99.12 ± 0.18 | 75.97 ± 0.81 | 63.82 ± 1.21 | 85.43 ± 1.52 | 73.05 ± 1.11 | 62.21 ± 1.16 | 49.12 ± 2.89 | 54.82 ± 1.49 | 62.83 ± 1.03 | 83.35 ± 1.17 | 70.40 ± 0.99 |

Table 21: Zero-shot noisy TTA results for CIFAR-10/100 as the ID datasets with different random seeds in terms of AUROC and FPR95. The results are the mean ± standard deviation with five random seeds. The **bold** indicates the best performance on each dataset.

| ID | Method | SVHN | | LSUN | | Texture | | Places | | Avg | |
|---|---|---|---|---|---|---|---|---|---|---|---|
| | | AUROC↑ | FPR95↓ | AUROC↑ | FPR95↓ | AUROC↑ | FPR95↓ | AUROC↑ | FPR95↓ | AUROC↑ | FPR95↓ |
| CIFAR-10 | ZS-CLIP | 98.45 ± 0.00 | 6.75 ± 0.00 | 97.75 ± 0.00 | 10.64 ± 0.00 | 94.75 ± 0.00 | 28.08 ± 0.00 | 87.47 ± 0.00 | 50.18 ± 0.00 | 94.60 ± 0.00 | 23.91 ± 0.00 |
| | Tent | 75.11 ± 2.21 | 48.89 ± 3.14 | 87.08 ± 1.79 | 34.46 ± 2.54 | 96.87 ± 0.02 | 16.01 ± 0.13 | 87.64 ± 0.10 | 46.19 ± 0.38 | 86.68 ± 1.03 | 36.39 ± 1.55 |
| | SoTTA | 95.27 ± 0.66 | 22.67 ± 2.38 | 97.72 ± 0.16 | 11.47 ± 0.73 | 97.32 ± 0.03 | 13.05 ± 0.18 | 91.57 ± 0.09 | 33.88 ± 0.15 | 95.47 ± 0.23 | 20.27 ± 0.86 |
| | TPT | 98.48 ± 0.00 | 6.80 ± 0.02 | 97.62 ± 0.01 | 10.73 ± 0.04 | 94.19 ± 0.01 | 28.21 ± 0.05 | 85.33 ± 0.03 | 50.19 ± 0.02 | 93.91 ± 0.01 | 23.98 ± 0.03 |
| | ZS-NTTA (Ours) | 99.95 ± 0.01 | 0.13 ± 0.04 | 99.82 ± 0.02 | 0.41 ± 0.07 | 99.70 ± 0.01 | 0.58 ± 0.08 | 98.80 ± 0.02 | 2.38 ± 0.08 | 99.57 ± 0.02 | 0.87 ± 0.07 |
| CIFAR-100 | ZS-CLIP | 85.11 ± 0.00 | 86.42 ± 0.00 | 85.88 ± 0.00 | 72.58 ± 0.00 | 71.09 ± 0.00 | 95.35 ± 0.00 | 58.47 ± 0.00 | 98.97 ± 0.00 | 75.14 ± 0.00 | 88.33 ± 0.00 |
| | Tent | 45.44 ± 0.82 | 81.05 ± 0.59 | 84.67 ± 0.33 | 62.67 ± 0.76 | 80.38 ± 0.08 | 73.38 ± 0.47 | 68.94 ± 0.06 | 91.61 ± 0.18 | 69.86 ± 0.32 | 77.18 ± 0.50 |
| | SoTTA | 88.78 ± 0.24 | 51.05 ± 0.54 | 87.99 ± 0.13 | 55.39 ± 0.79 | 81.47 ± 0.08 | 70.58 ± 0.32 | 70.59 ± 0.11 | 89.96 ± 0.37 | 82.20 ± 0.14 | 66.74 ± 0.51 |
| | TPT | 84.81 ± 0.01 | 86.46 ± 0.06 | 85.39 ± 0.01 | 72.59 ± 0.01 | 69.65 ± 0.02 | 95.35 ± 0.00 | 55.61 ± 0.04 | 98.97 ± 0.00 | 73.86 ± 0.02 | 88.34 ± 0.02 |
| | ZS-NTTA (Ours) | 99.06 ± 0.13 | 3.76 ± 0.95 | 98.23 ± 0.22 | 5.95 ± 1.32 | 93.11 ± 0.63 | 21.45 ± 2.37 | 77.72 ± 0.75 | 67.59 ± 1.69 | 92.03 ± 0.43 | 24.69 ± 1.58 |

Table 22: Ablation studies on the queue capacity $L$ for noise detector updates in the method with CIFAR-10/100 as the ID datasets.

| ID | $L$ | SVHN | | | LSUN | | | Texture | | | Places | | | Avg | | |
|---|---|---|---|---|---|---|---|---|---|---|---|---|---|---|---|---|
| | | $Acc_S$ | $Acc_N$ | $Acc_H$ | $Acc_S$ | $Acc_N$ | $Acc_H$ | $Acc_S$ | $Acc_N$ | $Acc_H$ | $Acc_S$ | $Acc_N$ | $Acc_H$ | $Acc_S$ | $Acc_N$ | $Acc_H$ |
| CIFAR-10 | 32 | 88.92 | 99.95 | 94.11 | 87.98 | 99.77 | 93.50 | 89.09 | 97.86 | 93.27 | 88.95 | 87.98 | 88.46 | 88.73 | 96.39 | 92.33 |
| | 64 | 89.37 | 99.96 | 94.37 | 88.21 | 99.68 | 93.59 | 89.57 | 98.39 | 93.77 | 89.49 | 90.16 | 89.82 | 89.16 | 97.05 | 92.89 |
| | 128 | 89.46 | 99.90 | 94.39 | 88.56 | 99.66 | 93.78 | 89.60 | 98.54 | 93.86 | 89.65 | 93.04 | 91.31 | 89.32 | 97.79 | 93.34 |
| | 256 | 89.03 | 99.75 | 94.09 | 88.01 | 99.08 | 93.22 | 89.21 | 97.69 | 93.26 | 89.16 | 93.68 | 91.36 | 88.85 | 97.55 | 92.98 |
| | 512 | 88.36 | 99.49 | 93.60 | 87.14 | 98.34 | 92.40 | 88.29 | 95.91 | 91.94 | 88.21 | 90.78 | 89.48 | 88.00 | 96.13 | 91.86 |
| CIFAR-100 | 32 | 61.31 | 99.91 | 75.99 | 59.80 | 99.68 | 74.75 | 60.52 | 86.47 | 71.20 | 57.14 | 57.55 | 57.34 | 59.69 | 85.90 | 69.82 |
| | 64 | 63.94 | 99.80 | 77.94 | 61.56 | 99.48 | 76.06 | 63.63 | 85.98 | 73.14 | 60.06 | 50.86 | 55.08 | 62.30 | 84.03 | 70.55 |
| | 128 | 64.44 | 99.78 | 78.31 | 62.42 | 99.15 | 76.61 | 65.17 | 84.84 | 73.72 | 63.50 | 44.21 | 52.13 | 63.88 | 81.99 | 70.19 |
| | 256 | 63.64 | 99.38 | 77.59 | 61.10 | 97.82 | 75.22 | 64.17 | 83.45 | 72.55 | 63.65 | 41.41 | 50.18 | 63.14 | 80.51 | 68.89 |
| | 512 | 61.01 | 98.88 | 75.46 | 56.26 | 95.18 | 70.72 | 61.38 | 77.75 | 68.60 | 60.44 | 43.56 | 50.63 | 59.77 | 78.84 | 66.35 |

Table 23: Ablation studies on the queue capacity $N_q$ for queue length to store the output score in the method with CIFAR-10/100 as the ID datasets.

| ID | $N_q$ | SVHN | | | LSUN | | | Texture | | | Places | | | Avg | | |
|---|---|---|---|---|---|---|---|---|---|---|---|---|---|---|---|---|
| | | $Acc_S$ | $Acc_N$ | $Acc_H$ | $Acc_S$ | $Acc_N$ | $Acc_H$ | $Acc_S$ | $Acc_N$ | $Acc_H$ | $Acc_S$ | $Acc_N$ | $Acc_H$ | $Acc_S$ | $Acc_N$ | $Acc_H$ |
| CIFAR-10 | 64 | 88.16 | 99.90 | 93.66 | 87.35 | 99.60 | 93.07 | 88.70 | 98.81 | 93.48 | 89.10 | 94.59 | 91.76 | 88.33 | 98.22 | 92.99 |
| | 128 | 89.41 | 99.90 | 94.36 | 88.61 | 99.63 | 93.80 | 89.63 | 98.52 | 93.86 | 89.65 | 93.09 | 91.34 | 89.32 | 97.78 | 93.34 |
| | 256 | 89.46 | 99.90 | 94.39 | 88.53 | 99.65 | 93.76 | 89.62 | 98.52 | 93.86 | 89.65 | 93.08 | 91.33 | 89.31 | 97.79 | 93.33 |
| | 512 | 89.46 | 99.90 | 94.39 | 88.56 | 99.66 | 93.78 | 89.60 | 98.54 | 93.86 | 89.65 | 93.04 | 91.31 | 89.32 | 97.79 | 93.34 |
| | 1024 | 89.43 | 99.91 | 94.38 | 88.51 | 99.65 | 93.75 | 89.62 | 98.57 | 93.88 | 89.64 | 93.01 | 91.29 | 89.30 | 97.78 | 93.33 |
| CIFAR-100 | 64 | 64.19 | 99.65 | 78.08 | 62.64 | 98.73 | 76.65 | 64.61 | 84.64 | 73.28 | 62.10 | 46.96 | 53.48 | 63.38 | 82.49 | 70.37 |
| | 128 | 64.92 | 99.72 | 78.64 | 62.86 | 98.90 | 76.87 | 65.28 | 84.35 | 73.60 | 63.51 | 46.15 | 53.46 | 64.14 | 82.28 | 70.64 |
| | 256 | 64.68 | 99.76 | 78.48 | 62.65 | 98.97 | 76.73 | 65.33 | 84.55 | 73.71 | 63.58 | 44.45 | 52.32 | 64.06 | 81.93 | 70.31 |
| | 512 | 64.44 | 99.78 | 78.31 | 62.42 | 99.15 | 76.61 | 65.17 | 84.84 | 73.72 | 63.50 | 44.21 | 52.13 | 63.88 | 81.99 | 70.19 |
| | 1024 | 64.22 | 99.77 | 78.14 | 62.04 | 99.21 | 76.34 | 65.10 | 85.13 | 73.78 | 63.33 | 46.30 | 53.49 | 63.67 | 82.60 | 70.44 |

Table 24: Ablation studies for the different initialization steps in the method with CIFAR-10/100 as the ID datasets.

| ID | Step | SVHN | | | LSUN | | | Texture | | | Places | | | Avg | | |
|---|---|---|---|---|---|---|---|---|---|---|---|---|---|---|---|---|
| | | $Acc_S$ | $Acc_N$ | $Acc_H$ | $Acc_S$ | $Acc_N$ | $Acc_H$ | $Acc_S$ | $Acc_N$ | $Acc_H$ | $Acc_S$ | $Acc_N$ | $Acc_H$ | $Acc_S$ | $Acc_N$ | $Acc_H$ |
| CIFAR-10 | 0 | 88.49 | 98.64 | 93.29 | 86.79 | 97.99 | 92.05 | 89.27 | 97.48 | 93.19 | 89.51 | 92.45 | 90.96 | 88.52 | 96.64 | 92.37 |
| | 10 | 89.46 | 99.90 | 94.39 | 88.56 | 99.66 | 93.78 | 89.60 | 98.54 | 93.86 | 89.65 | 93.04 | 91.31 | 89.32 | 97.79 | 93.34 |
| | 20 | 89.13 | 99.76 | 94.15 | 88.35 | 99.66 | 93.66 | 89.20 | 98.28 | 93.52 | 89.20 | 92.18 | 90.67 | 88.97 | 97.47 | 93.00 |
| | 30 | 88.81 | 99.68 | 93.93 | 88.03 | 99.53 | 93.43 | 88.80 | 97.91 | 93.13 | 88.75 | 91.23 | 89.97 | 88.60 | 97.09 | 92.62 |
| | 40 | 88.48 | 99.56 | 93.69 | 87.70 | 99.42 | 93.19 | 88.37 | 97.44 | 92.68 | 88.34 | 89.85 | 89.09 | 88.22 | 96.57 | 92.16 |
| | 50 | 88.21 | 99.45 | 93.49 | 87.49 | 99.20 | 92.98 | 88.01 | 96.97 | 92.27 | 87.98 | 88.33 | 88.15 | 87.92 | 95.99 | 91.72 |
| CIFAR-100 | 0 | 63.39 | 98.72 | 77.21 | 60.72 | 98.15 | 75.03 | 64.42 | 83.91 | 72.88 | 62.12 | 43.74 | 51.33 | 62.66 | 81.13 | 69.11 |
| | 10 | 64.44 | 99.78 | 78.31 | 62.42 | 99.15 | 76.61 | 65.17 | 84.84 | 73.72 | 63.50 | 44.21 | 52.13 | 63.88 | 81.99 | 70.19 |
| | 20 | 63.75 | 99.61 | 77.74 | 62.76 | 99.00 | 76.82 | 64.14 | 85.99 | 73.47 | 62.68 | 45.20 | 52.52 | 63.33 | 82.45 | 70.14 |
| | 30 | 62.51 | 99.52 | 76.79 | 61.89 | 98.97 | 76.16 | 62.80 | 86.48 | 72.76 | 61.25 | 47.26 | 53.35 | 62.11 | 83.06 | 69.77 |
| | 40 | 61.30 | 99.41 | 75.84 | 60.77 | 98.72 | 75.23 | 61.42 | 86.51 | 71.84 | 59.79 | 48.74 | 53.70 | 60.82 | 83.34 | 69.15 |
| | 50 | 60.37 | 99.29 | 75.09 | 59.90 | 98.46 | 74.49 | 60.25 | 86.08 | 70.89 | 58.47 | 51.19 | 54.59 | 59.75 | 83.75 | 68.76 |

Table 25: Ablation studies on VLM's architecture with CIFAR-10 as the ID datasets.

| Backbone | Method | SVHN | | | LSUN | | | Texture | | | Places | | | Avg | | |
|---|---|---|---|---|---|---|---|---|---|---|---|---|---|---|---|---|
| | | $Acc_S$ | $Acc_N$ | $Acc_H$ | $Acc_S$ | $Acc_N$ | $Acc_H$ | $Acc_S$ | $Acc_N$ | $Acc_H$ | $Acc_S$ | $Acc_N$ | $Acc_H$ | $Acc_S$ | $Acc_N$ | $Acc_H$ |
| RN50 | ZS-CLIP | 51.73 | 99.84 | 68.15 | 49.90 | 97.47 | 66.01 | 50.09 | 93.91 | 65.33 | 47.75 | 71.13 | 57.14 | 49.87 | 90.59 | 64.16 |
| | Tent | 16.81 | 47.90 | 24.89 | 20.31 | 63.46 | 30.77 | 29.19 | 54.21 | 37.95 | 24.82 | 33.72 | 28.59 | 22.78 | 49.82 | 30.55 |
| | SoTTA | 19.12 | 64.20 | 29.46 | 22.07 | 84.22 | 34.97 | 29.62 | 83.94 | 43.79 | 26.22 | 61.15 | 36.70 | 24.26 | 73.38 | 36.23 |
| | TPT | 51.02 | **99.87** | 67.54 | 48.90 | 97.52 | 65.14 | 49.01 | 94.13 | 64.46 | 46.20 | 72.34 | 56.39 | 48.78 | 90.97 | 63.38 |
| | AdaND (Ours) | **67.05** | 99.69 | **80.18** | **63.29** | 98.39 | **77.03** | **68.95** | 96.85 | **80.55** | **68.66** | 84.88 | **75.91** | **66.99** | 94.95 | **78.42** |
| ViT-L/14 | ZS-CLIP | 90.71 | 96.46 | 93.50 | 90.85 | 95.70 | 93.21 | 90.55 | 91.77 | 91.16 | 89.91 | 75.72 | 82.21 | 90.50 | 89.91 | 90.02 |
| | Tent | 90.54 | 36.08 | 51.60 | 93.52 | 80.30 | 86.41 | 93.94 | 90.35 | 92.11 | 93.42 | 68.95 | 79.34 | 92.86 | 68.92 | 77.37 |
| | SoTTA | 93.90 | 70.96 | 80.83 | 94.14 | 91.80 | 92.96 | 94.02 | 90.28 | 92.11 | 93.74 | 71.44 | 81.08 | 93.95 | 81.12 | 86.74 |
| | TPT | 89.98 | 96.55 | 93.15 | 90.14 | 95.98 | 92.97 | 89.78 | 91.99 | 90.87 | 89.23 | 76.13 | 82.16 | 89.78 | 90.16 | 89.79 |
| | AdaND (Ours) | **94.77** | **99.65** | **97.15** | **94.50** | **99.67** | **97.02** | **94.87** | **98.65** | **96.72** | **94.86** | **89.95** | **92.34** | **94.75** | **96.98** | **95.81** |

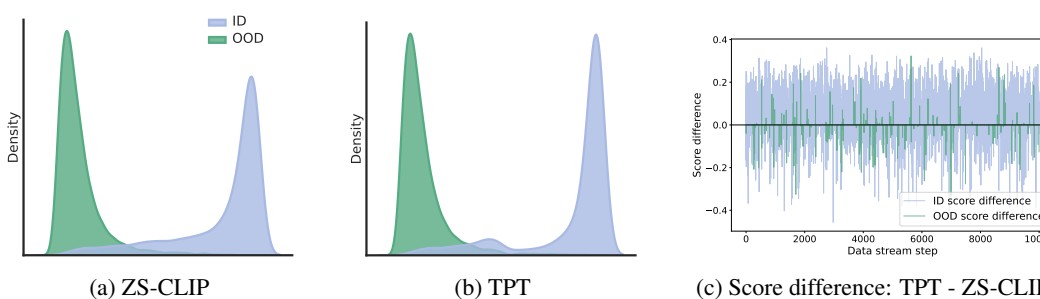

(a) ZS-CLIP     (b) TPT     (c) Score difference: TPT - ZS-CLIP

Figure 8: Failure case analysis of TPT (Shu et al., 2022) in ZS-NTTA. (a) and (b) show the score distributions of ZS-CLIP and TPT, respectively. ID dataset: CIFAR-10; OOD dataset: SVHN.

Table 26: Ablation studies for pseudo-labels generated by the noise detector under various noise ratios. Red indicates a performance drop when using the outputs of the noise detector as pseudo-labels in terms of Acc$_H$. For CIFAR-10/100, results are averaged across four OOD datasets: SVHN, LSUN, Texture, and Places. For other ID datasets, averaging includes four OOD datasets: iNaturalist, SUN, Texture, and Places. Note that 0% indicates the clean data stream.

| ID | Pseudo-label | 0% | | | 25% | | | 50% | | | 75% | | |
|---|---|---|---|---|---|---|---|---|---|---|---|---|---|
| | | Acc$_S$ | Acc$_N$ | Acc$_H$ | Acc$_S$ | Acc$_N$ | Acc$_H$ | Acc$_S$ | Acc$_N$ | Acc$_H$ | Acc$_S$ | Acc$_N$ | Acc$_H$ |
| CIFAR-10 | Noise Detector | 89.16 | - | - | 89.42 | 75.47 | 74.31 | 89.53 | 98.94 | 94.00 | 89.34 | 99.24 | 94.03 |
| | Frozen Model | 89.16 | - | - | 89.29 | 95.85 | 92.43 | 89.32 | 97.79 | 93.34 | 89.10 | 95.75 | 92.21 |
| CIFAR-100 | Noise Detector | 64.82 | - | - | 65.20 | 72.57 | 65.80 | 66.33 | 96.08 | 78.44 | 66.53 | 75.22 | 61.74 |
| | Frozen Model | 62.52 | - | - | 63.24 | 75.14 | 65.29 | 63.88 | 81.99 | 70.19 | 64.28 | 82.21 | 70.83 |
| CUB-200-2011 | Noise Detector | 52.47 | - | - | 53.72 | 88.55 | 66.84 | 53.94 | 95.85 | 69.03 | 54.67 | 97.93 | 70.17 |
| | Frozen Model | 49.47 | - | - | 51.00 | 86.08 | 63.98 | 52.10 | 90.77 | 66.14 | 53.39 | 83.99 | 65.17 |
| STANFORD-CARS | Noise Detector | 62.07 | - | - | 62.82 | 99.27 | 76.94 | 63.11 | 99.66 | 77.28 | 63.37 | 99.75 | 77.51 |
| | Frozen Model | 58.53 | - | - | 62.41 | 99.03 | 76.57 | 62.80 | 99.66 | 77.05 | 63.10 | 99.75 | 77.30 |
| Food-101 | Noise Detector | 86.23 | - | - | 86.38 | 98.00 | 91.82 | 86.45 | 99.17 | 92.37 | 86.49 | 99.58 | 92.57 |
| | Frozen Model | 86.21 | - | - | 86.36 | 98.31 | 91.95 | 86.44 | 98.85 | 92.23 | 86.51 | 98.53 | 92.12 |
| Oxford-IIIT Pet | Noise Detector | 84.95 | - | - | 85.42 | 96.84 | 90.77 | 85.85 | 98.18 | 91.60 | 85.91 | 98.81 | 91.91 |
| | Frozen Model | 84.91 | - | - | 85.39 | 96.94 | 90.80 | 85.84 | 98.06 | 91.54 | 85.89 | 97.59 | 91.36 |
| ImageNet | Noise Detector | 66.23 | - | - | 66.15 | 26.11 | 23.30 | 65.57 | 48.47 | 41.34 | 65.21 | 47.62 | 40.08 |
| | Frozen Model | 63.96 | - | - | 62.53 | 86.82 | 72.62 | 62.24 | 88.67 | 73.09 | 61.53 | 85.52 | 71.52 |
| ImageNet-K | Noise Detector | 45.42 | - | - | 45.72 | 30.26 | 24.83 | 45.61 | 98.24 | 62.30 | 45.55 | 99.13 | 62.42 |
| | Frozen Model | 36.54 | - | - | 38.40 | 85.81 | 52.98 | 39.78 | 83.15 | 53.77 | 40.02 | 78.07 | 52.91 |
| ImageNet-A | Noise Detector | 45.52 | - | - | 45.06 | 20.25 | 26.84 | 45.49 | 57.15 | 45.39 | 45.25 | 53.84 | 39.55 |
| | Frozen Model | 45.20 | - | - | 42.84 | 70.45 | 52.86 | 43.36 | 81.06 | 56.41 | 44.06 | 73.46 | 55.00 |
| ImageNet-V2 | Noise Detector | 58.59 | - | - | 58.67 | 19.27 | 27.53 | 57.98 | 50.21 | 45.42 | 57.31 | 51.58 | 44.22 |
| | Frozen Model | 58.42 | - | - | 56.37 | 76.77 | 64.70 | 56.05 | 85.89 | 67.72 | 56.34 | 83.12 | 67.04 |
| ImageNet-R | Noise Detector | 73.43 | - | - | 73.55 | 28.53 | 27.97 | 72.97 | 97.75 | 83.56 | 72.56 | 98.50 | 83.57 |
| | Frozen Model | 71.54 | - | - | 71.23 | 91.61 | 80.05 | 71.13 | 92.28 | 80.25 | 70.95 | 88.95 | 78.79 |

Table 27: Failure case study of existing TTA methods with CIFAR-100 as ID dataset. Green indicates an improvement over ZS-CLIP in average Acc$_H$, while red indicates the opposite.

| Method | SVHN | | | LSUN | | | Texture | | | Places | | | Avg | | |
|---|---|---|---|---|---|---|---|---|---|---|---|---|---|---|---|
| | Acc$_S$ | Acc$_N$ | Acc$_H$ | Acc$_S$ | Acc$_N$ | Acc$_H$ | Acc$_S$ | Acc$_N$ | Acc$_H$ | Acc$_S$ | Acc$_N$ | Acc$_H$ | Acc$_S$ | Acc$_N$ | Acc$_H$ |
| ZS-CLIP | 48.52 | 97.58 | 64.81 | 49.29 | 94.97 | 64.90 | 46.76 | 81.58 | 59.45 | 45.36 | 64.52 | 53.27 | 47.48 | 84.66 | 60.61 |
| Tent (GT) | 62.11 | 92.92 | 74.45 | 61.28 | 89.73 | 72.83 | 60.24 | 80.42 | 68.88 | 58.55 | 65.11 | 61.66 | 60.55 | 82.05 | 69.45 (+8.85%) |
| Tent (Normal) | 55.39 | 42.41 | 48.04 | 60.06 | 83.37 | 69.82 | 59.31 | 79.13 | 67.80 | 57.52 | 62.24 | 59.79 | 58.07 | 66.79 | 61.36 (+0.75%) |
| Tent (All-update) | 52.41 | 29.85 | 38.04 | 54.74 | 59.92 | 57.21 | 58.91 | 75.83 | 66.31 | 57.98 | 61.08 | 59.49 | 56.01 | 56.67 | 55.26 (-5.35%) |
| SoTTA (GT) | 61.28 | 94.23 | 74.26 | 60.64 | 91.56 | 72.96 | 59.37 | 81.91 | 68.84 | 57.49 | 66.47 | 61.65 | 59.70 | 83.54 | 69.43 (+8.82%) |
| SoTTA (Normal) | 60.56 | 89.24 | 72.15 | 60.28 | 88.89 | 71.84 | 58.79 | 81.56 | 68.33 | 57.01 | 65.73 | 61.06 | 59.16 | 81.36 | 68.34 (+7.74%) |
| SoTTA (All-update) | 60.77 | 89.61 | 72.42 | 60.23 | 88.37 | 71.64 | 58.93 | 81.48 | 68.39 | 57.17 | 65.93 | 61.24 | 59.28 | 81.35 | 68.42 (+7.81%) |
| TPT (GT) | 54.07 | 98.11 | 69.72 | 54.77 | 95.52 | 69.62 | 52.32 | 82.86 | 64.14 | 51.20 | 67.43 | 58.20 | 53.09 | 85.98 | 65.42 (+4.81%) |
| TPT (Normal) | 46.09 | 97.87 | 62.67 | 46.90 | 95.36 | 62.88 | 43.87 | 83.10 | 57.42 | 42.48 | 66.86 | 51.95 | 44.84 | 85.80 | 58.73 (-1.88%) |
| TPT (All-update) | 52.35 | 84.64 | 64.69 | 53.84 | 87.67 | 66.71 | 51.01 | 62.39 | 56.13 | 49.87 | 39.74 | 44.23 | 51.77 | 68.61 | 57.94 (-2.67%) |

Table 28: Failure case study of existing TTA methods. Green indicates an improvement over ZS-CLIP in average $Acc_H$, while red indicates the opposite.

| ID | Method | iNaturalist $Acc_S$ | $Acc_N$ | $Acc_H$ | SUN $Acc_S$ | $Acc_N$ | $Acc_H$ | Texture $Acc_S$ | $Acc_N$ | $Acc_H$ | Places $Acc_S$ | $Acc_N$ | $Acc_H$ | Avg $Acc_S$ | $Acc_N$ | $Acc_H$ |
|---|---|---|---|---|---|---|---|---|---|---|---|---|---|---|---|---|
| CUB-200-2011 | ZS-CLIP | 38.13 | 88.06 | 53.22 | 38.10 | 87.86 | 53.15 | 37.56 | 79.11 | 50.94 | 38.00 | 87.81 | 53.04 | 37.95 | 85.71 | 52.59 |
| | Tent (GT) | 42.98 | 84.67 | 57.02 | 43.46 | 87.74 | 58.13 | 43.19 | 80.96 | 56.33 | 43.27 | 87.02 | 57.80 | 43.23 | 85.10 | 57.32 (+4.73%) |
| | Tent (Normal) | 37.02 | 46.95 | 41.40 | 38.61 | 55.55 | 45.56 | 34.98 | 41.77 | 38.07 | 40.41 | 74.83 | 52.48 | 37.75 | 54.78 | 44.38 (-8.21%) |
| | Tent (All-update) | 32.90 | 28.23 | 30.39 | 34.95 | 46.81 | 40.02 | 34.11 | 43.92 | 38.40 | 36.27 | 57.90 | 44.60 | 34.56 | 44.22 | 38.35 (-14.23%) |
| | SoTTA (GT) | 42.16 | 86.33 | 56.65 | 42.63 | 88.67 | 57.58 | 42.45 | 82.75 | 56.11 | 42.48 | 88.48 | 57.40 | 42.43 | 86.56 | 56.93 (+4.35%) |
| | SoTTA (Normal) | 41.67 | 84.37 | 55.79 | 42.08 | 86.83 | 56.69 | 41.44 | 77.58 | 54.02 | 42.04 | 86.52 | 56.59 | 41.81 | 83.82 | 55.77 (+3.19%) |
| | SoTTA (All-update) | 41.69 | 84.24 | 55.78 | 41.98 | 86.77 | 56.58 | 41.30 | 77.12 | 53.79 | 41.86 | 86.49 | 56.42 | 41.71 | 83.66 | 55.64 (+3.05%) |
| | TPT (GT) | 48.38 | 90.78 | 63.12 | 48.48 | 91.00 | 63.26 | 48.29 | 82.99 | 61.05 | 48.53 | 90.42 | 63.16 | 48.42 | 88.80 | 62.65 (+10.06%) |
| | TPT (Normal) | 37.41 | 89.57 | 52.78 | 37.49 | 89.67 | 52.87 | 36.88 | 81.67 | 50.81 | 37.44 | 89.45 | 52.79 | 37.30 | 87.59 | 52.31 (-0.27%) |
| | TPT (All-update) | 46.67 | 65.10 | 54.37 | 46.34 | 64.86 | 54.06 | 46.69 | 58.51 | 51.94 | 46.62 | 64.55 | 54.14 | 46.58 | 63.25 | 53.63 (+1.04%) |
| STANFORD-CARS | ZS-CLIP | 50.25 | 96.59 | 66.11 | 53.28 | 98.81 | 69.23 | 53.49 | 99.09 | 69.48 | 53.22 | 98.08 | 69.00 | 52.56 | 98.14 | 68.45 |
| | Tent (GT) | 52.14 | 95.00 | 67.33 | 55.22 | 98.21 | 70.69 | 55.42 | 98.25 | 70.87 | 55.16 | 97.48 | 70.45 | 54.48 | 97.23 | 69.83 (+1.38%) |
| | Tent (Normal) | 44.12 | 52.33 | 47.88 | 54.27 | 94.51 | 68.95 | 54.60 | 97.37 | 69.97 | 54.33 | 96.65 | 69.56 | 51.83 | 85.22 | 64.09 (-4.36%) |
| | Tent (All-update) | 41.25 | 40.75 | 41.00 | 42.71 | 54.01 | 47.70 | 39.10 | 33.10 | 35.85 | 44.96 | 66.23 | 53.56 | 42.01 | 48.52 | 44.53 (-23.93%) |
| | SoTTA (GT) | 52.20 | 95.86 | 67.59 | 55.05 | 98.39 | 70.60 | 55.19 | 98.64 | 70.78 | 55.02 | 97.74 | 70.41 | 54.37 | 97.66 | 69.84 (+1.39%) |
| | SoTTA (Normal) | 51.51 | 92.84 | 66.26 | 54.81 | 97.57 | 70.19 | 55.06 | 98.50 | 70.64 | 54.75 | 96.96 | 69.98 | 54.03 | 96.47 | 69.27 (+0.81%) |
| | SoTTA (All-update) | 51.32 | 92.79 | 66.09 | 54.75 | 97.66 | 70.16 | 55.08 | 98.50 | 70.65 | 54.70 | 96.78 | 69.90 | 53.96 | 96.43 | 69.20 (+0.75%) |
| | TPT (GT) | 58.16 | 97.52 | 72.86 | 60.08 | 99.00 | 74.78 | 59.81 | 99.26 | 74.64 | 59.81 | 98.50 | 74.43 | 59.47 | 98.57 | 74.18 (+5.72%) |
| | TPT (Normal) | 49.24 | 96.97 | 65.31 | 52.40 | 98.83 | 68.49 | 52.75 | 99.27 | 68.89 | 52.42 | 98.39 | 68.40 | 51.70 | 98.36 | 67.77 (-0.68%) |
| | TPT (All-update) | 55.90 | 81.32 | 66.26 | 58.08 | 89.74 | 70.52 | 59.00 | 95.38 | 72.90 | 58.13 | 90.14 | 70.68 | 57.78 | 89.14 | 70.09 (+1.64%) |
| Food-101 | ZS-CLIP | 80.63 | 94.79 | 87.14 | 80.72 | 95.98 | 87.69 | 80.50 | 93.10 | 86.34 | 80.65 | 94.59 | 87.07 | 80.62 | 94.62 | 87.06 |
| | Tent (GT) | 83.30 | 91.89 | 87.38 | 83.41 | 93.33 | 88.09 | 83.22 | 90.78 | 86.84 | 83.33 | 91.95 | 87.43 | 83.31 | 91.99 | 87.44 (+0.38%) |
| | Tent (Normal) | 75.83 | 25.09 | 37.70 | 82.86 | 85.10 | 83.97 | 82.54 | 87.03 | 84.73 | 82.26 | 80.13 | 81.18 | 80.87 | 69.34 | 71.90 (-15.16%) |
| | Tent (All-update) | 74.39 | 21.10 | 32.88 | 71.45 | 55.31 | 62.35 | 71.60 | 56.89 | 63.40 | 74.72 | 52.35 | 61.57 | 73.04 | 46.41 | 55.05 (-32.01%) |
| | SoTTA (GT) | 82.49 | 93.22 | 87.53 | 82.63 | 94.93 | 88.35 | 82.42 | 91.52 | 86.73 | 82.59 | 93.40 | 87.66 | 82.53 | 93.27 | 87.57 (+0.51%) |
| | SoTTA (Normal) | 81.84 | 84.09 | 82.95 | 82.49 | 93.34 | 87.58 | 82.05 | 90.10 | 85.89 | 82.44 | 91.62 | 86.79 | 82.20 | 89.79 | 85.80 (-1.26%) |
| | SoTTA (All-update) | 81.59 | 82.76 | 82.17 | 82.47 | 92.98 | 87.41 | 81.99 | 89.35 | 85.51 | 82.34 | 91.25 | 86.57 | 82.10 | 89.09 | 85.41 (-1.64%) |
| | TPT (GT) | 84.36 | 95.11 | 89.41 | 84.42 | 96.24 | 89.94 | 84.32 | 93.55 | 88.70 | 84.43 | 95.02 | 89.41 | 84.38 | 94.98 | 89.37 (+2.31%) |
| | TPT (Normal) | 79.70 | 94.93 | 86.65 | 79.92 | 96.19 | 87.30 | 79.70 | 93.86 | 86.20 | 79.76 | 95.14 | 86.77 | 79.77 | 95.03 | 86.73 (-0.33%) |
| | TPT (All-update) | 83.60 | 71.41 | 77.03 | 83.79 | 80.42 | 82.07 | 83.84 | 81.36 | 82.58 | 83.95 | 78.85 | 81.32 | 83.80 | 78.01 | 80.75 (-6.31%) |
| Oxford-IIIT Pet | ZS-CLIP | 78.58 | 88.31 | 83.16 | 79.77 | 87.26 | 83.35 | 80.12 | 91.17 | 85.29 | 79.56 | 86.30 | 81.86 | 79.51 | 87.76 | 83.42 |
| | Tent (GT) | 81.15 | 86.49 | 83.73 | 82.16 | 86.05 | 84.06 | 82.38 | 89.99 | 86.02 | 82.01 | 83.45 | 82.72 | 81.92 | 86.49 | 84.13 (+0.72%) |
| | Tent (Normal) | 80.07 | 78.09 | 79.07 | 81.19 | 68.30 | 74.19 | 81.48 | 74.72 | 77.95 | 80.64 | 62.51 | 70.43 | 80.84 | 70.91 | 75.41 (-8.01%) |
| | Tent (All-update) | 77.58 | 70.76 | 74.01 | 79.32 | 62.61 | 69.98 | 78.60 | 61.46 | 68.98 | 79.02 | 54.96 | 64.83 | 78.63 | 62.45 | 69.45 (-13.97%) |
| | SoTTA (GT) | 80.72 | 86.37 | 83.45 | 82.09 | 86.37 | 84.18 | 82.51 | 90.42 | 86.28 | 81.79 | 83.47 | 82.62 | 81.78 | 86.66 | 84.13 (+0.72%) |
| | SoTTA (Normal) | 80.07 | 83.54 | 81.77 | 81.78 | 83.83 | 82.79 | 82.09 | 87.52 | 84.72 | 81.49 | 81.25 | 81.37 | 81.36 | 84.03 | 82.66 (-0.75%) |
| | SoTTA (All-update) | 79.96 | 83.52 | 81.70 | 81.55 | 83.63 | 82.58 | 81.97 | 87.64 | 84.71 | 81.37 | 81.28 | 81.32 | 81.21 | 84.02 | 82.58 (-0.84%) |
| | TPT (GT) | 83.39 | 89.99 | 86.56 | 83.96 | 88.41 | 86.13 | 83.82 | 92.31 | 87.86 | 83.83 | 85.41 | 84.61 | 83.75 | 89.03 | 86.29 (+2.88%) |
| | TPT (Normal) | 77.56 | 89.71 | 83.19 | 78.87 | 89.82 | 83.99 | 79.17 | 92.26 | 85.22 | 78.62 | 87.32 | 82.74 | 78.56 | 89.78 | 83.78 (+0.37%) |
| | TPT (All-update) | 82.77 | 58.09 | 68.27 | 83.43 | 62.39 | 71.39 | 83.26 | 70.69 | 76.46 | 83.13 | 59.06 | 69.06 | 83.15 | 62.56 | 71.30 (-12.12%) |
| ImageNet | ZS-CLIP | 54.01 | 86.46 | 66.49 | 53.32 | 83.87 | 65.19 | 52.66 | 78.69 | 63.10 | 53.25 | 80.40 | 64.07 | 53.31 | 82.35 | 64.71 |
| | Tent (GT) | 56.15 | 79.49 | 65.81 | 55.93 | 78.31 | 65.25 | 55.34 | 72.69 | 62.84 | 55.81 | 75.31 | 64.11 | 55.81 | 76.45 | 64.50 (-0.21%) |
| | Tent (Normal) | 48.56 | 35.74 | 41.18 | 55.44 | 75.54 | 63.95 | 54.94 | 70.93 | 61.92 | 55.76 | 73.98 | 63.59 | 53.67 | 64.05 | 57.66 (-7.05%) |
| | Tent (All-update) | 48.08 | 31.28 | 37.90 | 53.25 | 72.27 | 61.32 | 54.25 | 68.27 | 60.46 | 54.27 | 72.26 | 61.96 | 52.46 | 61.00 | 55.41 (-9.30%) |
| | SoTTA (GT) | 55.51 | 75.20 | 63.87 | 55.32 | 75.54 | 63.87 | 54.91 | 73.13 | 62.72 | 55.25 | 73.63 | 63.13 | 55.25 | 74.38 | 63.40 (-1.32%) |
| | SoTTA (Normal) | 53.15 | 62.68 | 57.52 | 53.16 | 68.76 | 59.96 | 53.64 | 68.05 | 59.99 | 53.60 | 69.16 | 60.39 | 53.39 | 67.16 | 59.47 (-5.25%) |
| | SoTTA (All-update) | 53.06 | 61.97 | 57.17 | 52.89 | 67.70 | 59.39 | 53.59 | 66.80 | 59.47 | 53.00 | 68.06 | 59.59 | 53.14 | 66.13 | 58.91 (-5.81%) |
| | TPT (GT) | 61.95 | 88.28 | 72.81 | 61.81 | 85.44 | 71.73 | 61.26 | 80.43 | 69.55 | 61.54 | 82.33 | 70.43 | 61.64 | 84.12 | 71.13 (+6.42%) |
| | TPT (Normal) | 52.58 | 88.93 | 66.09 | 51.91 | 86.09 | 64.77 | 51.11 | 80.01 | 62.38 | 51.80 | 82.89 | 63.76 | 51.85 | 84.48 | 64.25 (-0.46%) |
| | TPT (All-update) | 60.85 | 61.41 | 61.13 | 60.97 | 62.85 | 61.90 | 60.33 | 57.91 | 59.10 | 60.70 | 61.99 | 61.34 | 60.71 | 61.04 | 60.87 (-3.85%) |
| ImageNet-K | ZS-CLIP | 34.14 | 83.35 | 48.44 | 33.32 | 81.16 | 47.24 | 32.66 | 75.53 | 45.60 | 33.37 | 77.12 | 46.58 | 33.37 | 79.29 | 46.97 |
| | Tent (GT) | 37.40 | 75.98 | 50.13 | 37.14 | 75.43 | 49.77 | 36.39 | 68.41 | 47.51 | 37.07 | 72.19 | 48.99 | 37.00 | 73.00 | 49.10 (+2.13%) |
| | Tent (Normal) | 30.46 | 26.86 | 28.55 | 36.57 | 71.82 | 48.46 | 36.37 | 66.63 | 47.06 | 36.87 | 70.32 | 48.38 | 35.07 | 58.91 | 43.11 (-3.85%) |
| | Tent (All-update) | 31.15 | 28.84 | 29.95 | 35.38 | 69.67 | 46.93 | 35.94 | 65.09 | 46.31 | 36.00 | 69.07 | 47.33 | 34.62 | 58.17 | 42.63 (-4.34%) |
| | SoTTA (GT) | 37.69 | 72.29 | 49.55 | 37.60 | 75.21 | 50.14 | 36.93 | 70.68 | 48.51 | 37.51 | 71.81 | 49.28 | 37.43 | 72.50 | 49.37 (+2.40%) |
| | SoTTA (Normal) | 36.18 | 61.70 | 45.61 | 36.28 | 67.19 | 47.12 | 35.91 | 65.31 | 46.34 | 36.57 | 67.09 | 47.34 | 36.23 | 65.32 | 46.60 (-0.36%) |
| | SoTTA (All-update) | 35.49 | 59.76 | 44.53 | 36.29 | 66.56 | 46.97 | 35.96 | 63.72 | 45.97 | 36.38 | 66.50 | 47.03 | 36.03 | 64.13 | 46.12 (-0.84%) |
| | TPT (GT) | 39.52 | 86.67 | 54.29 | 39.34 | 83.88 | 53.56 | 38.95 | 78.30 | 53.56 | 39.21 | 80.42 | 52.92 | 39.26 | 82.32 | 53.15 (+6.18%) |
| | TPT (Normal) | 32.16 | 86.52 | 46.89 | 31.55 | 83.86 | 45.85 | 30.74 | 77.39 | 44.00 | 31.56 | 80.05 | 45.27 | 31.50 | 81.95 | 45.50 (-1.46%) |
| | TPT (All-update) | 38.25 | 59.33 | 46.51 | 38.45 | 60.41 | 46.99 | 37.96 | 54.98 | 44.91 | 38.33 | 59.67 | 46.68 | 38.25 | 58.60 | 46.27 (-0.69%) |
| ImageNet-A | ZS-CLIP | 34.73 | 80.69 | 48.56 | 34.20 | 78.83 | 47.70 | 33.97 | 76.60 | 47.07 | 33.96 | 75.11 | 46.77 | 34.22 | 77.81 | 47.53 |
| | Tent (GT) | 35.51 | 79.29 | 49.05 | 34.99 | 77.60 | 48.23 | 34.75 | 75.80 | 47.65 | 34.73 | 74.24 | 47.32 | 34.99 | 76.73 | 48.06 (+0.55%) |
| | Tent (Normal) | 34.99 | 77.19 | 48.15 | 34.83 | 77.05 | 47.97 | 34.36 | 75.19 | 47.17 | 34.60 | 73.83 | 47.12 | 34.70 | 75.81 | 47.60 (+0.09%) |
| | Tent (All-update) | 34.85 | 77.48 | 48.08 | 34.07 | 76.71 | 47.18 | 33.72 | 74.89 | 46.50 | 34.11 | 73.75 | 46.65 | 34.19 | 75.71 | 47.10 (-0.41%) |
| | SoTTA (GT) | 37.09 | 78.79 | 50.44 | 36.73 | 77.72 | 49.88 | 36.25 | 76.52 | 49.19 | 36.37 | 74.36 | 48.85 | 36.61 | 76.85 | 49.59 (+2.06%) |
| | SoTTA (Normal) | 36.85 | 76.83 | 49.81 | 36.47 | 77.08 | 49.51 | 35.60 | 75.37 | 48.36 | 36.07 | 73.87 | 48.47 | 36.25 | 75.79 | 49.04 (+1.51%) |
| | SoTTA (All-update) | 36.87 | 76.93 | 49.85 | 36.55 | 77.00 | 49.57 | 35.80 | 75.08 | 48.48 | 36.37 | 73.79 | 48.72 | 36.40 | 75.70 | 49.16 (+1.63%) |
| | TPT (GT) | 45.37 | 82.39 | 58.52 | 44.60 | 80.80 | 57.47 | 44.67 | 79.19 | 57.12 | 44.45 | 77.51 | 56.50 | 44.77 | 79.97 | 57.40 (+9.88%) |
| | TPT (Normal) | 34.12 | 81.17 | 48.04 | 33.20 | 80.23 | 46.97 | 33.12 | 79.92 | 46.83 | 33.05 | 77.00 | 46.25 | 33.37 | 79.58 | 47.02 (-0.50%) |
| | TPT (All-update) | 43.31 | 56.05 | 48.86 | 43.05 | 53.93 | 47.88 | 43.68 | 58.71 | 50.09 | 42.99 | 52.81 | 47.40 | 43.26 | 55.38 | 48.56 (+1.03%) |
| ImageNet-V2 | ZS-CLIP | 48.05 | 85.77 | 61.59 | 47.43 | 83.33 | 60.45 | 46.72 | 77.70 | 58.35 | 47.45 | 79.44 | 59.41 | 47.41 | 81.56 | 59.95 |
| | Tent (GT) | 48.89 | 82.71 | 61.45 | 48.16 | 80.94 | 60.39 | 47.54 | 75.31 | 58.29 | 48.14 | 77.72 | 59.45 | 48.18 | 79.17 | 59.89 (-0.06%) |
| | Tent (Normal) | 47.94 | 76.98 | 59.08 | 48.28 | 80.50 | 60.36 | 47.56 | 74.47 | 58.05 | 48.34 | 77.37 | 59.50 | 48.03 | 77.33 | 59.25 (-0.70%) |
| | Tent (All-update) | 47.51 | 73.10 | 57.59 | 47.52 | 79.52 | 59.49 | 47.47 | 73.93 | 57.82 | 47.87 | 76.55 | 58.90 | 47.59 | 75.78 | 58.45 (-1.50%) |
| | SoTTA (GT) | 48.80 | 82.74 | 61.39 | 48.23 | 80.61 | 60.35 | 47.63 | 76.11 | 58.59 | 48.22 | 77.03 | 59.31 | 48.22 | 79.12 | 59.91 (-0.04%) |
| | SoTTA (Normal) | 48.24 | 78.59 | 59.78 | 47.80 | 78.67 | 59.47 | 47.27 | 74.82 | 57.94 | 48.26 | 76.05 | 59.05 | 47.89 | 77.03 | 59.06 (-0.89%) |
| | SoTTA (All-update) | 48.06 | 78.74 | 59.69 | 47.71 | 78.64 | 59.39 | 47.49 | 74.42 | 57.98 | 48.10 | 75.97 | 58.90 | 47.84 | 76.94 | 58.99 (-0.96%) |
| | TPT (GT) | 55.52 | 87.89 | 68.05 | 55.37 | 85.12 | 67.10 | 54.95 | 79.99 | 65.15 | 55.44 | 81.84 | 66.10 | 55.32 | 83.71 | 66.60 (+6.65%) |
| | TPT (Normal) | 46.63 | 88.37 | 61.05 | 46.12 | 85.58 | 59.94 | 45.21 | 79.14 | 57.55 | 46.02 | 81.95 | 58.94 | 46.00 | 83.76 | 59.37 (-0.58%) |
| | TPT (All-update) | 54.50 | 60.62 | 57.40 | 54.65 | 62.15 | 58.16 | 54.00 | 56.35 | 55.15 | 54.46 | 61.48 | 57.76 | 54.40 | 60.15 | 57.12 (-2.83%) |
| ImageNet-R | ZS-CLIP | 61.96 | 94.43 | 74.82 | 61.77 | 88.98 | 72.92 | 60.92 | 77.08 | 68.05 | 61.69 | 84.81 | 71.43 | 61.59 | 86.33 | 71.81 |
| | Tent (GT) | 65.48 | 92.13 | 76.55 | 65.32 | 86.94 | 74.60 | 64.63 | 75.81 | 69.77 | 65.17 | 82.68 | 72.89 | 65.15 | 84.39 | 73.45 (+1.65%) |
| | Tent (Normal) | 65.22 | 91.45 | 76.14 | 65.06 | 85.61 | 73.93 | 63.33 | 69.99 | 66.49 | 64.93 | 82.38 | 72.62 | 64.64 | 82.36 | 72.30 (+0.49%) |
| | Tent (All-update) | 64.66 | 90.75 | 75.52 | 63.73 | 84.00 | 72.47 | 62.22 | 67.19 | 64.61 | 64.30 | 81.49 | 71.88 | 63.73 | 80.86 | 71.12 (-0.69%) |
| | SoTTA (GT) | 67.58 | 91.75 | 77.83 | 67.66 | 86.68 | 76.00 | 66.98 | 76.52 | 71.43 | 67.45 | 82.49 | 74.22 | 67.42 | 84.36 | 74.87 (+3.06%) |
| | SoTTA (Normal) | 66.78 | 86.98 | 75.55 | 66.71 | 83.99 | 74.36 | 65.92 | 72.69 | 69.14 | 66.60 | 80.53 | 72.91 | 66.50 | 81.05 | 72.99 (+1.19%) |
| | SoTTA (All-update) | 66.63 | 85.68 | 74.96 | 66.90 | 83.64 | 74.34 | 65.92 | 71.65 | 68.67 | 66.63 | 80.06 | 72.73 | 66.52 | 80.26 | 72.68 (+0.87%) |
| | TPT (GT) | 70.39 | 95.01 | 80.87 | 70.24 | 89.87 | 78.85 | 69.81 | 77.91 | 73.64 | 70.24 | 85.76 | 77.23 | 70.17 | 87.14 | 77.65 (+5.84%) |
| | TPT (Normal) | 60.95 | 94.80 | 74.20 | 60.85 | 89.98 | 72.60 | 59.98 | 77.79 | 67.73 | 60.67 | 85.79 | 71.08 | 60.61 | 87.09 | 71.40 (-0.40%) |
| | TPT (All-update) | 69.10 | 72.12 | 70.58 | 69.14 | 66.38 | 67.73 | 68.64 | 56.45 | 61.95 | 68.85 | 63.42 | 66.02 | 68.93 | 64.59 | 66.57 (-5.24%) |

Table 29: Failure case study of existing TTA methods with CIFAR-10/100 as ID datasets. Green indicates an improvement over ZS-CLIP in average Acc$_H$, while red indicates the opposite.

| ID | Method | SVHN | | LSUN | | Texture | | Places | | Avg | |
|---|---|---|---|---|---|---|---|---|---|---|---|
| | | AUROC↑ | FPR95↓ | AUROC↑ | FPR95↓ | AUROC↑ | FPR95↓ | AUROC↑ | FPR95↓ | AUROC↑ | FPR95↓ |
| CIFAR-10 | ZS-CLIP | 98.45 | 6.75 | 97.75 | 10.64 | 94.75 | 28.08 | 87.47 | 50.18 | 94.60 | 23.91 |
| | Tent (GT) | 99.17 | 3.55 | 98.37 | 8.28 | 97.36 | 12.99 | 92.25 | 31.91 | 96.79 (+2.18%) | 14.18 (-9.73%) |
| | Tent (Normal) | 74.35 | 50.27 | 89.47 | 31.18 | 96.85 | 15.95 | 87.75 | 45.57 | 87.10 (-7.50%) | 35.74 (+11.83%) |
| | Tent (All-update) | 62.78 | 65.12 | 73.23 | 54.20 | 95.80 | 22.24 | 82.53 | 56.35 | 78.59 (-16.02%) | 49.48 (+25.56%) |
| | SoTTA (GT) | 99.24 | 3.13 | 98.51 | 7.24 | 97.44 | 11.89 | 92.17 | 31.41 | 96.84 (+2.24%) | 13.42 (-10.50%) |
| | SoTTA (Normal) | 95.77 | 20.74 | 97.57 | 11.68 | 97.27 | 13.02 | 91.43 | 33.91 | 95.51 (+0.91%) | 19.84 (-4.08%) |
| | SoTTA (All-update) | 93.29 | 30.24 | 97.46 | 12.79 | 97.21 | 13.76 | 91.47 | 33.75 | 94.86 (+0.25%) | 22.63 (-1.28%) |
| | TPT (GT) | 99.28 | 3.07 | 98.93 | 4.61 | 96.94 | 14.88 | 91.21 | 35.75 | 96.59 (+1.98%) | 14.58 (-9.34%) |
| | TPT (Normal) | 98.48 | 6.76 | 97.61 | 10.67 | 94.19 | 28.26 | 85.37 | 50.18 | 93.91 (-0.69%) | 23.97 (+0.05%) |
| | TPT (All-update) | 98.28 | 7.50 | 96.15 | 23.66 | 91.20 | 50.48 | 81.46 | 69.41 | 91.77 (-2.83%) | 37.76 (+13.85%) |
| CIFAR-100 | ZS-CLIP | 85.11 | 86.42 | 85.88 | 72.58 | 71.09 | 95.35 | 58.47 | 98.97 | 75.14 | 88.33 |
| | Tent (GT) | 92.11 | 40.90 | 89.09 | 52.30 | 82.14 | 67.79 | 72.01 | 87.97 | 83.84 (+8.70%) | 62.24 (-26.09%) |
| | Tent (Normal) | 46.39 | 79.90 | 84.91 | 62.45 | 80.28 | 73.90 | 68.92 | 91.80 | 70.12 (-5.01%) | 77.01 (-11.32%) |
| | Tent (All-update) | 37.15 | 94.38 | 63.31 | 80.78 | 77.80 | 79.30 | 68.91 | 91.43 | 61.79 (-13.35%) | 86.47 (-1.86%) |
| | SoTTA (GT) | 92.29 | 41.42 | 89.60 | 51.31 | 81.96 | 69.89 | 71.43 | 89.36 | 83.82 (+8.68%) | 63.00 (-25.33%) |
| | SoTTA (Normal) | 88.72 | 51.10 | 87.95 | 54.48 | 81.45 | 70.58 | 70.60 | 90.18 | 82.18 (+7.04%) | 66.59 (-21.74%) |
| | SoTTA (All-update) | 88.99 | 49.96 | 87.76 | 55.49 | 81.40 | 71.23 | 70.66 | 89.85 | 82.20 (+7.06%) | 66.63 (-21.70%) |
| | TPT (GT) | 88.66 | 76.97 | 89.25 | 63.17 | 76.87 | 90.57 | 66.27 | 97.82 | 80.26 (+5.12%) | 82.13 (-6.20%) |
| | TPT (Normal) | 84.80 | 86.43 | 85.37 | 72.58 | 69.62 | 95.34 | 55.59 | 98.97 | 73.84 (-1.29%) | 88.33 (0.00%) |
| | TPT (All-update) | 75.97 | 94.94 | 82.55 | 81.02 | 62.82 | 95.60 | 48.79 | 98.87 | 67.53 (-7.61%) | 92.61 (+4.28%) |

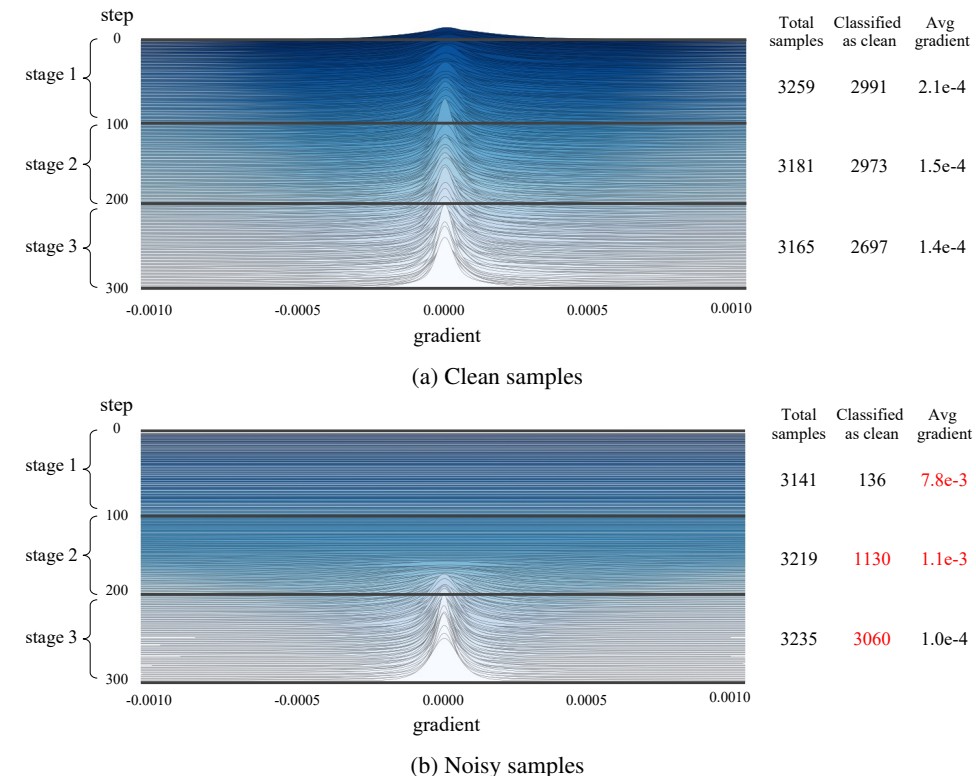

(a) Clean samples

(b) Noisy samples

Figure 9: The impact of clean and noisy samples on the gradients. Note that the gradient magnitudes of clean and noisy samples are not on the same scale; for clarity, the figure does not show gradients with magnitudes greater than 0.0010. The gradients of noisy samples are substantially larger in the first and second stages. The model effectively filters out noisy samples in the first stage but gradually struggles to distinguish between clean and noisy samples. ID set: CIFAR-10; OOD set: SVHN.

Table 30: Failure case study of existing TTA methods. Green indicates an improvement over ZS-CLIP in average Acc$_H$, while red indicates the opposite.

| ID | Method | SVHN AUROC↑ | SVHN FPR95↓ | LSUN AUROC↑ | LSUN FPR95↓ | Texture AUROC↑ | Texture FPR95↓ | Places AUROC↑ | Places FPR95↓ | Avg AUROC↑ | Avg FPR95↓ |
|---|---|---|---|---|---|---|---|---|---|---|---|
| CUB-200-2011 | ZS-CLIP | 80.79 | 59.31 | 80.18 | 61.71 | 72.79 | 69.44 | 79.84 | 62.83 | 78.40 | 63.32 |
| | Tent (GT) | 81.24 | 59.61 | 84.05 | 55.01 | 79.18 | 61.47 | 83.31 | 56.86 | 81.94 (+3.54%) | 58.24 (-5.09%) |
| | Tent (Normal) | 46.85 | 82.59 | 55.06 | 80.18 | 40.14 | 91.71 | 69.92 | 75.90 | 52.99 (-25.41%) | 82.59 (+19.27%) |
| | Tent (All-update) | 29.60 | 91.45 | 46.45 | 86.77 | 41.22 | 93.95 | 54.73 | 85.76 | 43.00 (-35.40%) | 89.48 (+26.16%) |
| | SoTTA (GT) | 81.80 | 58.61 | 84.00 | 55.12 | 79.76 | 59.37 | 83.41 | 56.69 | 82.24 (+3.84%) | 57.45 (-5.88%) |
| | SoTTA (Normal) | 79.75 | 62.44 | 81.98 | 58.85 | 74.79 | 68.23 | 81.37 | 61.23 | 79.47 (+1.07%) | 62.69 (-0.64%) |
| | SoTTA (All-update) | 79.54 | 62.84 | 81.89 | 59.22 | 74.44 | 69.00 | 81.31 | 61.23 | 79.30 (+0.89%) | 63.07 (-0.25%) |
| | TPT (GT) | 90.36 | 41.25 | 90.37 | 43.41 | 85.78 | 53.15 | 90.03 | 43.84 | 89.13 (+10.73%) | 45.41 (-17.91%) |
| | TPT (Normal) | 80.52 | 59.54 | 80.19 | 61.41 | 73.62 | 68.26 | 79.97 | 62.60 | 78.57 (+0.17%) | 62.95 (-0.37%) |
| | TPT (All-update) | 72.08 | 75.99 | 71.29 | 79.66 | 68.31 | 79.48 | 71.40 | 79.71 | 70.77 (-7.63%) | 78.71 (+15.39%) |
| STANFORD-CARS | ZS-CLIP | 94.15 | 27.03 | 98.18 | 8.26 | 98.29 | 8.64 | 97.71 | 9.79 | 97.08 | 13.43 |
| | Tent (GT) | 93.95 | 27.11 | 98.06 | 8.62 | 98.10 | 8.91 | 97.67 | 9.82 | 96.95 (-0.14%) | 13.62 (+0.19%) |
| | Tent (Normal) | 55.00 | 65.81 | 95.13 | 17.17 | 97.10 | 13.56 | 96.75 | 13.39 | 86.00 (-11.09%) | 27.48 (+14.05%) |
| | Tent (All-update) | 43.67 | 77.60 | 58.11 | 63.76 | 39.69 | 77.72 | 68.48 | 58.33 | 52.49 (-44.59%) | 69.35 (+55.92%) |
| | SoTTA (GT) | 94.47 | 26.05 | 98.10 | 8.43 | 98.28 | 8.70 | 97.75 | 9.80 | 97.15 (+0.07%) | 13.25 (-0.18%) |
| | SoTTA (Normal) | 91.62 | 34.06 | 97.31 | 11.47 | 98.09 | 9.24 | 97.37 | 11.08 | 96.10 (-0.98%) | 16.46 (+3.03%) |
| | SoTTA (All-update) | 91.40 | 33.99 | 97.45 | 10.56 | 97.93 | 9.90 | 97.24 | 11.59 | 96.01 (-1.08%) | 16.51 (+3.08%) |
| | TPT (GT) | 97.41 | 13.87 | 99.17 | 3.78 | 99.25 | 4.01 | 98.99 | 4.52 | 98.70 (+1.62%) | 6.54 (-6.89%) |
| | TPT (Normal) | 93.99 | 26.57 | 97.92 | 8.48 | 98.29 | 8.84 | 97.66 | 9.52 | 96.97 (-0.12%) | 13.35 (-0.08%) |
| | TPT (All-update) | 87.21 | 45.63 | 93.42 | 22.78 | 97.10 | 12.61 | 93.80 | 21.41 | 92.88 (-4.20%) | 25.61 (+12.18%) |
| Food-101 | ZS-CLIP | 97.71 | 11.36 | 98.10 | 10.16 | 96.52 | 13.09 | 97.60 | 13.05 | 97.48 | 11.91 |
| | Tent (GT) | 97.57 | 13.26 | 98.04 | 11.55 | 96.44 | 14.66 | 97.59 | 13.90 | 97.41 (-0.07%) | 13.34 (+1.43%) |
| | Tent (Normal) | 39.44 | 80.04 | 95.00 | 21.35 | 95.08 | 20.90 | 91.77 | 27.63 | 80.32 (-17.16%) | 37.48 (+25.57%) |
| | Tent (All-update) | 35.67 | 85.78 | 67.60 | 56.23 | 72.00 | 56.92 | 69.85 | 58.67 | 61.28 (-36.20%) | 64.40 (+52.49%) |
| | SoTTA (GT) | 97.78 | 12.00 | 98.22 | 10.28 | 96.57 | 14.13 | 97.74 | 12.94 | 97.58 (+0.10%) | 12.34 (+0.42%) |
| | SoTTA (Normal) | 94.51 | 24.94 | 97.81 | 12.34 | 95.97 | 16.73 | 97.31 | 15.01 | 96.40 (-1.08%) | 17.26 (+5.34%) |
| | SoTTA (All-update) | 93.96 | 27.45 | 97.71 | 13.12 | 95.78 | 17.53 | 97.17 | 15.76 | 96.16 (-1.33%) | 18.46 (+6.55%) |
| | TPT (GT) | 98.68 | 7.07 | 98.91 | 5.65 | 97.77 | 8.43 | 98.65 | 7.22 | 98.50 (+1.02%) | 7.09 (-4.82%) |
| | TPT (Normal) | 97.18 | 11.73 | 97.84 | 10.42 | 96.49 | 13.24 | 97.33 | 13.07 | 97.21 (-0.27%) | 12.12 (+0.20%) |
| | TPT (All-update) | 93.04 | 38.70 | 95.20 | 28.53 | 94.64 | 26.53 | 94.88 | 29.09 | 94.44 (-3.04%) | 30.71 (+18.80%) |
| Oxford-IIIT Pet | ZS-CLIP | 94.16 | 30.78 | 94.46 | 21.68 | 96.44 | 16.79 | 93.48 | 26.51 | 94.64 | 23.94 |
| | Tent (GT) | 94.80 | 28.92 | 94.91 | 21.39 | 96.69 | 16.87 | 94.07 | 26.07 | 95.12 (+0.48%) | 23.31 (-0.63%) |
| | Tent (Normal) | 90.22 | 42.32 | 85.35 | 42.35 | 89.68 | 37.86 | 81.56 | 50.01 | 86.70 (-7.93%) | 43.13 (+19.19%) |
| | Tent (All-update) | 86.45 | 55.78 | 80.66 | 50.18 | 80.20 | 55.53 | 75.25 | 58.32 | 80.64 (-14.00%) | 54.95 (+31.01%) |
| | SoTTA (GT) | 94.44 | 29.92 | 94.78 | 20.98 | 96.71 | 16.27 | 93.85 | 25.80 | 94.94 (+0.31%) | 23.24 (-0.70%) |
| | SoTTA (Normal) | 93.13 | 35.13 | 93.64 | 24.75 | 95.79 | 19.91 | 92.52 | 29.94 | 93.77 (-0.87%) | 27.43 (+3.49%) |
| | SoTTA (All-update) | 92.91 | 36.18 | 93.56 | 25.16 | 95.71 | 20.33 | 92.51 | 30.18 | 93.67 (-0.96%) | 27.96 (+4.02%) |
| | TPT (GT) | 97.80 | 13.57 | 97.60 | 12.25 | 98.59 | 8.25 | 97.19 | 16.12 | 97.80 (+3.16%) | 12.55 (-11.39%) |
| | TPT (Normal) | 93.54 | 30.85 | 94.57 | 21.05 | 96.23 | 16.26 | 93.36 | 24.59 | 94.43 (-0.21%) | 23.19 (-0.75%) |
| | TPT (All-update) | 89.66 | 53.33 | 90.17 | 43.58 | 93.48 | 34.00 | 88.89 | 47.78 | 90.55 (-4.09%) | 44.67 (+20.73%) |
| ImageNet | ZS-CLIP | 86.64 | 50.48 | 83.89 | 58.14 | 79.53 | 64.25 | 81.86 | 60.39 | 82.98 | 58.31 |
| | Tent (GT) | 83.71 | 58.57 | 82.26 | 59.91 | 77.43 | 67.63 | 80.57 | 62.80 | 80.99 (-1.99%) | 62.23 (+3.91%) |
| | Tent (Normal) | 44.17 | 85.40 | 80.35 | 63.64 | 75.98 | 69.87 | 79.70 | 64.76 | 70.05 (-12.93%) | 70.92 (+12.60%) |
| | Tent (All-update) | 40.42 | 88.23 | 76.97 | 71.66 | 73.99 | 74.25 | 77.68 | 69.64 | 67.27 (-15.72%) | 75.94 (+17.63%) |
| | SoTTA (GT) | 81.37 | 62.61 | 80.75 | 63.16 | 78.29 | 66.61 | 79.70 | 64.89 | 80.03 (-2.95%) | 64.32 (+6.00%) |
| | SoTTA (Normal) | 71.13 | 77.69 | 75.14 | 71.50 | 73.72 | 73.10 | 75.81 | 71.74 | 73.95 (-9.03%) | 73.51 (+15.19%) |
| | SoTTA (All-update) | 70.64 | 78.51 | 74.42 | 72.73 | 73.32 | 73.59 | 75.12 | 72.88 | 73.38 (-9.61%) | 74.43 (+16.11%) |
| | TPT (GT) | 92.46 | 37.37 | 90.62 | 43.32 | 87.41 | 51.49 | 89.38 | 46.77 | 89.97 (+6.99%) | 44.74 (-13.58%) |
| | TPT (Normal) | 85.80 | 49.83 | 83.83 | 57.09 | 79.05 | 64.33 | 81.89 | 59.53 | 82.64 (-0.34%) | 57.70 (-0.62%) |
| | TPT (All-update) | 77.46 | 67.63 | 79.04 | 67.53 | 75.26 | 74.62 | 78.47 | 68.83 | 77.56 (-5.42%) | 69.65 (+11.34%) |
| ImageNet-K | ZS-CLIP | 75.13 | 78.60 | 71.38 | 83.15 | 66.30 | 85.24 | 69.02 | 82.68 | 70.46 | 82.42 |
| | Tent (GT) | 74.04 | 77.74 | 73.11 | 77.83 | 66.63 | 83.49 | 70.89 | 78.90 | 71.17 (+0.71%) | 79.49 (-2.93%) |
| | Tent (Normal) | 28.86 | 93.52 | 69.55 | 81.79 | 64.34 | 86.03 | 68.69 | 82.37 | 57.86 (-12.60%) | 85.93 (+3.51%) |
| | Tent (All-update) | 31.41 | 93.29 | 66.33 | 86.45 | 62.78 | 87.26 | 66.81 | 85.18 | 56.83 (-13.62%) | 88.05 (+5.63%) |
| | SoTTA (GT) | 72.08 | 77.26 | 73.44 | 76.55 | 68.62 | 81.10 | 71.41 | 77.44 | 71.39 (+0.93%) | 78.09 (-4.33%) |
| | SoTTA (Normal) | 61.84 | 85.96 | 65.98 | 84.10 | 63.25 | 85.66 | 66.28 | 84.24 | 64.34 (-6.12%) | 84.99 (+2.57%) |
| | SoTTA (All-update) | 60.37 | 86.86 | 65.73 | 84.49 | 62.59 | 86.17 | 65.93 | 84.55 | 63.66 (-6.80%) | 85.52 (+3.10%) |
| | TPT (GT) | 83.51 | 69.23 | 80.76 | 74.80 | 76.34 | 79.50 | 78.80 | 76.23 | 79.85 (+9.40%) | 74.94 (-7.48%) |
| | TPT (Normal) | 74.55 | 78.10 | 71.52 | 82.15 | 65.70 | 85.08 | 69.18 | 82.05 | 70.24 (-0.22%) | 81.84 (-0.57%) |
| | TPT (All-update) | 63.59 | 86.87 | 65.12 | 85.90 | 60.34 | 90.42 | 64.26 | 87.02 | 63.33 (-7.13%) | 87.55 (+5.13%) |
| ImageNet-A | ZS-CLIP | 76.23 | 68.27 | 72.73 | 76.64 | 70.65 | 78.17 | 70.04 | 78.07 | 72.41 | 75.29 |
| | Tent (GT) | 75.68 | 69.65 | 72.53 | 77.04 | 70.41 | 78.79 | 70.08 | 78.47 | 72.17 (-0.24%) | 75.99 (+0.70%) |
| | Tent (Normal) | 73.67 | 72.67 | 71.90 | 77.77 | 69.71 | 79.51 | 69.53 | 79.07 | 71.20 (-1.21%) | 77.25 (+1.97%) |
| | Tent (All-update) | 73.75 | 72.64 | 71.38 | 79.05 | 69.10 | 80.17 | 69.22 | 79.60 | 70.86 (-1.55%) | 77.87 (+2.58%) |
| | SoTTA (GT) | 76.00 | 69.33 | 73.66 | 75.27 | 72.01 | 76.83 | 71.11 | 76.89 | 73.20 (+0.78%) | 74.58 (-0.71%) |
| | SoTTA (Normal) | 74.29 | 71.40 | 73.00 | 75.65 | 70.84 | 78.63 | 70.41 | 77.28 | 72.14 (-0.28%) | 75.74 (+0.45%) |
| | SoTTA (All-update) | 74.29 | 71.84 | 73.00 | 75.48 | 70.73 | 78.73 | 70.42 | 77.60 | 72.11 (-0.30%) | 75.91 (+0.62%) |
| | TPT (GT) | 84.33 | 58.37 | 81.88 | 67.53 | 80.45 | 68.52 | 79.71 | 71.19 | 81.59 (+9.18%) | 66.40 (-8.88%) |
| | TPT (Normal) | 74.46 | 68.41 | 71.32 | 76.63 | 70.66 | 77.85 | 69.01 | 78.23 | 71.36 (-1.05%) | 75.28 (-0.01%) |
| | TPT (All-update) | 64.16 | 87.03 | 63.25 | 86.79 | 66.39 | 83.79 | 61.68 | 89.65 | 63.87 (-8.54%) | 86.81 (+11.53%) |
| ImageNet-V2 | ZS-CLIP | 83.54 | 60.57 | 80.49 | 67.14 | 75.84 | 72.67 | 78.36 | 69.03 | 79.56 | 67.35 |
| | Tent (GT) | 82.25 | 62.78 | 79.93 | 67.93 | 75.18 | 73.55 | 77.94 | 69.41 | 78.83 (-0.73%) | 68.42 (+1.06%) |
| | Tent (Normal) | 77.74 | 68.87 | 79.65 | 67.76 | 74.68 | 73.93 | 77.75 | 69.90 | 77.45 (-2.10%) | 70.12 (+2.76%) |
| | Tent (All-update) | 75.04 | 72.01 | 78.67 | 71.31 | 74.11 | 75.03 | 77.09 | 71.47 | 76.23 (-3.33%) | 72.45 (+5.10%) |
| | SoTTA (GT) | 82.30 | 63.32 | 79.58 | 68.54 | 75.83 | 73.04 | 77.80 | 70.07 | 78.88 (-0.68%) | 68.74 (+1.39%) |
| | SoTTA (Normal) | 78.71 | 69.58 | 78.31 | 70.26 | 74.67 | 74.84 | 76.99 | 72.12 | 77.17 (-2.39%) | 71.70 (+4.35%) |
| | SoTTA (All-update) | 78.72 | 69.89 | 78.15 | 70.67 | 74.67 | 75.14 | 76.96 | 72.11 | 77.11 (-2.45%) | 71.95 (+4.60%) |
| | TPT (GT) | 89.97 | 48.36 | 87.98 | 54.92 | 84.31 | 61.75 | 86.48 | 57.28 | 87.19 (+7.63%) | 55.58 (-11.77%) |
| | TPT (Normal) | 82.67 | 60.16 | 80.43 | 66.19 | 75.24 | 72.61 | 78.39 | 68.48 | 79.18 (-0.38%) | 66.86 (-0.49%) |
| | TPT (All-update) | 73.47 | 75.07 | 75.18 | 74.93 | 70.93 | 81.09 | 74.53 | 75.83 | 73.53 (-6.03%) | 76.73 (+9.38%) |
| ImageNet-R | ZS-CLIP | 90.99 | 49.03 | 87.88 | 56.24 | 79.39 | 70.05 | 85.26 | 58.80 | 85.88 | 58.53 |
| | Tent (GT) | 91.08 | 48.73 | 88.46 | 54.89 | 80.91 | 68.98 | 85.94 | 57.86 | 86.60 (+0.72%) | 57.62 (-0.91%) |
| | Tent (Normal) | 90.45 | 51.24 | 87.60 | 57.99 | 77.01 | 73.38 | 85.41 | 59.69 | 85.12 (-0.76%) | 60.58 (+2.05%) |
| | Tent (All-update) | 89.59 | 55.75 | 85.96 | 61.95 | 74.76 | 76.14 | 84.51 | 62.67 | 83.70 (-2.17%) | 64.13 (+5.60%) |
| | SoTTA (GT) | 91.36 | 47.40 | 89.30 | 50.67 | 82.60 | 63.73 | 86.63 | 54.50 | 87.47 (+1.59%) | 54.07 (-4.46%) |
| | SoTTA (Normal) | 88.40 | 57.05 | 87.46 | 55.56 | 79.79 | 69.31 | 85.09 | 58.76 | 85.19 (-0.69%) | 60.17 (+1.64%) |
| | SoTTA (All-update) | 87.64 | 59.58 | 87.37 | 56.37 | 79.50 | 69.86 | 84.93 | 59.50 | 84.86 (-1.02%) | 61.33 (+2.80%) |
| | TPT (GT) | 95.13 | 28.09 | 93.09 | 37.64 | 86.96 | 53.23 | 91.08 | 40.58 | 91.56 (+5.69%) | 39.89 (-18.64%) |
| | TPT (Normal) | 90.58 | 49.41 | 87.30 | 56.40 | 78.51 | 69.69 | 84.51 | 59.32 | 85.22 (-0.66%) | 58.70 (+0.17%) |
| | TPT (All-update) | 84.12 | 71.49 | 82.01 | 72.54 | 75.36 | 78.05 | 79.35 | 75.21 | 80.21 (-5.67%) | 74.32 (+15.79%) |

