# OpenReview forum: "Noisy Test-Time Adaptation in Vision-Language Models"
_ICLR.cc/2025/Conference — ICLR 2025 Poster_

### Official Review · Reviewer_SvHV · 2024-10-29

**Soundness:** 3
**Presentation:** 3
**Contribution:** 2
**Rating:** 6
**Confidence:** 3

**Summary:**

The paper introduces a novel setting called Zero-Shot Noisy TTA and builds benchmarks. It provides a theoretical analysis of why existing TTA methods struggle in noisy scenarios. The paper proposes AdaND, which decouples the classifier and detector to identify noisy data. Experiments show that AdaND works well in ZS-NTTA and achieves computational efficiency.

**Strengths:**

The paper is well-organized, and the idea of the Zero-Shot Noisy TTA setting is novel. It offers a comprehensive analysis of existing methods that suffer from Zero-Shot Noisy TTA, and this analysis is easy to follow. The results demonstrate a notable improvement across many benchmarks, and the proposed method AdaND is computationally efficient.

**Weaknesses:**

Though Zero-shot Noisy TTA is a new task, the method of utilizing two stages and pseudo labels to train a noisy detector is not novel; the idea of using unlabel test data to train a more robust OOD classifier has been explored in other scenarios, such as [1] for zero-shot OOD detection.

It would be better to provide more experiments with more complex datasets, such as near ID/OOD datasets, for example, Imagenet and NINCO[2].

Some recent test-time adaptation works are not compared in the experiments, such as [4], which adopts a training-free adapter to achieve efficiency. I am concerned whether the noise detector can be replaced by another training-free module to be more efficient for test time and get better performance, such as cache memory.

[1] Du, Xuefeng, et al. "How Does Unlabeled Data Provably Help Out-of-Distribution Detection?." ICLR 2024
[2] Bitterwolf, Julian, Maximilian Mueller, and Matthias Hein. "In or out? fixing imagenet out-of-distribution detection evaluation."
[3] Karmanov, Adilbek, et al. "Efficient Test-Time Adaptation of Vision-Language Models." In CVPR 2024.

**Questions:**

As the authors analyze in Section 3.3, some inaccuracies during the early stages of TTA can gradually lead the model to overfit. I wonder whether the pseudo-labels in the early stages are accurate and the inaccuracies in the Noise Detector will accumulate.

---

> ### Author Response · Authors · 2024-11-21
> **Response to Reviewer SvHV (1/2)**
>
> We thank the reviewer SvHV for the valuable feedback. We addressed all the comments. Please find the point-to-point responses below. Any further comments and discussions are welcomed!
>
> **W1:** Though Zero-shot Noisy TTA is a new task, the method of utilizing two stages and pseudo labels to train a noisy detector is not novel; the idea of using unlabel test data to train a more robust OOD classifier has been explored in other scenarios, such as [1] for zero-shot OOD detection.
>
> **Reply:**
> Thanks for your feedback. We would like to clarify our method and contribution here:
> - First, we introduced a practical and previously overlooked setting - Zero-shot Noisy TTA (ZS-NTTA). All Reviewers acknowledge this as a novel and important contribution. This is our paper's first key contribution to the field.
> - Second, we comprehensively analyzed why existing TTA methods suffer from performance decline and underperform the model-frozen method in the ZS-NTTA setting (detailed in Section 3). These analyses, culminating in three novel and insightful observations, constitute our second major contribution. **Reviewer JjFG also offered the following feedback:**
> >The three observations are interesting and novel. They reveal the reasons why the ZS-NTTA is difficult and contributes to the community.
> - Third, regarding methodology, we would like to emphasize that **our primary contribution and novelty lies in the conceptual framework rather than specific implementation details**. We identified limitations in existing TTA approaches through detailed analysis and proposed a simple yet effective decoupled framework - separating the classifier and detector while keeping the classifier frozen. This conceptual design prevents detrimental effects caused by the classifier adapting to noisy samples. While our current implementation uses a two-stage approach with pseudo labels, this is just one possible implementation within the framework. We believe this conceptual insight and framework design can inspire future research in this field.
> **Reviewer JjFG also offered the following feedback:**
> >The proposed method is reasonable and achieve good results in the experiment.
>
> **Reviewer GcDg also offered the following feedback:**
> >Although the method of training an additional noise detector is simple, it proves to be quite effective according to the experimental results.
>
> - Finally, our experimental results demonstrate substantial improvements over existing methods, validating the effectiveness of our approach.
>
> We discuss and cite SAL [1] in related work (Appendix B) in our revised paper. While SAL [1] also leverages unlabeled test data to train a robust OOD classifier, our work differs in its focus and contribution. We primarily address the ZS-NTTA task, where our core contribution lies in proposing a conceptual framework that decouples the detector from the classifier. This decoupling prevents classifier degradation during noisy sample adaptation, with pseudo-label-based detector training serving merely as one implementation detail of our approach.
>
> **W2:** It would be better to provide more experiments with more complex datasets, such as near ID/OOD datasets, for example, Imagenet and NINCO.
>
> **Reply:** Thanks for your feedback. Following your suggestion, we conducted additional experiments on ImageNet and NINCO datasets, with results shown in Table A. The results demonstrate that our method can outperform all the baseline methods. What’s more, our method achieves the best ID classification accuracy $Acc_S$ among all approaches. We have added this experiment to our revised paper in Appendix E.
>
> Table A: Experiments on more complex datasets, ID: ImageNet, OOD: NINCO
> |  |  Acc_S | Acc_N |  Acc_H |
> | :---: | :---: | :---: | :---: |
> | ZS-CLIP |51.44 | 71.90 | 59.97 |
> | Tent | 54.14 | 65.84 | 59.42 |
> | SoTTA | 52.87 | 60.50 | 56.43 |
> | Ours (With Gaussian noise) | **60.10** | 55.70 | 57.82 |
> | Ours (Without Gaussian noise) |50.25 | **77.99**  | **61.12** |
>
> Note that ZS-NTTA is inherently more challenging than traditional OOD detection, as it requires simultaneous classification and detection capabilities under the noisy data stream. Specifically, ZS-NTTA requires online, real-time classification and detection results: for each input sample, the model must immediately determine whether it is ID/OOD, and if ID, perform classification. In contrast, existing OOD detection methods typically report ID classification accuracy under the assumption of a clean data stream.

---

> ### Author Response · Authors · 2024-11-21
> **Response to Reviewer SvHV (2/2)**
>
> **W3:** Some recent test-time adaptation works are not compared in the experiments, such as TDA [3].
>
> **Reply:** Thanks for your valuable feedback. We conducted comparisons with TDA, with results shown in Table B. Our experimental results demonstrate that TDA's performance is inferior to our approach. This indicates the necessity of training a noise detector to detect noisy samples in the ZS-NTTA setting. We have added this experiment to our revised paper in Appendix E.
>
> Table B: Performance comparison with TDA using ImageNet as ID dataset. Results are averaged across four OOD datasets: iNaturalist, SUN, Texture, and Places.
> |  |  Acc_S | Acc_N |  Acc_H |
> | :---: | :---: | :---: | :---: |
> | ZS-CLIP | 53.38 | 82.38 | 64.77 |
> | TDA | 53.47 | 82.37 | 64.84 |
> | Ours (With Gaussian noise) | **62.24** | 88.67 | **73.09** |
> | Ours (Without Gaussian noise) | 60.64  | **91.73** | 73.00 |
>
>
> **Q1:** As the authors analyze in Section 3.3, some inaccuracies during the early stages of TTA can gradually lead the model to overfit. I wonder whether the pseudo-labels in the early stages are accurate and the inaccuracies in the Noise Detector will accumulate
>
> **Reply:** Thanks for your feedback. We would like to clarify that the pseudo-labels in both stages are generated by ZS-CLIP in our method. Therefore, the inaccuracies in pseudo-labels are similar between the first and second stages. We only use ZS-CLIP's outputs as final results in the early stages and switch to the noise detector's outputs in the second stage, considering that the Noise Detector may not be sufficiently trained initially.
>
> Following your suggestion, we report the accuracy of pseudo-labels generated by ZS-CLIP in both the early stage and the second stage in Table C. Even with relatively low pseudo-label accuracy on ImageNet, our method achieves strong performance (a notable improvement of 8.32% in harmonic mean accuracy compared to ZS-CLIP).
>
> Furthermore, we have discussed in Appendix F.4 (Table 24) why we use ZS-CLIP's outputs rather than the noise detector's outputs as pseudo-labels. Using noise detector's outputs as pseudo-labels could lead to cumulative errors, resulting in significant performance degradation on certain datasets. Therefore, using ZS-CLIP's outputs as pseudo-labels proves to be more robust. For detailed analysis and results, please refer to Appendix F.4 and Table 24.
>
>
> Table C: Accuracy of pseudo-labels. ID dataset: ImageNet, OOD dataset: iNaturalist
> | | The first stage | The second stage |
> | :---: | :---: | :---: |
> | Pseudo-label accuracy (clean samples) | 65.88% | 66.52% |
> | Pseudo-label accuracy (noisy samples) | 83.62% | 86.50%|

---

> ### Author Response · Authors · 2024-11-24
> **Would you mind checking our responses and confirming whether you have any further questions?**
>
> Dear Reviewer SvHV,
>
> Thank you again for your time and valuable comments.
>
> In the rebuttal period, we have provided detailed responses to all your comments and questions point-by-point. Specifically, we
> - clarify the novelty of our method lies in decoupling the classifier and detector according to our analysis (**Weakness1**);
> - provide more experiments with more complex datasets (**Weakness2**);
> - conduct comparisons with TDA (**Weakness3**);
> - report the accuracy of pseudo-labels (**Question1**).
>
> **As the deadline for the paper discussion phase is approaching**, would you mind checking our responses and confirming whether you have any further questions?
>
> Any comments and discussions are welcome!
>
> Thanks for your attention and best regards,
> Authors of #7153

---

> > ### Comment · Reviewer_SvHV · 2024-11-26
> > **Thanks for response**
> >
> > Thanks for the author's response. I have reviewed the rebuttal as well as the comments from the other reviewers. I have the same question as reviewer GcDg: ZS-NTTA simultaneously emphasizes both ID classification and OOD detection, but in paper [1], for the Noisy TTA setting, it emphasizes the classification results of three corruption benchmarks: ImageNet-C, CIFAR10-C, and CIFAR100-C. Therefore, I believe the experiments should focus on classification under corrupted datasets; however, the current version does not include these results. In light of this, I will not raise my current rating.
> >
> >
> >
> > [1] SoTTA: Robust Test-Time Adaptation on Noisy Data Streams

---

> > > ### Author Response · Authors · 2024-11-29
> > > **Further Response to Reviewer SvHV (1/3)**
> > >
> > > The authors appreciate reviewer SvHV’s valuable feedback.
> > >
> > > Following your suggestion, we conducted experiments on ImageNet-C, CIFAR10-C, and CIFAR100-C, with results shown in the tables below.  **These experiments demonstrate that our method outperforms existing methods on corrupted datasets**. Due to time constraints, we use iNaturalist as the OOD dataset for ImageNet-C and SVHN as the OOD dataset for CIFAR-10-C and CIFAR-100-C. We will incorporate these experimental results into our paper and conduct experiments on more OOD datasets.
> > >
> > > In fact, since our approach is based on VLMs, the datasets selected for our experiments follow TPT [a], which focuses on zero-shot TTA. And our paper already includes experiments on ImageNet-Adversarial, ImageNet-Sketch, and ImageNet-Rendition. These datasets exhibit significant domain gaps from ImageNet, and our method demonstrates superior performance across these datasets.
> > >
> > > Furthermore, we note that SoTTA lacks the capability to detect noisy samples, which we consider a significant limitation since directly classifying noisy samples as clean samples could lead to severe consequences. This is why ZS-NTTA simultaneously emphasizes both ID classification and OOD detection.
> > >
> > > Based on the above responses, together with the original contents in our draft, we strongly believe that our approach is powerful and generalizable. We have demonstrated its superior performance on different domains and corruption datasets. Therefore, we would kindly request that you reconsider our paper and your rating.
> > >
> > > [a] Shu, Manli, et al. "Test-time prompt tuning for zero-shot generalization in vision-language models." In NeurIPS 2022.
> > >
> > >
> > > Table 1: Experiments on CIFAR-10-C, CIFAR-100-C, and ImageNet-C. Results are averaged across 15 corruption types.
> > > |   | CIFAR-10-C |  | | CIFAR-100-C | |  | ImageNet-C | | |
> > > | :---: | :---: | :---: | :---: | :---: | :---: | :---: | :---: | :---: | :---: |
> > > |  |  Acc_S | Acc_N |  Acc_H | Acc_S | Acc_N |  Acc_H |  Acc_S | Acc_N |  Acc_H |
> > > | ZS-CLIP | 74.14 | 98.24 | 83.96 | 39.43 | 96.74 | 55.49 |  44.14 | 84.73 | 57.91 |
> > > | Tent | 80.25 | 57.86 | 66.83 | 45.29 | 41.15 | 42.83 |  37.10 | 16.50 | 22.79 |
> > > | SoTTA | 83.68 | 85.00 | 84.06 |51.46 | 88.70 | 64.73 | 46.41 | 62.48 | 53.21 |
> > > | Ours | 82.15 | 99.89 | **89.84** | 54.89 | 99.18 | **70.18** |52.63 | 94.78 | **67.55** |

---

> > > ### Author Response · Authors · 2024-11-29
> > > **Further Response to Reviewer SvHV (3/3)**
> > >
> > > Table 3.2: Experiments on CIFAR-100-C with 7 types of corruptions. The remaining 8 corruption types are shown in Table 3.2.
> > > | corruption   | frost | |  | fog | | | brightness | | | contrast | | |  elastic_transform | | | pixelate | | | jpeg_compression | | |
> > > | :---: | :---: | :---: | :---: | :---: | :---: | :---: | :---: | :---: | :---: | :---: | :---: | :---: | :---: | :---: | :---: | :---: | :---: | :---: | :---: | :---: | :---: |
> > > |  |  Acc_S | Acc_N |  Acc_H | Acc_S | Acc_N |  Acc_H |  Acc_S | Acc_N |  Acc_H | Acc_S | Acc_N |  Acc_H |  Acc_S | Acc_N |  Acc_H |  Acc_S | Acc_N |  Acc_H | Acc_S | Acc_N |  Acc_H |
> > > |ZS-CLIP| 43.62 | 97.20 | 60.22  | 48.72 | 97.51 | 64.98  | 47.93 | 97.53 | 64.27  | 47.99 | 97.49 | 64.32  | 36.66 | 96.80 | 53.18  | 42.33 | 97.13 | 58.96  | 33.01 | 96.25 | 49.16  |
> > > | Tent| 49.03 | 43.12 | 45.89  | 54.30 | 42.17 | 47.47  | 54.52 | 42.78 | 47.94  | 54.00 | 41.84 | 47.15  | 42.06 | 40.20 | 41.11  | 48.76 | 43.73 | 46.11  | 36.98 | 38.85 | 37.89 |
> > > | SoTTA| 54.41 | 87.95 | 67.23  | 60.04 | 88.55 | 71.56  | 60.09 | 89.21 | 71.81  | 59.57 | 87.80 | 70.98  | 49.05 | 89.26 | 63.31  | 55.81 | 91.24 | 69.26  | 43.25 | 86.96 | 57.77 |
> > > |Ours| 59.58 | 99.65 | **74.57**  | 64.29 | 99.73 | **78.18**  | 63.73 | 99.78 | **77.78**  | 63.50 | 99.70 | **77.59**  | 51.98 | 99.39 | **68.26**  | 57.89 | 99.66 | **73.24**  | 47.53 | 99.23 | **64.27** |
> > >
> > >
> > >
> > > Table 4.1: Experiments on ImageNet-C with 8 types of corruptions. The remaining 7 corruption types are shown in Table 4.2.
> > > | corruption  | gaussian_noise |  | | shot_noise | |  | impulse_noise | | | defocus_blur | | | glass_blur | | |  motion_blur | | | zoom_blur | | | snow | | |
> > > | :---: | :---: | :---: | :---: | :---: | :---: | :---: | :---: | :---: | :---: | :---: | :---: | :---: | :---: | :---: | :---: | :---: | :---: | :---: | :---: | :---: | :---: | :---: | :---: | :---: |
> > > |  |  Acc_S | Acc_N |  Acc_H | Acc_S | Acc_N |  Acc_H |  Acc_S | Acc_N |  Acc_H | Acc_S | Acc_N |  Acc_H |  Acc_S | Acc_N |  Acc_H |  Acc_S | Acc_N |  Acc_H | Acc_S | Acc_N |  Acc_H | Acc_S | Acc_N |  Acc_H |
> > > | ZS-CLIP| 45.21 | 85.33 | 59.10  | 44.38 | 85.20 | 58.36  | 36.76 | 83.41 | 51.03  | 40.34 | 83.49 | 54.40  | 39.08 | 83.72 | 53.29  | 47.20 | 85.09 | 60.72  | 34.55 | 82.20 | 48.65  | 41.70 | 84.41 | 55.82 |
> > > | Tent| 39.04 | 16.62 | 23.31  | 38.15 | 16.86 | 23.39  | 31.91 | 16.35 | 21.62  | 35.13 | 17.31 | 23.19  | 32.98 | 16.93 | 22.37  | 38.75 | 16.63 | 23.27  | 30.71 | 16.13 | 21.15  | 34.65 | 15.97 | 21.86  |
> > > | SoTTA| 46.67 | 62.33 | 53.38  | 45.92 | 63.23 | 53.20  | 41.24 | 60.74 | 49.13  | 44.78 | 63.75 | 52.61  | 44.60 | 64.33 | 52.68  | 48.15 | 63.32 | 54.70  | 41.23 | 62.18 | 49.58  | 44.28 | 60.26 | 51.05 |
> > > | Ours| 53.71 | 96.49 | **69.01**  | 52.73 | 96.08 | **68.09**  | 43.41 | 93.06 | **59.20**  | 49.73 | 92.58 | **64.70**  | 46.42 | 92.91 | **61.91**  | 56.79 | 95.37 | **71.19**  | 41.65 | 88.66 | **56.68**  | 49.21 | 94.45 | **64.71** |
> > >
> > >
> > > Table 4.2: Experiments on ImageNet-C with 7 types of corruptions. The remaining 8 corruption types are shown in Table 4.1.
> > > | corruption   | frost | |  | fog | | | brightness | | | contrast | | |  elastic_transform | | | pixelate | | | jpeg_compression | | |
> > > | :---: | :---: | :---: | :---: | :---: | :---: | :---: | :---: | :---: | :---: | :---: | :---: | :---: | :---: | :---: | :---: | :---: | :---: | :---: | :---: | :---: | :---: |
> > > |  |  Acc_S | Acc_N |  Acc_H | Acc_S | Acc_N |  Acc_H |  Acc_S | Acc_N |  Acc_H | Acc_S | Acc_N |  Acc_H |  Acc_S | Acc_N |  Acc_H |  Acc_S | Acc_N |  Acc_H | Acc_S | Acc_N |  Acc_H |
> > > |ZS-CLIP | 43.81 | 84.99 | 57.82  | 48.50 | 85.33 | 61.85  | 53.39 | 86.34 | 65.98  | 50.33 | 85.51 | 63.36  | 45.38 | 84.93 | 59.15  | 46.87 | 85.53 | 60.56  | 44.57 | 85.40 | 58.57 |
> > > | Tent| 34.85 | 16.24 | 22.16  | 41.33 | 16.15 | 23.22  | 42.89 | 16.55 | 23.88  | 42.83 | 16.56 | 23.88  | 36.21 | 16.15 | 22.34  | 38.97 | 16.37 | 23.06  | 38.14 | 16.61 | 23.14 |
> > > | SoTTA| 44.35 | 60.66 | 51.24  | 49.88 | 63.20 | 55.76  | 52.28 | 62.39 | 56.89  | 51.34 | 63.41 | 56.74  | 46.89 | 63.05 | 53.78  | 48.10 | 61.72 | 54.07  | 46.44 | 62.70 | 53.36  |
> > > |Ours | 50.45 | 95.41 | **66.00**  | 58.01 | 95.68 | **72.23**  | 62.65 | 96.78 | **76.06**  | 60.70 | 96.16 | **74.42**  | 53.89 | 94.83 | **68.72**  | 56.13 | 96.30 | **70.92**  | 53.96 | 97.00 | **69.34** |

---

> > > ### Author Response · Authors · 2024-12-01
> > > **Would you mind checking our further responses and confirming if you have any remaining questions?**
> > >
> > > Dear Reviewer SvHV,
> > >
> > > We sincerely appreciate your valuable feedback.
> > >
> > > As the deadline for the paper discussion phase is approaching, we would like to check if you have any other remaining concerns about our paper.
> > >
> > > We sincerely thank you for your dedication and effort in evaluating our submission. Please do not hesitate to let us know if you need any clarification.
> > >
> > > Thanks for your attention and best regards,
> > >
> > > Authors of #7153

---

> ### Author Response · Authors · 2024-11-29
> **Further Response to Reviewer SvHV (2/3)**
>
> Below are the detailed results across all 15 corruption types.
>
>
> Table 2.1: Experiments on CIFAR-10-C with 8 types of corruptions. The remaining 7 corruption types are shown in Table 2.2.
> | corruption  | gaussian_noise |  | | shot_noise | |  | impulse_noise | | | defocus_blur | | | glass_blur | | |  motion_blur | | | zoom_blur | | | snow | | |
> | :---: | :---: | :---: | :---: | :---: | :---: | :---: | :---: | :---: | :---: | :---: | :---: | :---: | :---: | :---: | :---: | :---: | :---: | :---: | :---: | :---: | :---: | :---: | :---: | :---: |
> |  |  Acc_S | Acc_N |  Acc_H | Acc_S | Acc_N |  Acc_H |  Acc_S | Acc_N |  Acc_H | Acc_S | Acc_N |  Acc_H |  Acc_S | Acc_N |  Acc_H |  Acc_S | Acc_N |  Acc_H | Acc_S | Acc_N |  Acc_H | Acc_S | Acc_N |  Acc_H |
> | ZS-CLIP | 64.52 | 98.12 | 77.85  | 72.39 | 98.19 | 83.34  | 78.95 | 98.31 | 87.57  | 82.81 | 98.32 | 89.90  | 37.83 | 97.71 | 54.54  | 77.43 | 98.28 | 86.62  | 74.46 | 98.26 | 84.72  | 78.55 | 98.30 | 87.32 |
> |Tent| 75.34 | 69.58 | 72.35  | 80.19 | 64.64 | 71.58  | 83.43 | 55.78 | 66.86  | 86.70 | 53.91 | 66.48  | 50.95 | 58.89 | 54.63  | 82.72 | 56.67 | 67.26  | 79.46 | 55.57 | 65.40  | 83.44 | 58.96 | 69.10 |
> |SoTTA| 78.95 | 94.91 | **86.20**  | 82.82 | 93.59 | 87.88  | 85.98 | 77.80 | 81.69  | 89.43 | 83.76 | 86.50  | 61.23 | 81.34 | **69.87**  | 85.63 | 80.88 | 83.19  | 83.56 | 84.52 | 84.04  | 85.99 | 83.20 | 84.57 |
> |Ours| 75.23 | 99.91 | 85.83  | 81.43 | 99.93 | **89.74**  | 85.97 | 99.93 | **92.43**  | 88.48 | 99.90 | **93.84**  | 51.55 | 99.77 | 67.98  | 85.17 | 99.90 | **91.95**  | 82.18 | 99.88 | **90.17**  | 86.15 | 99.91 | **92.52**|
>
>
>
>
>
> Table 2.2: Experiments on CIFAR-10-C with 7 types of corruptions. The remaining 8 corruption types are shown in Table 2.1.
> | corruption   | frost | |  | fog | | | brightness | | | contrast | | |  elastic_transform | | | pixelate | | | jpeg_compression | | |
> | :---: | :---: | :---: | :---: | :---: | :---: | :---: | :---: | :---: | :---: | :---: | :---: | :---: | :---: | :---: | :---: | :---: | :---: | :---: | :---: | :---: | :---: |
> |  |  Acc_S | Acc_N |  Acc_H | Acc_S | Acc_N |  Acc_H |  Acc_S | Acc_N |  Acc_H | Acc_S | Acc_N |  Acc_H |  Acc_S | Acc_N |  Acc_H |  Acc_S | Acc_N |  Acc_H | Acc_S | Acc_N |  Acc_H |
> |ZS-CLIP| 79.35 | 98.34 | 87.83  | 82.95 | 98.34 | 89.99  | 83.21 | 98.37 | 90.16  | 83.53 | 98.36 | 90.34  | 72.87 | 98.21 | 83.66  | 76.77 | 98.29 | 86.21  | 66.49 | 98.17 | 79.28|
> |Tent| 84.05 | 56.82 | 67.80  | 86.73 | 54.83 | 67.19  | 87.20 | 54.01 | 66.70  | 86.83 | 53.62 | 66.30  | 78.25 | 57.89 | 66.55  | 83.69 | 56.33 | 67.34  | 74.80 | 60.42 | 66.85 |
> |SoTTA| 86.95 | 85.02 | 85.97  | 89.28 | 81.39 | 85.15  | 89.71 | 81.18 | 85.23  | 89.46 | 82.55 | 85.87  | 81.78 | 85.20 | 83.45  | 86.44 | 85.22 | 85.83  | 77.95 | 94.39 | 85.39 |
> |Ours| 86.74 | 99.92 | **92.86**  | 88.73 | 99.90 | **93.98**  | 89.09 | 99.90 | **94.19**  | 89.10 | 99.89 | **94.19**  | 80.97 | 99.89 | **89.44**  | 84.34 | 99.90 | **91.46**  | 77.10 | 99.89 | **87.03** |
>
>
>
> Table 3.1: Experiments on CIFAR-100-C with 8 types of corruptions. The remaining 7 corruption types are shown in Table 3.2.
> | corruption  | gaussian_noise |  | | shot_noise | |  | impulse_noise | | | defocus_blur | | | glass_blur | | |  motion_blur | | | zoom_blur | | | snow | | |
> | :---: | :---: | :---: | :---: | :---: | :---: | :---: | :---: | :---: | :---: | :---: | :---: | :---: | :---: | :---: | :---: | :---: | :---: | :---: | :---: | :---: | :---: | :---: | :---: | :---: |
> |  |  Acc_S | Acc_N |  Acc_H | Acc_S | Acc_N |  Acc_H |  Acc_S | Acc_N |  Acc_H | Acc_S | Acc_N |  Acc_H |  Acc_S | Acc_N |  Acc_H |  Acc_S | Acc_N |  Acc_H | Acc_S | Acc_N |  Acc_H | Acc_S | Acc_N |  Acc_H |
> |ZS-CLIP| 29.28 | 96.13 | 44.89  | 34.70 | 96.66 | 51.07  | 41.45 | 97.34 | 58.14  | 47.67 | 97.46 | 64.02  | 15.85 | 92.22 | 27.05  | 40.96 | 97.17 | 57.63  | 39.41 | 96.98 | 56.04  | 41.81 | 97.25 | 58.48 |
> |Tent| 37.89 | 41.58 | 39.65  | 42.34 | 43.03 | 42.68  | 45.03 | 41.28 | 43.07  | 54.09 | 42.27 | 47.46  | 20.11 | 33.50 | 25.13  | 46.91 | 40.80 | 43.64  | 44.61 | 38.69 | 41.44  | 48.65 | 43.36 | 45.85  |
> |SoTTA| 44.03 | 90.11 | 59.16  | 48.15 | 90.16 | 62.77  | 52.07 | 91.95 | 66.49  | 59.61 | 89.35 | 71.51  | 27.10 | 84.24 | **41.01**  | 53.18 | 88.15 | 66.34  | 51.34 | 87.30 | 64.66  | 54.20 | 88.22 | 67.15  |
> |Ours| 45.65 | 98.85 | **62.46**  | 52.25 | 99.30 | **68.47**  | 57.64 | 99.62 | **73.03**  | 63.57 | 99.73 | **77.65**  | 26.04 | 94.39 | 40.82  | 57.18 | 99.47 | **72.62**  | 54.45 | 99.51 | **70.39**  | 58.13 | 99.66 | **73.43** |

---

### Official Review · Reviewer_GcDg · 2024-10-31

**Soundness:** 3
**Presentation:** 2
**Contribution:** 3
**Rating:** 6
**Confidence:** 5

**Summary:**

This paper introduces a novel task setting termed zero-shot noisy Test-Time Adaptation (ZS-NTTA), focusing on adapting models to target data containing noisy samples during test-time in a zero-shot manner. To address this challenge, the authors propose the AdaND method, which trains a noise detector online using detected noisy samples. The results of the experiments are reported in terms of classification accuracy and Out-of-Distribution (OOD) detection metrics.

**Strengths:**

1. The proposed task addresses issues commonly encountered in real-world applications of classification models, which are often overlooked in current research. For instance, existing OOD detection methods assume a clean stream when reporting accuracy, and clip models are applied in noisy TTA tasks for zero-shot usage.
2. Although the method of training an additional noise detector is simple, it proves to be quite effective according to the experimental results.
3. There is improved performance in both classification and OOD detection metrics.

**Weaknesses:**

1. I have concerns regarding the definition of the task name. The ultimate goal of the proposed ZS-NTTA task is to detect OOD samples and correctly classify In-Distribution (ID) samples. I understand the definitions of noisy TTA and OOD detection, where both aim to handle test sets containing both ID and OOD cases. In my view, TTA leans more towards a methodological strategy focusing on enhancing model performance and adaptability during testing, while OOD detection leans towards a task definition, primarily studying how to handle test datasets with noise/OOD samples. Thus, from this perspective, the task might be more appropriately named something like "test-time OOD detection," as there are similar task definitions in academia, such as [1,2], which require discussion and comparison.

[1] Test-Time Linear Out-of-Distribution Detection, CVPR2024

[2] Atta: Anomaly-aware test-time adaptation for out-of-distribution detection in segmentation. NIPS2023

2. Assuming there are no issues with the paper's title or task definition, and considering that TTA focuses more on classification accuracy while OOD detection focuses on the capability of detecting OOD/noise, there appears to be a mismatch between the methodology and the title of the paper. Specifically, the method emphasizes training an OOD detector with a fixed classifier, focusing on OOD detection capability rather than improving classification accuracy. I suspect the improvement in classification accuracy reported in the paper's tables is due to successful OOD detection rather than enhancements to the classifier, correct?
3. TTA methods are generally sensitive to the ratio of clean to noise in samples, ranging from 1:1000 to 1000:1. Is the performance of this method stable across such variations?
4. Line 376 is ambiguous and could lead readers to misunderstand that the clean and noisy streams are known. If I understand correctly, the paper suggests that the test stream is unknown, and that inserting Gaussian noise can handle various unknown test stream scenarios, including both clean and noisy streams.

**Questions:**

See weaknesses.

---

> ### Author Response · Authors · 2024-11-21
> **Response to Reviewer GcDg (1/3)**
>
> We thank the reviewer GcDg for the valuable feedback. We addressed all the comments. Please find the point-to-point responses below. Any further comments and discussions are welcomed!
>
> **W1:** The task might be more appropriately named something like "test-time OOD detection," as there are similar task definitions in academia, such as [1,2], which require discussion and comparison.
>
> **Reply:** Thank you for your valuable feedback. While Test-time OOD detection and ZS-NTTA share some similarities, such as having both ID and OOD samples in the test data stream, they are fundamentally different tasks. We can summarize their differences in three key points:
> - The first distinction lies in their objectives. ZS-NTTA simultaneously **emphasizes both ID classification and OOD detection**, with results presented using harmonic mean accuracy. In contrast, test-time OOD detection **primarily focuses on detection**, with results evaluated using the area under the ROC curve.
> - Secondly, **regarding ID classification, the test-time OOD detection task only requires evaluating ID classification under the assumption of a clean data stream.** However, the ZS-NTTA task requires evaluating ID classification and OOD detection under the noisy data stream. From this perspective, ZS-NTTA is more challenging than Test-time OOD detection.
> - Furthermore, **ZS-NTTA requires online, real-time classification and detection results**: for each input sample, the model must immediately determine whether it is ID/OOD, and if ID, perform classification. Existing OOD detection/test-time OOD detection methods are evaluated by processing the entire test set or batch, determining the optimal threshold based on ground truth.
>
> Besides, **the noisy TTA setting has gained recognition in academia (OWTTT [a], SoTTA [b])**, and given our utilization of pretrained VLM's zero-shot capabilities, we refer to our setting as ZS-NTTA.
>
> In terms of task definitions, a comprehensive comparison is further summarized in Table A. Based on the above analysis of the differences, we believe that we need a new name for the new setting.
>
> Table A: Comparison between ZS-NTTA and test-time OOD detection setting[1, 2]
> |  | [1] | [2] | ZS-NTTA |
> | :---: | :---: | :---: | :---: |
> | Focus on ID classification |$\times$ | $\times$ | $\checkmark$ |
> | Focus on OOD detection | $\checkmark$ | $\checkmark$ | $\checkmark$ |
> | Evaluate ID classification |Clean data stream | Clean data stream | Noisy data stream |
> | Metrics | AUROC, FPR95 | AUROC, FPR95 | Harmonic mean accuracy: $Acc_H$ |
> | Domain shift | $\times$ | $\checkmark$ | $\checkmark$ |
> | Online evaluation | $\times$ | $\times$ | $\checkmark$ |
> | Zero-shot | $\times$ | $\times$  | $\checkmark$ |
>
> In addition to task definitions, we discuss and compare AdaND with [1, 2] (in terms of method) to further illustrate the distinctions.
> - RTL [1]: RTL used linear regression to make a more precise OOD prediction. In other words, RTL leverages the TTA method to enhance OOD detection while fundamentally remaining an OOD detection task. However, our work fundamentally differs in its focus: we focus on the TTA setting itself, where test samples may contain noise, resulting in severe performance degradation of existing TTA methods.
> - ATTA [2]: ATTA cannot be extended to the ZS-NTTA setting since it relies on measuring the distributional distance between test and training features in the normalization layers of the segmentation network. In the context of pretrained VLMs like CLIP, we don't have access to the training data, making ATTA's approach inapplicable to our setting.
>
> We have added the above discussion and comparison to our paper in Appendix A.
>
> [1] Fan, Ke, et al. "Test-Time Linear Out-of-Distribution Detection." In CVPR 2024.
>
> [2] Gao, Zhitong, et al. "ATTA: Anomaly-aware Test-Time Adaptation for Out-of-Distribution Detection in Segmentation." In NeurIPS 2023.
>
> [a] Li, Yushu, et al. "On the Robustness of Open-World Test-Time Training: Self-Training with Dynamic Prototype Expansion." In ICCV 2023.
>
> [b] Gong, Taesik, et al. "SoTTA: Robust Test-Time Adaptation on Noisy Data Streams." In NeurIPS 2023.

---

> ### Author Response · Authors · 2024-11-21
> **Response to Reviewer GcDg (2/3)**
>
> **W2:** There appears to be a mismatch between the methodology and the title of the paper. I suspect the improvement in classification accuracy reported in the paper's tables is due to successful OOD detection rather than enhancements to the classifier, correct?
>
> **Reply:** Your understanding of the improved classification accuracy stemming from successful OOD detection is correct. However, **this does not mean our method is a mismatch with the task**. We would like to clarify this further.
>
> First of all, it's crucial to distinguish between tasks and methods, as methods serve as a means to accomplish the task's goals. In the ZS-NTTA setting, our goal is to effectively perform both classification and detection in an online manner. **After analyzing how mainstream TTA methods perform in the ZS-NTTA setting  (detailed in Section 3), our analysis led us to decouple the classifier and detector**, focusing on developing an individual detector while keeping the classifier frozen. This conceptual design prevents detrimental effects caused by the classifier adapting to noisy samples. Our experimental results demonstrate that this strategy successfully achieves both classification and detection objectives, aligning well with the task requirements.
>
> Furthermore, we would like to emphasize that **TTA does not necessarily require model adaptation**. Several existing TTA works, such as [c], [d], and [e], have successfully achieved their objectives while maintaining fixed models. These approaches demonstrate that effective test-time adaptation can be achieved through various strategies beyond model updating.
>
> In summary, our method was specifically designed to address the challenges of the ZS-NTTA task. Based on empirical observations and analysis, we developed an approach that strongly correlates with the ZS-NTTA’s goal, focusing on effective classification and OOD detection.
>
> [c] Karmanov, Adilbek, et al. "Efficient Test-Time Adaptation of Vision-Language Models." In CVPR 2024.
>
> [d] Zhang, Taolin, et al. "BoostAdapter: Improving Vision-Language Test-Time Adaptation via Regional Bootstrapping." In NeurIPS 2024.
>
> [e] Han, Zongbo, et al. "DOTA: Distributional Test-Time Adaptation of Vision-Language Models." arXiv preprint arXiv:2409.19375 (2024).

---

> ### Author Response · Authors · 2024-11-21
> **Response to Reviewer GcDg (3/3)**
>
> **W3:** TTA methods are generally sensitive to the ratio of clean to noise in samples, ranging from 1:1000 to 1000:1. Is the performance of this method stable across such variations?
>
> **Reply:**
> Thanks for your feedback. In the noisy TTA setting, existing method [a] typically experiments with ID:OOD ratios ranging from 1:0.2 to 1:1. In the OOD detection literature [f, g, h], a balanced 1:1 ratio is commonly used as the standard evaluation setting.
>
> In our paper, we have performed experiments with various noise ratios (0%, 25%, 50%, 75%; i.e., 1:0, 3:1, 1:1, 1:3) in the data stream. The results in Table 17 show that our method maintains excellent performance across data streams with the above different noise ratios.
>
> Taking your suggestion, we additionally evaluated several practical settings with noise ratios ranging from 5:1 to 1:5. The results, presented in Table A, further demonstrate our method's stability across various noise ratios.
>
> Table A: Different noise ratios in the data stream. Results
> are averaged across four OOD datasets: SVHN, LSUN, Texture, and Places. ID: CIFAR-10.
> | ID:OOD  |  1:5 |  | |1:4 | |  | 1:3 |  |   | 1:2 | | | 1:1 |  |   | 2:1 | | | 3:1 | | | 4:1|||  5:1  |  |   |
> | :---: | :---: | :---: | :---: | :---: | :---: | :---: | :---: | :---: | :---: | :---: | :---: | :---: | :---: | :---: | :---: | :---: | :---: | :---: | :---: | :---: | :---: | :---: | :---: | :---: | :---: | :---: | :---: |
> |  |  Acc_S | Acc_N |  Acc_H | Acc_S | Acc_N |  Acc_H | Acc_S | Acc_N |  Acc_H | Acc_S | Acc_N |  Acc_H | Acc_S | Acc_N |  Acc_H |Acc_S | Acc_N |  Acc_H |Acc_S | Acc_N |  Acc_H |Acc_S | Acc_N |  Acc_H |Acc_S | Acc_N |  Acc_H |
> | ZS-CLIP | 83.67 | 89.84 | 86.40 | 83.52 | 90.05 | 86.42|83.29 | 90.27 | 86.42 |83.03 | 90.62 | 86.45| 82.64 | 91.07 | 86.47 |82.19 | 91.42 | 86.39| 81.83 | 91.82 | 86.39 |81.57 | 92.17 | 86.42| 81.27 | 92.42 | 86.37 |
> | Tent | 72.65 | 19.38 | 29.96 |75.90 | 22.89 | 34.32|80.99 | 30.33 | 42.93 | 84.50 | 43.89 | 56.23 | 88.69 |70.19 | 77.78| 88.99 | 87.23 | 87.88 | 88.62 | 89.07 | 88.66 | 88.27 | 89.80 | 88.83 |88.23 | 90.06 | 88.95 |
> | SoTTA | 89.24 | 56.77 | 68.09 |89.77 | 58.95 | 69.51 |90.14 | 64.30 | 73.56|90.14 | 72.39 | 79.53| 89.73 | 84.47 | 86.88 |88.80 | 89.77 | 89.11| 88.29 | 90.58 | 89.26 |88.01 | 91.14 | 89.40| 87.91 | 91.31 | 89.43 |
> | TPT | 81.96 | 90.43 | 85.76 | 81.66 | 90.61 | 85.69|81.53 | 90.83 | 85.72| 81.19 | 91.11 | 85.68|80.90 | 91.52 | 85.72| 80.38 | 91.84 | 85.57 |80.02 | 92.22 | 85.56 |79.77 | 92.45 |85.51| 79.51 | 92.74 | 85.50 |
> | Ours | 88.90 | 94.47 | 91.43 |88.98 | 95.05 | 91.78| 89.10 |95.75 | 92.21 |89.17 | 96.67 | 92.70| 89.32 | 97.79 | 93.34| 89.37 | 97.41 |93.20|89.29 | 95.85 | 92.43 |89.37 | 93.69 | 91.41 | 89.19 | 91.54 | 90.23 |
>
> [f] Hendrycks, Dan, et al. "A baseline for detecting misclassified and out-of-distribution examples in neural networks." In ICLR 2017.
>
> [g] Liu, Weitang, et al. "Energy-based out-of-distribution detection." In NeurIPS 2020.
>
> [h] Ming, Yifei, et al. "Delving into out-of-distribution detection with vision-language representations." In NeurIPS 2022.
>
> **W4:** Line 376 is ambiguous and could lead readers to misunderstand that the clean and noisy streams are known.
>
> **Reply:** Thank you for your valuable feedback. You are correct in your understanding of our approach. We have revised Line 376 to avoid ambiguity and better reflect that the test stream is unknown during deployment. The revised text now reads: "During testing, we insert a Gaussian noise sample for every M input sample in the data stream, regardless of whether the stream is clean or noisy. Note that we don't have prior knowledge about whether the data stream is clean or noisy."

---

> > ### Comment · Reviewer_GcDg · 2024-11-21
> >
> > Thank you for the responses provided. However, there are still some areas of confusion that remain unaddressed, specifically:
> >
> > 1. As stated, "ZS-NTTA simultaneously emphasizes both ID classification and OOD detection. In contrast, test-time OOD detection primarily focuses on detection." My confusion lies in the fact that the method presented in the paper seems to focus on how to perform OOD detection online without making substantial contributions to improving ID classification. While TTA methods can "maintain fixed models," they are still expected to enhance ID classification, such as by introducing a dynamic adapter for improvement, as suggested in reference [c]. In this paper, although there appears to be an improvement in ID classification, this enhancement seems to stem primarily from better OOD differentiation, with ID classification performance remaining identical to the original CLIP. Therefore, I believe that the main contribution of this paper lies in OOD detection rather than both ID classification and OOD detection.
> > 2. Regarding "online, real-time classification and detection" and the "assumption of a clean data stream," I agree with the author's perspective that the ZS-NTTA is different from OOD detection in these aspects.  However, a more fundamental point is whether there is an effective strategy to adaptively determine the threshold for OOD detection during the testing process, and whether this strategy can generalize across different datasets. Using a threshold for OOD detection can also address online classification with a noisy stream; however, the optimal threshold may vary across different datasets, and manually adjusting this threshold is impractical in practical applications. Does the paper propose a method for automatically generating this threshold? Does the threshold generation strategy require tuning, such as adjusting hyperparameters, for different datasets? Overall, I am curious whether a single set of threshold strategies (with fixed hyperparameters) could be applicable across various datasets.

---

> > > ### Author Response · Authors · 2024-11-21
> > > **Further Response to Reviewer GcDg (1/1)**
> > >
> > > The authors appreciate reviewer GcDg's valuable feedback. Below is our response to your two further comments.
> > >
> > > **Comment 1:** The method presented in the paper seems to focus on how to perform OOD detection online without making substantial contributions to improving ID classification.
> > >
> > > **Reply:**
> > > Thanks for your feedback. We would like to clarify our method further.
> > >
> > > - Previous TTA methods for enhancing ID classification typically assume a clean test stream scenario. However, our analysis demonstrates that in noisy TTA settings, inadequate handling of noisy samples can significantly degrade classifier performance. While our OOD detector does not directly enhance the classifier's intrinsic classification capabilities, it effectively filters out noisy samples that would otherwise degrade classifier performance, thus **preserving the good performance (or significantly avoiding degradation of the performance) of classification accuracy on ID samples**.
> > >
> > > - Furthermore, **our method is plug-and-play, making it readily compatible with existing TTA approaches for enhanced classifiers’ classification capability in noisy data streams**. To demonstrate this, we integrated our method with Tent: after updating the OOD detector for N steps (N=10 in our implementation), we leverage samples identified as ID by the detector to update the classifier. Experimental results in Table 1 below show that our method not only improves OOD sample detection performance but also **enhances the classifier's intrinsic classification capabilities on ID samples under noisy data streams**. This demonstrates that our approach can effectively boost the robustness of existing TTA methods in noisy data streams.
> > >
> > >
> > > Table 1:  Results of integrating AdaND (Ours) with Tent for enhanced classifiers' classification performance in noisy data streams. Results are averaged across four OOD datasets (SVHN, LSUN, Texture, and Places) with CIFAR-10 as the ID dataset.
> > > |  |  Acc_S | Acc_N |  Acc_H |
> > > | :---: | :---: | :---: | :---: |
> > > | ZS-CLIP | 82.64 | 91.07 | 86.47 |
> > > | Tent | 88.69 | 70.19 | 77.78 |
> > > | Ours | 89.32 | **97.79** | 93.34|
> > > | Ours (with Tent) | **93.61** | 94.30 | **93.79** |
> > >
> > > **Comment 2.1:** Does the paper propose a method for automatically generating this threshold?
> > >
> > > **Reply:** In our paper, we employ an adaptive threshold to perform online, real-time classification and detection **without manually adjusting this threshold across different datasets** (Detailed in Line 176 in Section 2). According to OWTTT [a], the distribution of OOD scores follows a bimodal distribution. Therefore, we can minimize intra-class variance to determine the adaptive threshold:
> > > $\min_{\lambda} \frac{1}{N_{\text{id}}} \sum_{i} [S(x_i) - \frac{1}{N_{\text{id}}} \sum_{j} \mathbb{1}(S(x_j) > \lambda)S(x_j)]^2 + \frac{1}{N_{\text{ood}}} \sum_{i} [S(x_i) - \frac{1}{N_{\text{ood}}} \sum_{j} \mathbb{1}(S(x_j) \leq \lambda)S(x_j)]^2$
> > >
> > > $N_{\text{id}} = \sum_{i}^{N_q} \mathbb{1}(S(x_i) > \lambda)$, $N_{\text{ood}} = \sum_{i}^{N_q} \mathbb{1}(S(x_i) \leq \lambda)$, and $N_q$ is the length of a queue at test-time to update the score distribution. However, the score in OWTTT relies on source prototypes, which are unavailable in pre-trained VLMs. In our paper, we propose using the MCM score, i.e., the softmax of CLIP's cosine similarity score, as an alternative.
> > >
> > > [a] Li, Yushu, et al. "On the Robustness of Open-World Test-Time Training: Self-Training with Dynamic Prototype Expansion." In ICCV 2023.
> > >
> > > **Comment 2.2:** Does the threshold generation strategy require tuning, such as adjusting hyperparameters, for different datasets? Overall, I am curious whether a single set of threshold strategies (with fixed hyperparameters) could be applicable across various datasets.
> > >
> > > **Reply:** **Our method maintains consistent hyperparameters across all datasets.** Moreover, we comprehensively compared adaptive and fixed thresholds ranging from 0.1 to 0.9, as shown in Figure 6 in our paper. Experimental results demonstrate that adaptive threshold consistently outperforms single fixed threshold across different datasets.

---

> > > > ### Author Response · Authors · 2024-11-24
> > > > **Would you mind checking our further responses and confirming if you have any remaining questions?**
> > > >
> > > > Dear Reviewer GcDg,
> > > >
> > > > Thank you again for your time and valuable comments.
> > > >
> > > > We have provided detailed responses to your two further concerns and would like to follow up on our previous discussion.
> > > >
> > > > Regarding your first concern, we have clarified that **our method is plug-and-play, making it readily compatible with existing TTA approaches for enhancing classifiers' performance in noisy data streams** (as demonstrated in Table 1 of our previous response). Moreover, our method significantly **prevents performance degradation on ID samples**.
> > > >
> > > > Regarding your second concern, we have clarified that **our method maintains consistent hyperparameters across all datasets**, ensuring simplicity and generalizability.
> > > >
> > > > **As the deadline for the paper discussion phase is approaching**, we would like to confirm whether our additional responses have adequately addressed your concerns or if you have any remaining questions about our paper.
> > > >
> > > > Any comments and discussions are welcome!
> > > >
> > > > Thanks for your attention and best regards,
> > > > Authors of #7153

---

> > > > > ### Comment · Reviewer_GcDg · 2024-11-25
> > > > >
> > > > > Thanks for the clarification. Now, I have a deeper understanding of this paper and I will slightly improve my score to 6. However, there are still some limitations. For example, the paper combines adaptive thresholding and MCM score, both of which are directly borrowed from other papers. Inserting Gaussian Noise is the author's contribution, but it is difficult to independently support an entire paper. The proposed ZS-NTTA is essentially an NTTA problem, only extended to a zero-shot setting by leveraging CLIP's excellent zero-shot recognition capabilities. The main reason for my increased score is that this is the first paper, to my best knowledge, that applies CLIP to NTTA. I intend to encourage further exploration of VLMs in the NTTA domain.

---

> > > > > > ### Author Response · Authors · 2024-11-25
> > > > > > **Thanks for raising the score**
> > > > > >
> > > > > > We thank the reviewer for raising the score!   Moreover, we sincerely thank the reviewer for recognizing both our task definition and the match between our methodology and the ZS-NTTA setting.
> > > > > >
> > > > > > Regarding the contribution, we would like to clarify three key aspects further:
> > > > > > - First, as you noted, we are the first to introduce this practical and previously overlooked zero-shot NTTA setting. This is our paper's first key contribution to the field.
> > > > > > - Second, we comprehensively analyzed why existing TTA methods suffer from performance decline and underperform the model-frozen method in the ZS-NTTA setting (detailed in Section 3). These analyses, culminating in three novel and insightful observations, constitute our second major contribution. Reviewer JjFG also offered the following feedback:
> > > > > > >The three observations are interesting and novel. They reveal the reasons why the ZS-NTTA is difficult and contributes to the community.
> > > > > > - Third, regarding methodology, we would like to emphasize that our primary contribution and novelty lies in the conceptual framework, i.e., decoupling the classifier and detector. This conceptual design prevents detrimental effects caused by the classifier adapting to noisy samples. We demonstrate our method’s effectiveness through extensive experiments.
> > > > > >
> > > > > > Overall, our work contributes by (1) investigating an under-explored but important problem; (2) providing an insightful analysis of why existing methods fail under noisy conditions, supported by experiments that highlight the impact of noisy samples on model gradients and adaptation performance; and (3) proposing an effective and simple framework.
> > > > > >
> > > > > > We hope our work serves as a baseline for research in VLM-based NTTA and draws more attention to this important task. We believe our decoupling framework can provide valuable insights for future work in this direction. In future work, we aim to explore (1) how to leverage detected noisy samples as negative training data to enhance classifier robustness further; and (2) how to utilize OOD detector outputs to generate more accurate pseudo-labels.

---

### Official Review · Reviewer_JjFG · 2024-11-02

**Soundness:** 3
**Presentation:** 3
**Contribution:** 3
**Rating:** 6
**Confidence:** 4

**Summary:**

This paper deals with the test-time adaptation (TTA) task and extends it to a challenging scenario, i.e., zero-shot noisy test-time adaptation (ZS-NTTA). The ZS-NTTA  adapts the source model to deal with the noisy target data during test-time and allow the target data to be noisy. This work first studies which factors can decline the performance in the ZS-NTTA task, then proposes a new method to overcome these problems.

**Strengths:**

1. This work studies a challenging task of zero-shot noisy test-time adaptation, which is practical and applicable to in real-world applications.
2. The three observations are interesting and novel. They reveals the reasons why the NS-TTA is difficult and contribute to the community.
3. The proposed method is reasonable and achieve good results in the experiment.

**Weaknesses:**

1. The proposed method studies the ranking distribution of different methods in Fig. 2. Can these ranking really tell the superiority of these methods? It is possible that one method is good at dealing with challenging datasets, and the other is good at easier datasets. Why not evaluate them using the absolute accuracy (instead of the ranks in this work)?
2. How to inject the noise to the data is an important factor of the proposed method. Is there any theoretical analysis?
3. It is better to study how the portion of the noisy labels influence the performance.

**Questions:**

1. Please discuss the reason to evaluate the methods using rank in Fig. 2.
2. It is better to analyze the theoretical foundation of injecting the noise to the data using the proposed method.
3. It is better to study how the portion of the noisy labels influence the performance.

---

> ### Author Response · Authors · 2024-11-21
> **Response to Reviewer JjFG (1/2)**
>
> We thank the reviewer JjFG for the valuable feedback. We addressed all the comments. Please find the point-to-point responses below. Any further comments and discussions are welcomed!
>
>
> **W1:** Why use the ranking distribution in Fig. 2 rather than using the absolute accuracy?
>
> **Reply:**
> Thank you for your feedback. One important advantage of the violin ranking visualization is its ability to mitigate the impact of extreme cases. When using absolute accuracy, if a method performs particularly poorly on individual datasets, these extreme values significantly affect the average accuracy, leading to biased evaluation of the method's overall performance. For instance, on ImageNet-K/iNaturalist (ID/OOD), Tent achieves an $\text{Acc}_\text{H}$ of only 28.55%, while ZS-CLIP reaches 48.49%; however, on ImageNet-K/SUN, Tent achieves an Acc_H of 48.46%, while ZS-CLIP achieves 47.39%. Tent's extremely low value substantially reduces its average performance, overlooking its excellent performance on other datasets. In contrast, using ranking distribution visualization provides a more balanced evaluation perspective that doesn't overly amplify individual extreme cases.
> What’s more, we added the absolute accuracy for all methods across 44 ID-OOD dataset pairs (see Table A below). Our method still achieves the best performance, while most TTA methods perform worse than ZS-CLIP under the ZS-NTTA setting. Following your suggestion, we have added the absolute accuracy to our paper in Figure 7 in our revised paper
>
> Table A: Average absolute accuracy for all methods across 44 ID-OOD dataset pairs
> |   | ZS-CLIP | Tent | SoTTA | TPT | AdaND (Ours) |
> | :---: | :---: | :---: | :---: | :---: | :---: |
> | $\text{Acc}_\text{H}$ | 66.34%| 61.35% | 66.89% | 65.68% | **74.70%** |
>
> **W2:**  How to inject the noise to the data is an important factor of the proposed method. Is there any theoretical analysis?
>
> **Reply:**
> In fact, the main purpose of noise injection is to cover completely clean data streams. In our method, we have claimed that the injected noise should meet two conditions: 1) lying outside the ID label space and 2) being easily accessible without incurring extra costs for auxiliary data collection. More importantly, we conducted comprehensive ablation studies on different types of injected noise and noise injection frequencies. Our extensive experiments on 44 ID-OOD dataset pairs demonstrate the effectiveness of our strategy, which achieves strong performance while successfully handling completely clean test streams. Given that our noise injection strategy is designed to address the practical challenge of handling various test stream compositions and does not involve algorithm convergence or performance bounds derivation, we focus on empirical validation rather than theoretical analysis.

---

> ### Author Response · Authors · 2024-11-21
> **Response to Reviewer JjFG (2/2)**
>
> **W3:** It is better to study how the portion of the noisy labels influences the performance.
>
> **Reply:**
> We are uncertain whether "the portion of the noisy labels" referred to by the reviewer means the proportion of noisy samples in the test stream or the ratio of class numbers between noisy and clean datasets.
> Regarding the proportion of noisy samples in the test stream, we have already conducted ablation studies with various noise ratios (0%, 25%, 50%, 75%) to mimic real-world conditions. Our method demonstrates superior performance across different noise ratios. The detailed experimental results can be found in Table 17 of our paper, where we observe consistent performance improvements over baseline methods regardless of the noise ratio.
> For the ratio of class numbers between noisy and clean datasets, our experiments encompass 44 ID-OOD dataset pairs with diverse ratios of class numbers. The number of classes for each ID-OOD dataset pair is summarized in Table B below. Our method consistently performs best across all different ID-OOD class ratios. We have added the class ratio between noisy and clean datasets to our revised paper in Table 9.
>
>
> Table B: Number of classes in ID and OOD datasets. Each row shows an ID-OOD dataset pair with their respective number of classes.
> |  ID/OOD | iNaturalist | SUN | Texture | Places | SVHN | LSUN |
> | :---: | :---: | :---: | :---: | :---: | :---: | :---: |
> | CUB-200-2011 | 200:110| 200:50 | 200:47 | 200:50 | - | - |
> | STANFORD-CARS | 196:110| 196:50 | 196:47 | 196:50 | - | - |
> | Food-101 | 101:110| 101:50 | 101:47 | 101:50 | - | - |
> | Oxford-IIIT Pet | 37:110| 37:50 | 37:47 | 37:50 | - | - |
> | ImageNet | 1000:110| 1000:50 | 1000:47 | 1000:50 | - | - |
> | ImageNet-K | 1000:110| 1000:50 | 1000:47 | 1000:50 | - | - |
> | ImageNet-A | 200:110| 200:50 | 200:47 | 200:50 | - | - |
> | ImageNet-V2 | 1000:110| 1000:50 | 1000:47 | 1000:50 | - | - |
> | ImageNet-R | 200:110| 200:50 | 200:47 | 200:50 | - | - |
> | CIFAR-10 | - | - | 10:47 | 10:50 | 10:10 | 10:10 |
> | CIFAR-100 | - | - | 100:47 | 100:50 | 100:10 | 100:10 |
>
> Note: To avoid label space overlap between ID and OOD datasets, the iNaturalist, SUN, and Places datasets used in our experiments are subsets constructed by [1].
>
> [1] Huang, Rui, et al. "MOS: Towards Scaling Out-of-distribution Detection for Large Semantic Space" CVPR 2021

---

> ### Author Response · Authors · 2024-11-24
> **Would you mind checking our responses and confirming whether you have any further questions?**
>
> Dear Reviewer JjFG,
>
> Thank you again for your time and valuable comments.
>
> In the rebuttal period, we have provided detailed responses to all your comments and questions point-by-point. Specifically, we
> - clarify why we use ranking distribution in Fig. 2 and add absolute accuracy to evaluate TTA methods (**Weakness1**);
> - clarify the main purpose of our noise injection strategy (**Weakness2**);
> - clarify how the portion of the noisy labels influences the performance (**Weakness3**).
>
> **As the deadline for the paper discussion phase is approaching**, would you mind checking our responses and confirming whether you have any further questions?
>
> Any comments and discussions are welcome!
>
> Thanks for your attention and best regards,
> Authors of #7153

---

> ### Author Response · Authors · 2024-12-01
> **Would you mind checking our responses and confirming whether you have any further questions?**
>
> Dear Reviewer JjFG,
>
> We sincerely appreciate your valuable feedback.
>
> As the deadline for the paper discussion phase is approaching, we would like to check if you have any other remaining concerns about our paper.
>
> We sincerely thank you for your dedication and effort in evaluating our submission. Please do not hesitate to let us know if you need any clarification.
>
> Thanks for your attention and best regards,
>
> Authors of #7153

---

### Author Response · Authors · 2024-11-21
**A General Response by Authors**

**We would like to thank all the reviewers for their thoughtful suggestions on our paper.**

**We have received three reviews with ratings 6, 6, 5, and we are glad that all the reviewers have good impressions of our work**, including **(1)**  proposing a novel and practical setting (JjFG, GcDg, SvHV); **(2)** a simple, reasonable, and effective method (JjFG, GcDg);  **(3)** a comprehensive and easy to follow analysis with novel observations of existing TTA methods (JjFG, SvHV).

**In the rebuttal period, we have provided detailed responses to all the comments and questions point-by-point.** Following the reviewers' suggestions, we have revised the paper accordingly. Our main response can be summarized as:
 -  Clarify why we use ranking distribution in Fig. 2 and add absolute accuracy to evaluate TTA methods (Weakness1 for JjFG);
 - Clarify our ZS-NTTA setting, compare and discuss with test-time OOD detection (Weakness1 for GcDg);
 -  Clarify the connection between our method and our task (Weakness2 for GcDg);
 - Add experiments with more different noise ratios (Weakness3 for GcDg);
- Clarify the novelty of our method lies in decoupling the classifier and detector according to our analysis (Weakness1 for SvHV);
- Provide more experiments with more complex datasets (Weakness2 for SvHV);
- Conduct comparisons with TDA (Weakness3 for SvHV);
- Furthermore, we have updated our draft according to the reviewers' comments.

Lastly, we would appreciate all reviewers’ time again. Would you mind checking our response and confirming whether you have any further questions? **We are anticipating your post-rebuttal feedback!**

---

### Meta-Review · Area_Chair_EkBg · 2024-12-17

**Metareview:**

The paper introduces a novel research problem called zero-shot noisy test-time adaption, which aims to handle both ID and OOD test samples. The proposed method, AdaND, is simple yet effective, showing effective performance in classification and OOD detection metrics. Reviewers raised concerns about the definition of the task, technical details, experiments, and writing. The author's rebuttal successfully addressed the majority of concerns raised by the reviewers. At the end of the rebuttal, all reviewers agree to accept this paper, putting this paper above the acceptance bar.

**Additional Comments On Reviewer Discussion:**

Reviewers raised concerns about the task definition, technical details, experiments, and writing. The author's rebuttal successfully addressed the majority of these concerns. At the end of the rebuttal, all reviewers agreed to accept this paper, putting it above the acceptance bar.

---

### Decision · Program_Chairs · 2025-01-22

Accept (Poster)